# Hydrogen isotope labeling unravels origin of soil-bound organic contaminant residues in biodegradability testing

Sophie Lennartz [1,2], Harriet A. Byrne[1,3], Steffen Kümmel [4], Martin Krauss [5] & Karolina M. Nowak [4,6] ✉

Biodegradability testing in soil helps to identify safe synthetic organic chemicals but is still obscured by the formation of soil-bound 'non-extractable' residues (NERs). Present-day methodologies using radiocarbon or stable ($^{13}C$, $^{15}N$) isotope labeling cannot easily differentiate soil-bound parent chemicals or transformation products (xenoNERs) from harmless soil-bound biomolecules of microbial degraders (bioNERs). Hypothesizing a minimal retention of hydrogen in biomolecules, we here apply stable hydrogen isotope – deuterium (D) – labeling to unravel the origin of NERs. Soil biodegradation tests with D- and $^{13}C$-labeled 2,4-D, glyphosate and sulfamethoxazole reveal consistently lower proportions of applied D than $^{13}C$ in total NERs and in amino acids, a quantitative biomarker for bioNERs. Soil-bound D thus mostly represents xenoNERs and not bioNERs, enabling an efficient quantification of xenoNERs by just measuring the total bound D. D or tritium (T) labeling could thus improve the value of biodegradability testing results for diverse organic chemicals forming soil-bound residues.

Synthetic organic chemicals convey numerous benefits in daily life but may end up in soils after intentional or unintentional discharge[1,2]. Depending on the soil and compound properties, organic contaminants can undergo different binding processes to the soil, microbial degradation, or non-biological degradation influenced by abiotic factors such as sunlight, temperature, and pH[3–7]. Compared to water and air, assessing the fate of organic contaminants in soil is complicated by their binding to or entrapment in the solid matrix, making soils a potential 'temporary reservoir' or 'permanent sink' for contaminants[2,8]. However, organic contaminants that are only loosely attached or weakly bound to soil may relocate from soils to food crops, water, and air, putting human and ecosystem health at risk[1,2,9]. Contrastingly, the permanent binding of organic contaminants to soil or their mineralization to $CO_2$ and $H_2O$, together with assimilation into

microbial biomass, can greatly reduce this risk[2,6,10,11]. However, potentially 'safe' synthetic organic chemicals first have to be identified in biodegradability tests prior to their use for societal needs[12–14].

Biodegradability testing in soil is conducted as part of regulatory safety assessment frameworks around the world, e.g., by the US Environmental Protection Agency[15] or European Chemicals Agency (ECHA)[16]. Chemical (bio)degradation is simulated by applying heavy isotope-labeled analogs of a test substance (usually using $^{14}C$-tracers) to a reference soil and quantifying the isotope mass balance (see Fig. 1).

This mass balance comprises the isotope label in the mineralized fraction ($CO_2$, $H_2O$, or $N_2$), mobile solvent-extractable fraction (chemical and its transformation products), and in the immobile solid fraction operationally defined as 'bound residues' or 'non-extractable residues' (NERs). While the mineralized and mobile solvent-extractable

[1]Department of Molecular Environmental Biotechnology, Helmholtz Centre for Environmental Research – UFZ, Leipzig, Germany. [2]Department of Environmental Science, Aarhus University, Roskilde, Denmark. [3]Department of Environmental Analytical Chemistry, Helmholtz Centre for Environmental Research – UFZ, Leipzig, Germany. [4]Department of Technical Biogeochemistry, Helmholtz Centre for Environmental Research – UFZ, Leipzig, Germany. [5]Department of Exposure Science, Helmholtz Centre for Environmental Research – UFZ, Leipzig, Germany. [6]Chair of Geobiotechnology, Institute of Biotechnology, Technische Universität Berlin, Berlin, Germany. ✉e-mail: karolina.nowak@ufz.de

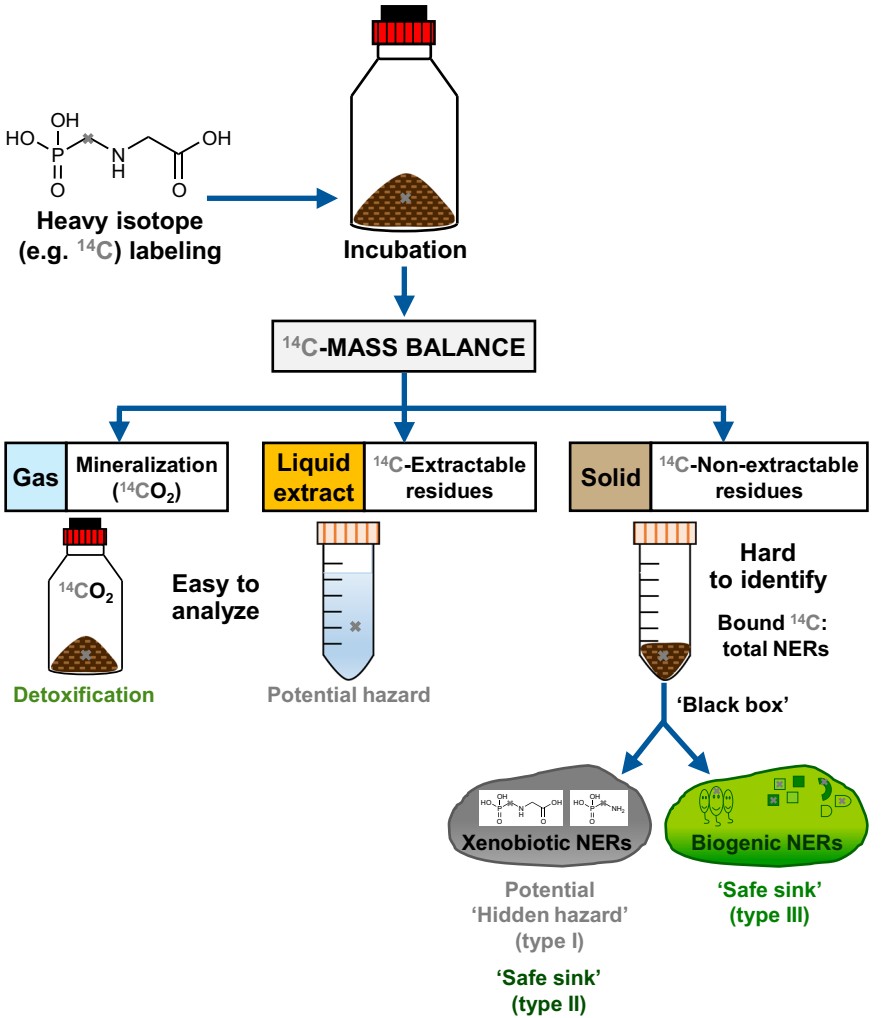

**Fig. 1 | Biodegradability testing of the model compound glyphosate in soil using heavy isotope labeling (here $^{14}$C) according to OECD (Organization for Economic Co-operation and Development) guideline 307[16].** The $^{14}$C-mass balance comprises easily quantifiable mineralization ($^{14}CO_2$), the $^{14}$C-labeled solvent-extractable chemical or its transformation products, as well as the difficult to identify soil-bound $^{14}$C as non-extractable residues (NERs). The $^{14}C_{NERs}$ are mostly quantified as total NERs that contain both the untransformed chemical and its transformation products entrapped in or bound to soil as xenobiotic NERs (xenoNERs) as well as biomolecules derived from microbial biodegradation as biogenic NERs (bioNERs). Mineralization of the compound leads to its ultimate detoxification, while extractable and thus potentially mobile chemical residues could relocate to other environmental compartments and cause toxic effects. In contrast, the hazard of NERs is not clear as they can be 'safe sink' bioNERs type III and xenoNERs type II or potential 'hidden hazard' xenoNERs type I.

fractions are easily analyzable, the NERs are a 'black box' in biodegradability testing as three types of NERs can be formed, which carry different hazards[7,17–21].

The first two NER types are xenobiotic NERs (xenoNERs) comprising the un-degraded parent chemical or its transformation products that are physically entrapped in soil (type I) or covalently bound to reactive surface groups of the soil (type II)[17–20]. In the environment, adsorbed chemicals and type I xenoNERs can detach from soil over varying time scales and thus pose a delayed 'hidden hazard'[17–19]. Changing environmental conditions, e.g., freezing, thawing, and altered precipitation, temperature, pH, or vegetation cover due to global warming could all exacerbate the release of potentially harmful type I xenoNERs[21–24]. To identify potentially mobile residues in laboratory tests, different 'soft' and 'harsh' extraction schemes are advocated, e.g., a mixture of aqueous and organic solvents along with physical agitation, heat, or pressure[17,19,20]. The physically entrapped type I xenoNERs are freed after the breakdown of soil aggregates using silylation or extraction with EDTA, while xenoNERs type II can only be released after bond breakage using acidic or alkaline hydrolysis[19,25,26]. The covalent bonds between the chemical and reactive groups of soil

are considered to be hardly breakable, xenoNERs type II are thus a 'safe sink'[17–19].

Next to xenoNERs, type III biogenic NERs (bioNERs) result from the integration of C, N, or H isotope labels of a biodegraded test chemical into the biomass of microorganisms[7,20,27,28]. This third NER type is identical to biomolecules produced naturally in soils and therefore considered as a permanent 'safe sink' for organic contaminants[7,17–20,27–31]. BioNERs comprise both living biomass and microbial biomass residues at various stages of decay that eventually get stabilized in the soil matrix[7,19,28]. NERs measured in laboratory tests thus contain both type I and type II xenoNERs as well as type III bioNERs that are weakly to strongly bound depending on the extraction method used, impeding the distinction between these three types of NERs.

Although differentiation between xenoNERs (type I and type II) and bioNERs (type III) is vital to assess the hazard posed by organic contaminants in soil, it is not part of routine biodegradability tests with $^{14}$C tracers, which readily quantify only the total $^{14}C_{NERs}$ (Fig. 1)[15–18,29,32,33]. Well-established protocols exist for the identification and rough quantification of bioNERs based on the analysis of stable carbon ($^{13}$C)

or nitrogen ($^{15}$N) isotope tracers in biomolecules; however, they are very laborious. Both $^{13}$C or $^{15}$N-labeling is also limited by the scarce supply and high costs of $^{13}$C or $^{15}$N-labeled compounds, and $^{15}$N is additionally restricted to N-containing molecules[7,20,34–36]. We here show that the stable H isotope – D – can be used as an alternative to existing tracers that are both cheaper and more accessible than $^{13}$C or $^{15}$N due to the common use of deuterated standards in analytical chemistry[37]. Furthermore, hypothesizing minimal retention of H in biomolecules and thus in bioNERs, H-labeling could enable a time-efficient distinction between xenoNERs and bioNERs.

Because of different turnover dynamics, H is expected to be much less retained in microbial biomass than C. In a prior one-year incubation study, for example, substrate-derived H in soil and microbial lipids was respectively 6-fold and over 10-fold lower compared to substrate-C[38]. The main reason is that total and bioavailable C is limited in soil, whereas H – due to its presence in soil water – is ~ 6–11 times more abundant considering the stochiometric requirements to build different biomolecules (see Supplementary Note 1). Microorganisms, therefore, constantly recycle C-substrates and necromass (biomass residues) of primary degraders for both the energy contained in C–H bonds (catabolism) and for C-building blocks required for biomass synthesis (anabolism) (Fig. 2)[39–43]. Substrate- or necromass-derived C can also be directly assimilated as a C-monomer (e.g., amino acid) into macromolecules (e.g., proteins). Moreover, also unutilized C in decaying microbial biomass is eventually stabilized in the soil matrix as bioNERs[8,27,44,45]. C can be slowly released from decaying soil biomass as $CO_2$ (Fig. 2) before re-uptake by newly growing microorganisms during $CO_2$ fixation[8,27,44,45] (accounting for ~ 4% of released $CO_2$ from 2,4-D[46]). Therefore, during microbial degradation of a $^{13}$C-labeled substrate, a high retention of the $^{13}$C tracer in bioNERs is generally observed.

In contrast, the direct incorporation of D from a D-labeled organic substrate into microbial biomass is expected to be low. After catabolic C–D bond cleavage, substrate-derived D is released from coenzymes to ambient water within a few minutes[47] (Fig. 2), as explained in more detail in Supplementary Fig. 7. While this process is fast, the dynamics of enzymatic C–D bond cleavage will vary depending on the biodegradability of the D-substrate. For example, in monomeric biomolecules like glucose or amino acids, 70% of the C–D bonds were broken already within the first 7 days of soil incubation[38]. Besides substrate-H, ambient water provides a highly abundant H source for the de novo formation of C–H bonds in C-monomers during anabolism[48–50]. In prior studies with heavy water ($D_2O$), up to 79% of the D-water was assimilated into microbial biomass[49], and incorporation was already visible after 20 min[51]. However, due to the strong dilution of the substrate-derived D with ambient water-H, potentially combined with isotopic fractionation[50,52,53], D reuptake into newly synthesized biomass and thus into bioNERs will be low. The dilution of $D_2O$ with ambient water varies depending on the substrate concentration, biodegradation pathways, and soil water contents but was estimated to result in D concentrations only about 0.001% above its natural abundance (0.015 at%) for substrate concentrations between 10–50 mg kg$^{-1}$ dry soil (Supplementary Note 2). High retention of substrate-D in bioNERs could only occur during the assimilation of D-monomers retaining C–D bonds of the primary D-substrate into macromolecules (e.g., glycine formed from glyphosate; Fig. 2).

Thus, we hypothesized that D-tracers would be minimally retained in bioNERs. If a bound D-chemical is not degraded by microorganisms, it will be contained in the soil as xenoNERs. Consequently, almost all of the total NERs measured using the D-labeling approach will be xenoNERs. D-labeling could thus allow for the expensive and laborious bioNER assessment to be skipped. To test the feasibility of D-labeling for a time-efficient xenoNER identification, soil incubations oriented at OECD (Organization for Economic Co-operation and Development) guideline 307[16] were performed with three compounds of environmental concern using both a D- and

$^{13}$C-labeling approach. The $^{13}$C-labeling served for the comparison of biogenic and total NER formation between H- and C-isotope tracers. We selected the herbicides 2,4-dichlorophenoxyacetic acid (2,4-D) and glyphosate (GLP) and the antibiotic sulfamethoxazole (SMX) as model compounds due to their different chemical structures allowing isotope labeling at aliphatic (GLP) and aromatic (2,4-D and SMX) moieties, and their different biodegradability in soils: fast for 2,4-D[54–57], medium for GLP[58–60] and low for SMX[61,62]. GLP was labeled with two D atoms at two different labeling positions (2-C–$D_{2\text{-GLP}}$ or 3-C–$D_{2\text{-GLP}}$) to cover differences between its two key degradation pathways via sarcosine or aminomethylphosphonic acid (AMPA, see Supplementary Figs. 8, 9)[36]. 2,4-D ($D_{3\text{-}2,4\text{-D}}$) and SMX ($D_{4\text{-SMX}}$) were labeled in the aromatic ring with three and four D atoms, respectively. The $^{13}$C-compounds were labeled at similar positions as their D analogs, i.e., GLP had only one $^{13}$C (2-$^{13}C_{GLP}$, 3-$^{13}C_{GLP}$), whereas all carbons in the aromatic ring of both 2,4-D ($^{13}C_{6\text{-}2,4\text{-D}}$) and SMX ($^{13}C_{6\text{-SMX}}$) were labeled with $^{13}$C. We analyzed the D and $^{13}$C incorporation into total NERs (bioNERs + xenoNERs) and into amino acids (AAs) as a quantitative biomarker for bioNERs in biologically active soils. We also hypothesized that the C–D bonds of all tested compounds are stable against abiotic cleavage, which is essential for the validity of the D-labeling approach. To this end, we incubated sterile soils and ultra-pure (Milli-Q) water with the same D- and $^{13}$C-labeled compounds. Our findings provide the first proof of concept that D is hardly incorporated into bioNERs compared to $^{13}$C, which could help to improve current biodegradability testing strategies in soil.

## Results

Total $^{13}C_{NERs}$ and $D_{NERs}$ were quantified by elemental analyzer-isotope ratio mass spectrometry (EA-IRMS) as the amount of isotope label remaining in the soil after solvent-extraction of the test compounds. The extraction efficiencies of the tested compounds from the soil directly after spiking were as follows: 98 ± 5% (2,4-D), 93 ± 3% (GLP), and 80 ± 10% (SMX; for details, see Supplementary Note 3). However, the $^{13}$C and $D_{NERs}$ measured on sampling day 0 in both sterile (Fig. 3) and biologically active soils (Supplementary Note 3) suggest a lower extraction efficiency, especially of GLP and SMX. Due to the laborious preparation of all parallel experimental treatments for incubation, it was impossible to perform soil extractions immediately after spiking of the test compounds (Supplementary Note 3). Therefore, we cannot exclude abiotic NER formation already in soils sampled on day 0. Moreover, in order to minimize the risk of potential release of the D from C–D bonds, we skipped the final 'harsh' extraction step mandated by ECHA, which uses heat or pressure aiming at the extraction of 'slowly desorbable' residues[19]. The total $^{13}$C and $D_{NERs}$ from 2,4-D, GLP, and SMX (Figs. 3, 4) thus comprised the sum of bioNERs, xenoNERs type I & II, and possibly also 'slowly desorbable' residues[29] which inflate total NER estimates. Although the 'slowly desorbable' residues are not considered 'NERs'[19], we kept the NER term for simplicity here. Please note that the current study was not performed for regulatory purposes but for proof of concept, and hence, it did not aim to accurately follow all extraction schemes or soil incubation conditions outlined in OECD guideline 307[16]. The aim of this study was to compare: (I) both the amounts of $^{13}$C- and D-compounds in ultra-pure water and the $^{13}C_{NERs}$ and $D_{NERs}$ (as defined by the employed methodology) in sterile soils (hypothesis 1: stable C–D bonds under abiotic conditions), and (II) the $^{13}$C and D incorporation into bioNERs in biologically active soils (hypothesis 2: minimal retention of D in bioNERs), using the same experimental conditions and extraction protocols for both H and C tracers.

### Total $^{13}$C- and D-labeled non-extractable residues (NERs) in sterile soils

Abiotically formed $^{13}C_{NERs}$ and $D_{NERs}$ of each model compound were nearly identical at all sampling dates (Fig. 3a–c; for detailed statistics,

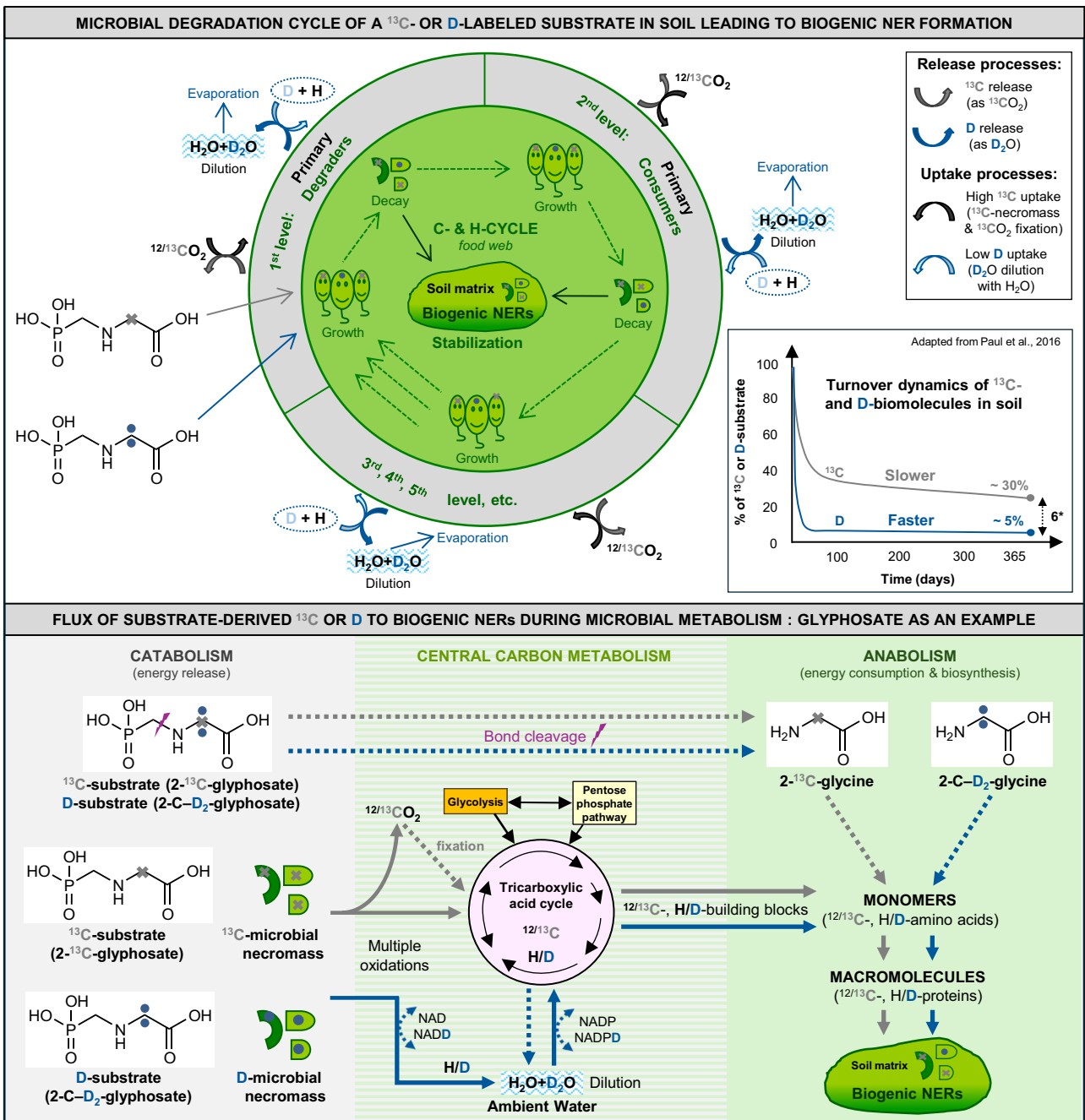

**Fig. 2 | Formation of bioNERs during the biodegradation of a heavy C- or H-labeled substrate (here: 2-$^{13}$C- and 2-C–$D_2$-glyphosate).** Please note that the same processes apply to $^{14}$C and T as well, but are for simplicity only shown for $^{13}$C/ D. The C/H-cycle starts with the breakdown of C–D bonds of the C–D-substrate by degrading microorganisms in multiple oxidation steps (catabolism) to free both energy and C($^{12}$C/$^{13}$C)-building blocks needed for biomass synthesis (anabolism). The C-building blocks mined during the central C-metabolism (e.g., acetyl-groups generated from the pyruvate produced by the glycolysis pathway[48]) are integrated into C-monomers during anabolism (1st level). Contrastingly, after C–D bonds breakage, most of the substrate-D is first lost via the coenzyme nicotinamide adenine dinucleotide (NAD(D)) into ambient water, where it is diluted with unlabeled H. Then, the H/D is transferred by the coenzyme nicotinamide adenine dinucleotide phosphate (NADP(D)) to C-building blocks during de novo C–H bonds

formation of C-monomers in anabolism[49]. When biomass decays, the necromass is either assimilated by other living microorganisms (primary consumers) or stabilized in the soil matrix forming bioNERs. A direct assimilation of C-monomers (here glycine) into microbial macromolecules is also possible after a partial breakdown of C-substrates (like glyphosate) or C-necromass. $^{13}$CO_2 is slowly released from the soil matrix during microbial degradation at each trophic level, whereas $D_2$O is lost more rapidly and hardly taken up again due to an estimated 100,000 times dilution with unlabeled $H_2$O. Hence the retention of an easily biodegradable H-substrate in soil after one year is about sixfold lower (5%) than for C (30%); see small inserted Fig. adapted from Paul et al.[38] according to a CC BY 3.0 license. The N-cycle is similar to the C-cycle except that N-substrates are mineralized to gaseous $N_2$O or $N_2$ and the substrate-derived N (e.g., a $NH_2$-group) is transferred to C-monomers (e.g., amino acids) in anabolism; therefore, it is not shown here.

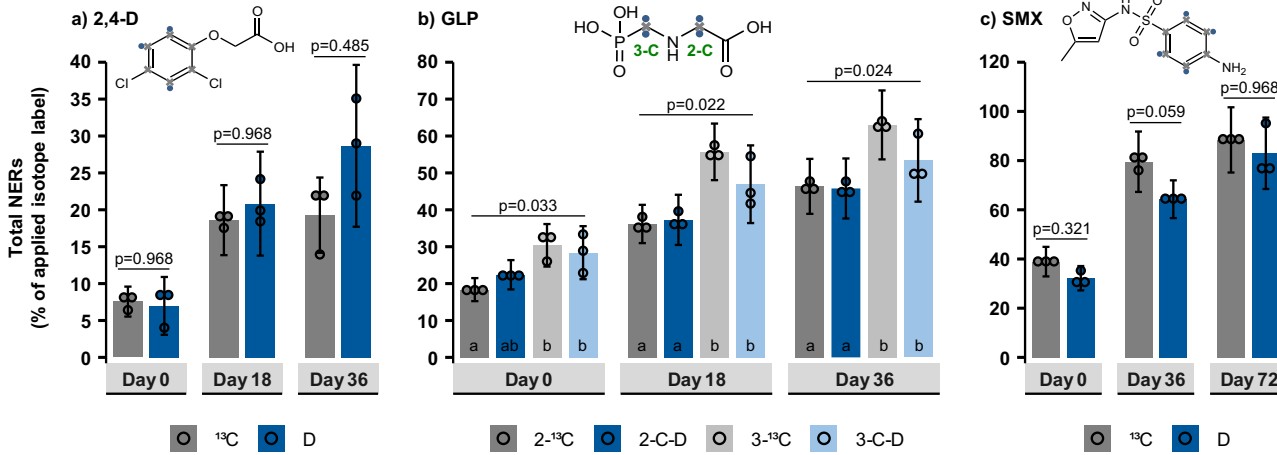

**Fig. 3 | Total NER formation of the three model compounds in sterile soils.**
**a** 2,4-dichlorophenoxyacetic acid (2,4-D), (**b**) glyphosate (GLP; 2-C and 3-C labeling position), and (**c**) sulfamethoxazole (SMX). Columns represent the average of independently processed replicate samples ($n = 3$), and error bars the propagated standard deviation considering the uncertainty of the total element (C/H) abundance as well as the isotopic abundance (at% $^{13}C/^{12}C$ or at% $D/^1H$) in the labeled treatment and unlabeled background. Corresponding formulas are provided in the methods section. Dots show the individual replicate values used for statistical analysis. These only represent the uncertainty of the isotopic abundance in the labeled treatment, as the mean element abundances and mean isotopic background values were used for the calculation of replicate values. Holm-adjusted $p$-values for Welch tests (2,4-D, SMX) or main effects of Kruskal-Wallis tests (GLP) are shown above the bars. Letters indicate statistically significant differences in Conover-Iman tests ($p_{adjusted} < 0.05$), and exact $p$-values for GLP are provided in Supplementary Note 4.1.1.

see Supplementary Note 4), proving that D was stable against abiotic breakage of C–D bonds. On the final date, the lowest NERs were observed for 2,4-D ($^{13}C_{NERs}$: 19 ± 5.1%; $D_{NERs}$: 29 ± 11%), followed by 2-C$_{GLP}$ ($^{13}C_{NERs}$: 46 ± 7.4%; $D_{NERs}$: 46 ± 8.1%) and 3-C$_{GLP}$ ($^{13}C_{NERs}$: 63 ± 9.3%; $D_{NERs}$: 53 ± 11%) while the highest NERs were formed by SMX ($^{13}C_{NERs}$: 88 ± 13%; $D_{NERs}$: 83 ± 15%). We noticed that the label position of GLP (2-C$_{GLP}$ and 3-C$_{GLP}$) slightly affected the $^{13}C/D_{NERs}$, which were somewhat higher for 3-C$_{GLP}$ than for 2-C$_{GLP}$ (Supplementary Note 4.1.1). However, this divergence was consistent at all sampling dates and might be due to slightly different amounts of added GLP on day 0.

### Total $^{13}C$- and D-labeled NERs in biologically active soils
As hypothesized, for all three model compounds the amount of total $^{13}C_{NERs}$ in biologically active soil was higher than that of total $D_{NERs}$ throughout the incubation periods (Fig. 4).

**2,4-D**. About 4–5 times higher contents of $^{13}C_{NERs}$ from $^{13}C_{6-2,4-D}$ were measured compared to $D_{NERs}$ from $D_{3-2,4-D}$, demonstrating much lower retention of D in the total NER pool compared to $^{13}C$ (Fig. 4a). The $^{13}C_{NERs}$ on both day 16 (20 ± 6.0% of the applied $^{13}C$) and 36 (14 ± 4.7% of the applied $^{13}C$) were significantly higher than the $D_{NERs}$ on day 16 (5.3 ± 2.6% of the applied D; $p_{adjusted} = 0.002$; Supplementary Table 3 in Supplementary Note 4.4) and day 36 (3.2 ± 2.1% of the applied D; $p_{adjusted} = 0.017$). The $^{13}C_{NERs}$ from $^{13}C_{6-2,4-D}$ were lower than previously reported for $^{14}C_{6-2,4-D}$ (26 ± 0.2%)[63] and $^{13}C_{6-2,4-D}$ (39 ± 2.6%)[57] in soils with similar properties.

Both the total $^{13}C_{NERs}$ and $D_{NERs}$ in biologically active soils were either comparable ($^{13}C$: day 16/18 & 36) or lower (D: day 16/18 & 36) than in sterile soils (Fig. 3a). Although this finding might seem to contradict the statement that 'if abiotically formed NERs are much lower than biotically formed NERs, this gives a clear indication on bioNER formation' in OECD guideline 307[16], bioNER formation does not necessitate that abiotic NERs are lower than NERs in biologically active soil. This is because, in biologically active soil, abiotic interactions leading to xenoNER formation compete with simultaneously proceeding biodegradation processes, which include both bioNER formation and complete mineralization of the compound. Thus, when a compound is quickly mineralized in biologically active soil, it may be removed before it can form xenobiotic NERs. In sterile soil, no mineralization occurs, and thus total abiotic NERs may be higher than biotic NERs for readily biodegradable compounds. Interestingly, in a previous study by Girardi et al.[57], the $^{13}C_{NERs}$ from $^{13}C_{6-2,4-D}$ in sterile soils (15 ± 1.8%) were lower than in biologically active soil (39 ± 2.6%). However, much less $^{13}C_{6-2,4-D}$ was mineralized in the study by Girardi et al.[57] (46 ± 2.9% after 32 days) than in this study (78 ± 8.8% after 36 days, see Supplementary Table 4 in Supplementary Note 5), suggesting that the rapid biodegradation of 2,4-D in our study prevented formation of xenoNERs in the biologically active soil.

**GLP**. The total NER formation of GLP depended on both the type of isotope tracer and the labeling position. About 2–5 times higher $^{13}C_{NER}$ contents were measured compared to $D_{NERs}$ on day 38 (Fig. 4b), resulting in moderate to high amounts of $^{13}C_{NERs}$ (2-$^{13}C_{GLP}$: 62 ± 4.7%; 3–$^{13}C_{GLP}$: 32 ± 3.8% of the applied $^{13}C$) but only relatively low amounts of $D_{NERs}$ (2-C–$D_{2-GLP}$: 12 ± 2.8% of the applied D; 3-C–$D_{2-GLP}$: 17 ± 3.4% of the applied D). The differences between $^{13}C_{NERs}$ and $D_{NERs}$ on day 38 were significant for both labeling positions (2-C$_{GLP}$: $p_{adjusted} = 0.000004$; 3-C$_{GLP}$: $p_{adjusted} = 0.0063$; see Supplementary Note 4.1.2). Notably, the $^{13}C_{NERs}$ from 2-$^{13}C_{GLP}$ amounted to 66 ± 4.8% on day 4 and remained fairly stable until day 38. In contrast, $D_{NERs}$ from 2-C–$D_{2-GLP}$ were twice lower already on day 4 (30 ± 5.3%, $p_{adjusted} = 0.000004$) and decreased by ~ 50% until the final day 38, showing a continuous release of D from the total NER pool. In the case of 3-C$_{GLP}$, between day 4 and day 38, the $^{13}C_{NERs}$ decreased from 45 ± 4.8% to 32 ± 3.8%, and the $D_{NERs}$ from 37 ± 6.8% to 17 ± 3.4%. The differences in NER formation between D and $^{13}C$ tracers were thus less pronounced for 3-C$_{GLP}$. Overall, $^{13}C_{NERs}$ from both 2- and 3-$^{13}C_{GLP}$ were close to the previously reported 40-50% of initially added $^{13}C$[59,64] as well as ~ 40% of initially, added $^{14}C$[65].

The total $^{13}C_{NERs}$ from 2-$^{13}C_{GLP}$ were higher in biologically active soil than in sterile soil, while the $^{13}C_{NERs}$ from 3-$^{13}C_{GLP}$ were lower (Fig. 3b). Contrastingly, the $D_{NERs}$ from both D-labeled GLP analogs were lower than the abiotically formed $D_{NERs}$. Mineralization of GLP was similar between 3-$^{13}C_{GLP}$ (50 ± 17%; Supplementary Table 4) and 2-$^{13}C_{GLP}$ (40 ± 12%), suggesting at first glance a comparable biodegradation of GLP labeled at the 2-C and 3-C positions. However, the

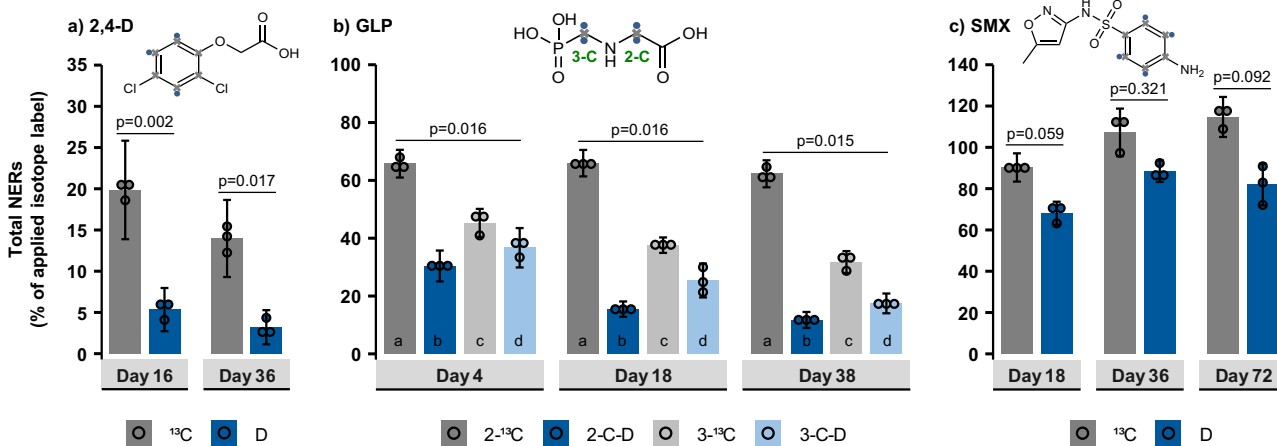

**Fig. 4 | Total NER formation of the three model compounds in biologically active soils. a** 2,4-D, (**b**) GLP (2-C and 3-C labeling position), and (**c**) SMX. Columns represent the average of independently processed replicates ($n = 3$), error bars the propagated standard deviation, and dots individual replicates used for statistical analysis. Holm-adjusted $p$-values for Welch tests (2,4-D, SMX) or Kruskal-Wallis tests (GLP) are shown above the columns. Letters indicate statistically significant differences in post-hoc Conover-Iman tests ($p_{adjusted} < 0.05$), and exact $p$-values for GLP are provided in Supplementary Note 4.1.2.

mineralization of a compound is not always the only factor dictating the total amounts of $^{13}C_{(bio)NERs}$, as shown for $^{13}C_{GLP}$. The labeling position of GLP played a crucial role here, as detailed in the following sections.

**SMX.** SMX formed very high amounts of total NERs (Fig. 4c). The $D_{NERs}$ ($68 \pm 5.6\%$ of the applied D) appeared slightly lower than $^{13}C_{NERs}$ ($90 \pm 6.8\%$ of the applied $^{13}C$) on day 18 ($p_{adjusted} = 0.059$), but no significant differences in NER contents were found between D and $^{13}C$ tracers on any sampling day ($p_{adjusted} > 0.05$, Fig. 4c). The $^{13}C_{NERs}$ peaked at $107 \pm 12$ to $115 \pm 9.7\%$ and the $D_{NERs}$ at $88 \pm 5.2$ to $82 \pm 10\%$ on day 36–72. Notably, higher $^{13}C_{NER}$ values compared to $D_{NERs}$ might be due to a higher amount of $^{13}C_{6-SMX}$ accidentally added to the soil on day 0 (total $^{13}C$-label recovery on day 0: 138%, Supplementary Table 2b in Supplementary Note 3). SMX is a hardly biodegradable antibiotic (only $2.3 \pm 0.5\%$ $^{13}CO_2$ after 72 days, Supplementary Table 4) expected to form mostly xenoNERs; thus, the $^{13}C_{NERs}$ in the biotic treatment should be nearly identical to the $^{13}C_{NERs}$ and $D_{NERs}$ measured in sterile soil (Fig. 3c). The overall high NER content are comparable to $^{14}C_{NER}$ formation from $^{14}C_{6-SMX}$ reported in prior studies[61,62].

**Proportion of $^{13}C$- and D-biogenic NERs (bioNERs) within the total NERs**

Total bioNER contents (Fig. 5) were estimated from total amino acids (tAAs) hydrolyzed from soil. The $^{13}C_{tAAs}$ and $D_{tAAs}$ were multiplied by 2 since tAAs makeup roughly 50–55% of microbial biomass and can thus be used as a quantitative biomarker for bioNERs[7,31]. Due to the uncertainty of this quantitation method, we additionally calculated $^{13}C_{bioNERs}$ for 2,4-D, GLP and SMX based on the released $^{13}CO_2$ (Supplementary Table 4) using the microbial turnover to biomass (MTB) approach[28] (Supplementary Note 5).

**2,4-D.** Total $^{13}C$-amino acids ($^{13}C_{tAAs}$) on day 16 ($11 \pm 1.0\%$ of the applied $^{13}C$) and 36 ($9.8 \pm 1.6\%$ of the applied $^{13}C$) were 6–7-fold higher than the total D-amino acids ($D_{tAAs}$) on both day 16 ($1.6 \pm 0.2\%$ of the applied D) and 36 ($1.7 \pm 0.5\%$ of the applied D; Fig. 5a). These differences between $D_{tAAs}$ and $^{13}C_{tAAs}$ were statistically significant at both time points (day 16: $p_{adjusted} = 0.037$; day 36: $p_{adjusted} = 0.035$). As hypothesized, the total amounts of $D_{bioNERs}$ from $D_{3-2,4-D}$ (tAAs + other bioNERs) were thus lower (six to seven times) than those of the $^{13}C_{bioNERs}$. The $^{13}C_{bioNERs}$ were nearly identical on day 16 ($22 \pm 2.0\%$ of the applied $^{13}C$)

and 36 ($20 \pm 3.2\%$ of the applied $^{13}C$). Also, the amounts of $D_{bioNERs}$ were comparable between day 16 ($3.2 \pm 0.4\%$ of applied D) and 36 ($3.4 \pm 1.0\%$ of applied D). Notably, these estimates of total $^{13}C_{bioNERs}$ and $D_{bioNERs}$ on both days 16 and 36 were nearly equal to the total measured $^{13}C_{NERs}$ and $D_{NERs}$ shown in Fig. 4a. Moreover, the $^{13}C_{bioNERs}$ determined as tAAs*2 fell within the range of predicted bioNERs based on the MTB model (13.5–32.6%, see Supplementary Table 6 in Supplementary Note 5), showing consistent estimates from both approaches. On day 36, tAA contents without application of the conversion factor (~10% of applied $^{13}C$) already made up the majority of the total $^{13}C_{NERs}$ ($14 \pm 4.7\%$ of applied $^{13}C$ on day 36, Fig. 4a), while the minimum predicted bioNERs based on MBT calculations (bioNER$_{min}$: 13.5%) were nearly identical to total $^{13}C_{NERs}$. Since 2,4-D is known to be easily biodegradable[35], it is thus likely that for both $^{13}C_{6-2,4-D}$ and $D_{3-2,4-D}$, the total $^{13}C_{NERs}$ and $D_{NERs}$ could be completely ascribed to 'safe sink' bioNERs on day 36.

**GLP.** In the GLP experiment, both $^{13}C_{tAAs}$ and $^{13}C_{bioNERs}$ were higher than their D-analogs (Fig. 5b). Furthermore, we observed a notable difference in the labeling pattern of tAAs and bioNERs with $^{13}C$ and D between the two GLP labeling positions (2-$C_{GLP}$ and 3-$C_{GLP}$).

In the case of 2-$C_{GLP}$, $^{13}C_{tAAs}$ on day 4 ($10 \pm 3.1\%$ of the applied $^{13}C$) and 18 ($12 \pm 1.6\%$ of the applied $^{13}C$) were about fourfold higher than $D_{tAAs}$ on both day 4 ($2.4 \pm 0.6\%$ of the applied D; $p_{adjusted} = 0.019$; Supplementary Note 4.1.3) and 18 ($3.4 \pm 0.6\%$ of the applied D; $p_{adjusted} = 0.024$). The total $^{13}C_{bioNERs}$ increased to $24 \pm 3.2\%$ of the applied $^{13}C$ on day 18 and then remained relatively stable until day 38, whilst $D_{bioNERs}$ were nearly identical at all time points ($5–7 \pm 1\%$ of the applied D). At the end of the incubation period, the $D_{bioNERs}$ were thus surprisingly only thrice lower than the $^{13}C_{bioNERs}$ (with no significant differences between $D_{tAA}$ and $^{13}C_{tAA}$ contents; $p_{adjusted} = 0.137$), comprising about 60% of the total $D_{NERs}$. In comparison, the total $^{13}C_{bioNERs}$ made up only about 30% of the total $^{13}C_{NERs}$ on day 38. Notably, predicted bioNERs from 2-$^{13}C_{GLP}$ based on the MTB model were somewhat lower than based on tAAs*2 – ranging from 6.7% to 16% for degradation via the sarcosine pathway and 2.8–6% via the AMPA pathway (Supplementary Table 6). This can be explained by the monomeric assimilation of $^{13}C_{glycine}$ during the biodegradation of 2-$^{13}C_{GLP}$ via the sarcosine pathway (for details, see the following section), which is not accounted for in the MTB model[66].

BioNER formation from 3-$C_{GLP}$ was much lower than from 2-$C_{GLP}$. Significantly higher $^{13}C_{tAAs}$ than $D_{tAAs}$ were measured on day

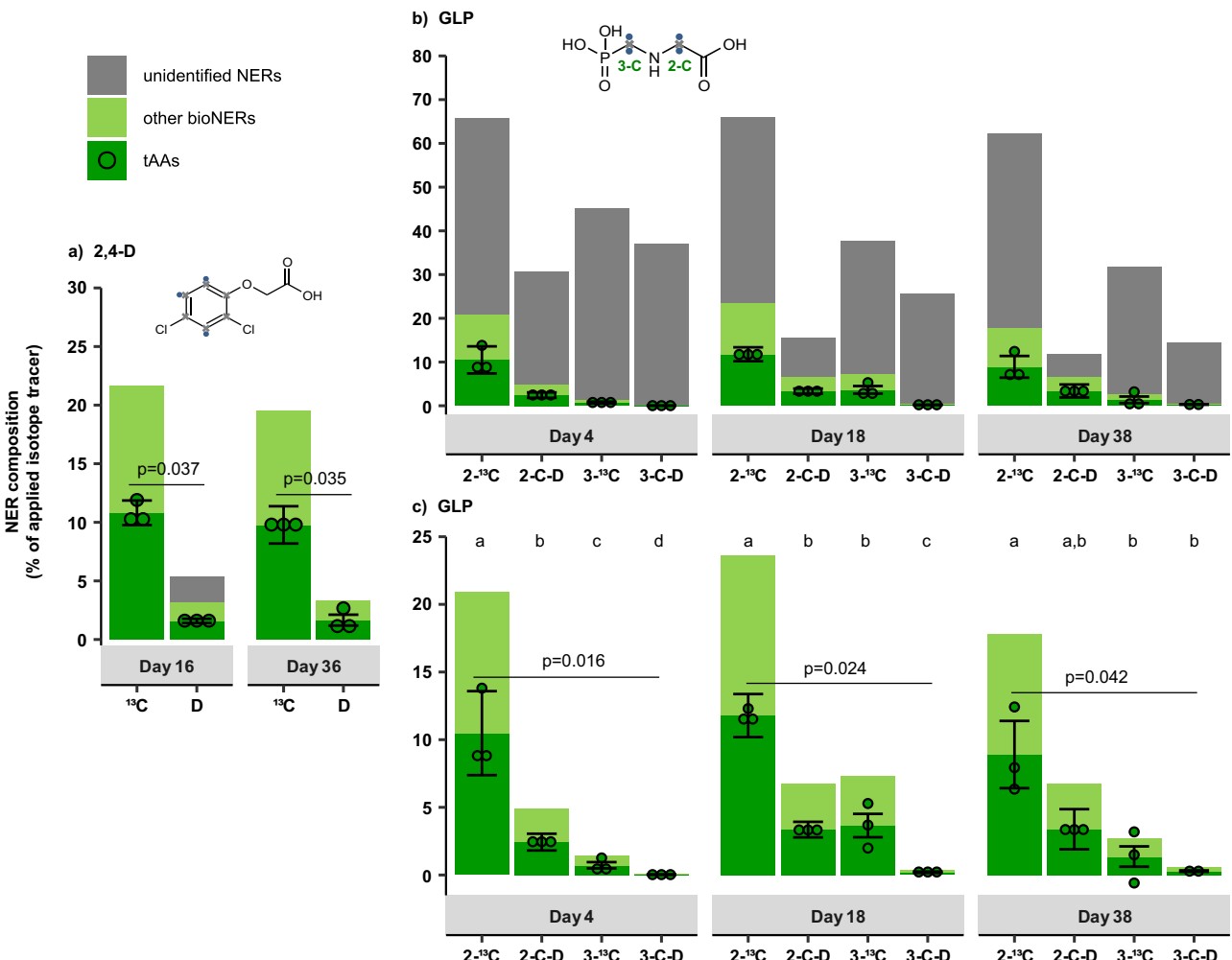

**Fig. 5 | Total NER and bioNER formation of 2,4-D and GLP. a** $^{13}C_{6-2,4-D}$ and $D_{3-2,4-D}$ (**b**) $^{13}C_{GLP}$ and $D_{2-GLP}$ (2-C and 3-C position, (**c**) figure zoomed-in). BioNERs were calculated from the measured contents of total amino acids (tAAs) hydrolyzed from proteins multiplied by 2 (tAAs*2), whereas unidentified NERs are the difference between the total NERs and the total bioNERs (total NERs – total bioNERs). Columns represent the mean of independently processed replicates ($n$ = 3, except 3-C–$D_{2-GLP}$ on day 38, where $n$ = 2) and error bars the propagated standard deviation of measured tAAs. Dots show individual replicates for statistical analysis, with one

negative value for 3-$^{13}C_{GLP}$ on day 38 due to a measured isotopic abundance in the labeled treatment below background levels (for details on calculations, see Data analysis in the Methods section). Holm-adjusted $p$-values for Welch tests (2,4-D) and Kruskal-Wallis tests (GLP) are shown above the columns. Letters indicate statistically significant differences in Conover-Iman post-hoc tests ($p_{adjusted} < 0.05$), and exact $p$-values for GLP are provided in Supplementary Note 4.1.3. No $^{13}C$ and $D_{tAAs}$ were found for SMX ($^{13}C_{6-SMX}$ and $D_{4-SMX}$); thus, total NERs from SMX were only xenobiotic NERs.

4 ($p_{adjusted}$ = 0.006; Supplementary Note 4.1.3) and day 18 ($p_{adjusted}$ = 0.019). On the final day, a fivefold higher formation of $^{13}C_{bioNERs}$ (2.7 ± 1.5% of the applied $^{13}C$) than $D_{bioNERs}$ was observed (0.6 ± 0.1% of the applied D; Fig. 5b), although this difference was not statistically significant ($^{13}C_{tAAs}$ vs $D_{tAAs}$: $p_{adjusted}$ = 0.473). The predicted bioNER$_{min}$ using the MTB model were only slightly higher than the tAAs*2 when considering the AMPA pathway (3.5–7.4%; Supplementary Table 6) but overestimated when based on the sarcosine pathway (8.2–19.5%). This may be because bioNERs based on tAAs*2 depend on the $^{13}C$-labeling position, whereas the MTB approach considers the turnover of all C atoms of the substrate equally[28].

**SMX.** No $^{13}C$ from $^{13}C_{6-SMX}$ or D from $D_{4-SMX}$ could be detected in the $^{13}C_{tAAs}$ and $D_{tAAs}$, suggesting that the NERs from SMX were almost exclusively xenoNERs. This is in line with low mineralization (2.3 ± 0.5% cumulative $^{13}CO_2$ evolution after 72 days, Supplementary Table 4) and negligible amounts of predicted bioNERs based on the MTB approach (min: 0.6%, max: 1.5%).

## Labeling position of GLP and its relevance for the $^{13}C$- and D-labeling pattern of bioNERs

**2-C-labeling position (2-C$_{GLP}$).** A closer look at the composition of $^{13}C$ and $D_{tAAs}$ revealed that the contents of $^{13}C_{glycine}$ (1.7 ± 0.6%– 3.6 ± 1.0% of applied $^{13}C$; Supplementary Fig. 10a) were comparable to those of $D_{glycine}$ (1.9 ± 0.6%–2.7 ± 1.5% of applied D) on all sampling days ($p_{adjusted}$ > 0.05, Supplementary Note 4.1.4), and both were fairly stable between day 4 and day 38. Nearly identical amounts of $D_{glycine}$ and its $^{13}C$-analog suggest that the C–D bonds resisted harsh acid hydrolysis. The $D_{tAAs}$ hydrolyzed from the soil are thus reliable for tracking the D integration into C–D bonds of amino acids during microbial metabolism of a D-compound.

With $^{13}C_{glycine}$ comprising 16–30% of $^{13}C_{tAAs}$ and $D_{glycine}$ 76–81% of $D_{tAAs}$, glycine was clearly the most predominant AA in the tAA pool (Supplementary Fig. 10a). These findings are unique to 2-C$_{GLP}$ due to the preservation of the isotope label at the 2-C-position during biodegradation. When 2-C$_{GLP}$ is degraded to sarcosine, which can then be oxidized to glycine in the sarcosine pathway (Supplementary Fig. 8), both sarcosine and glycine will retain the D- or $^{13}C$-labels of the parent

$2\text{-}C_{GLP}$[36,64,67]. Thus, a disproportionally high integration of the isotope label from $2\text{-}C_{GLP}$ into glycine can be expected. $^{13}C_{glycine}$ and $D_{glycine}$ from $2\text{-}^{13}C_{GLP}$ and $2\text{-}C\text{-}D_2\text{-}GLP$ were likely assimilated into microbial biomass as a monomeric 'building block' for proteins, which is more energy-efficient than the biosynthesis of macromolecules derived from smaller C-precursors like acetyl-groups (Fig. 2)[36,59]. $^{13}C_{glycine}$ and $D_{glycine}$ could then have been partially mineralized to $^{13}CO_2$ and $D_2O$. The $^{13}C$ from $^{13}CO_2$ and the $^{13}C$ incorporated into other biomolecules of microbial degraders (1st level) was possibly recycled for the synthesis of new biomolecules by the consumers (2nd, 3rd, 4th level, etc.; Fig. 2). Due to these $^{13}C$-recycling processes, other $^{13}C_{amino\ acids}$ than $^{13}C_{glycine}$ were also enriched in $^{13}C$. Notably, the share of $D_{glycine}$ (76–81%) in the $D_{tAAs}$ was much higher than that of its $^{13}C$ analog (16–30%), suggesting that D was only minimally retained in other $D_{amino\ acids}$. Unlike $^{13}C$, the D-label is rapidly released as $D_2O$ after the cleavage of C–D bonds of either $2\text{-}C\text{-}D_2\text{-}GLP$ or D-biomolecules of the necromass (Fig. 2). The $D_2O$ is then diluted with unlabeled $H_2O$, leading to estimated $D_2O$ concentrations in soil water of only 0.00012–0.00076% for the sarcosine pathway (Supplementary Fig. 1 in Supplementary Note 2). Therefore, only low amounts of D-label could have been re-incorporated into the other $D_{amino\ acids}$. Overall, based on the shares of the two isotopes in the tAA pool and their recycling processes, we can deduce that GLP is preferentially degraded into the amino acid glycine in the sarcosine pathway, possibly dictating the high bioNER formation for $2\text{-}C_{GLP}$.

Still, a large proportion of the total $^{13}C_{NERs}$ from $2\text{-}^{13}C_{GLP}$ (70–84%; Fig. 5b) remained unidentified. We speculate that the $^{13}C_{bioNERs}$ for $2\text{-}^{13}C_{GLP}$ could have been underestimated due to the uncertainty of the bioNER approximation based on the tAAs*2 and MTB approaches. Notably, $2\text{-}^{13}C_{glycine}$ formed from $2\text{-}^{13}C_{GLP}$ in the sarcosine pathway (Supplementary Fig. 8) is directly incorporated into microbial biomass without the release of $^{13}CO_2$, as demonstrated by Wang et al.[36]. The predicted amounts of $^{13}C_{bioNER}$ for $2\text{-}^{13}C_{GLP}$ (6.7–16% for the sarcosine pathway) based on the measured $^{13}CO_2$ were thus likely underestimated, showing the limitations of the MTB approach when monomeric substrate utilization occurs (Fig. 2). Another good example proving the limitations of both the tAAs*2 and MTB approaches can be taken from a recent degradation study of $^{13}C_2\text{-}glycine$ by Aslam et al.[64], where 37% of $^{13}C_2\text{-}glycine$ was measured in $^{13}CO_2$, 8.7% in $^{13}C_{tAAs}$, and 34% in total $^{13}C_{NERs}$. Based on the $^{13}C_{tAAs}$*2, about 17.4% of total $^{13}C_{bioNERs}$ were formed, which comprised 51% of the total $^{13}C_{NERs}$; thus, the other 49% of $^{13}C_{NERs}$ were unidentified. The total $^{13}C_{bioNER}$ amounts estimated for $^{13}C_2\text{-}glycine$ using the MTB model were underestimated as well (min: 4% and max: 8.8%, Supplementary Table 6). Glycine is an easily biodegradable biomolecule[68]; therefore, it cannot form $^{13}C_{xenoNERs}$, and all of the $^{13}C_{NERs}$ from $^{13}C_2\text{-}glycine$ should be exclusively $^{13}C_{bioNERs}$.

$2\text{-}^{13}C_{GLP}$ or $2\text{-}C\text{-}D_2\text{-}GLP$ is degraded to unlabeled AMPA in the AMPA pathway (Supplementary Fig. 9). Therefore, the large portion of unidentified $^{13}C_{NERs}$ from $2\text{-}^{13}C_{GLP}$ could stem either from unidentified $^{13}C_{bioNERs}$ not considered in the MTB or tAAs*2 calculation, or from the un-degraded parent molecule which could be the primary source for $^{13}C_{xenoNERs}$.

**3-C-labeling position ($3\text{-}C_{GLP}$).** Contrastingly, neither $^{13}C_{glycine}$ nor $D_{glycine}$ was the dominant amino acid within the tAA pool of $3\text{-}C_{GLP}$ (Supplementary Fig. 10b). When $3\text{-}C_{GLP}$ is degraded via the sarcosine pathway, the major resulting degradation product glycine will be unlabeled (Supplementary Fig. 9), explaining the much lower amounts of $^{13}C_{bioNERs}$ and $D_{bioNERs}$ compared to $2\text{-}C_{GLP}$. Although the contents of $^{13}C_{glycine}$ were not significantly different from those of $D_{glycine}$ for $3\text{-}C_{GLP}$ on any sampling day ($p_{adjusted} > 0.05$, see Supplementary Note 4.1.4), their amounts were at least 15-fold lower than for $2\text{-}C_{GLP}$. The percentages of both $^{13}C_{glycine}$ (1.1–15% of $^{13}C_{tAAs}$) and $D_{glycine}$ (3.2–25% of $D_{tAAs}$) in the tAA pool were also comparatively lower than for $2\text{-}C_{GLP}$.

In contrast to $2\text{-}C_{GLP}$, $3\text{-}C_{GLP}$ is degraded to labeled AMPA ($^{13}C_{AMPA}$ or $D_2\text{-}AMPA$) in the AMPA pathway (Supplementary Fig. 9) as the third C

of GLP is preserved in AMPA. As $^{13}C_{bioNERs}$ and $D_{bioNERs}$ were lower than for $2\text{-}C_{GLP}$, higher portions of $^{13}C$- and D-labeled xenoNERs could have been formed from $3\text{-}C_{GLP}$. Since similar behavior of the parent compounds is expected, this could be due to abiotic interactions between labeled $^{13}C_{AMPA}$ or $D_2\text{-}AMPA$ with reactive groups of the soil.

## Discussion

As hypothesized, the C–D bonds of all tested compounds were stable against abiotic cleavage for the duration of the soil incubations. This was evidenced by comparable $D_{NERs}$ and $^{13}C_{NERs}$ in sterile soil (Fig. 3) and by the stability of the D-compounds in ultra-pure water solutions (Supplementary Note 6). Therefore, the differences between D and $^{13}C$ tracers in biologically active soil can be ascribed to biological processes.

Compared to $^{13}C$-labeling, the D-labeling approach yielded lower estimates of total NERs for all three model compounds 2,4-D, GLP, and SMX. For the two biodegradable herbicides 2,4-D and GLP, also the $D_{bioNERs}$ were lower than $^{13}C_{bioNERs}$ while no bioNERs were measured for the antibiotic SMX. As expected, the D tracer was thus far less retained in microbial biomass than the $^{13}C$ tracer. This can be attributed to three factors: (I) the presence of $H_2O$ as an abundant alternative H source for biosynthesis[38,49,50], (II) the release of $D_2O$ to water during microbial degradation followed by dilution of D with H from soil water, leading to low reuptake of D by microorganisms, and (III) isotopic discrimination against the heavy D[69,70]. Our findings thus suggest that D-labeled compounds generally form low amounts of bioNERs compared to $^{13}C$-labeled compounds. Consequently, for biodegradable compounds, the total $D_{NERs}$ will be lower than the total $^{13}C_{NERs}$ and will typically comprise a higher fraction of xenoNERs.

Recalcitrant compounds like SMX will form predominantly xenoNERs and thus similar amounts of total $D_{NERs}$ and total $^{13}C_{NERs}$. Biodegradable compounds may also form $D_{xenoNERs}$ when the isotope label is preserved in xenobiotic transformation products. $3\text{-}C_{GLP}$, for example, could be degraded into labeled AMPA. Although biodegradable compounds like 2,4-D or GLP can also form some $D_{bioNERs}$, the amounts of total $D_{NERs}$ relevant for the NER hazard assessment will remain lower than total $^{13}C_{NERs}$ and D-labeling will give a closer estimate of the xenoNER fraction. Notably, monomeric compounds can also be directly integrated into microbial biomass from a biodegraded substrate, as explained in Fig. 2 and shown for $2\text{-}C_{GLP}$[36,59]. This potentially results in higher amounts of $D_{bioNERs}$ than when only D/H incorporation from water occurs. However, bioNER contents from $2\text{-}C\text{-}D_2\text{-}GLP$ were still three-fold lower than from $2\text{-}^{13}C_{GLP}$. Even though the assimilation of a monomeric compound into the bioNERs had occurred, the results obtained by D-labeling would still be helpful for xenoNER quantification and the related hazard assessment.

We have proven that H is only minimally retained in biomolecules in chemical biodegradability testing using the D-isotope; thus, the total NERs comprising mostly xenoNERs could be readily quantified from the H-isotope remaining unextractable. The H-labeling approach could, therefore, solve the problem of the uncertainty associated with calculating total bioNERs based on tAAs*2 or the MTB approach. Although this proof-of-concept study demonstrated D-labeling as a powerful tool for time-efficient xenoNER quantification, it has several limitations. Compared to $^{13}C$ or $^{15}N$ tracers, D-labeling is undoubtedly cheaper in terms of associated costs of the labeled compound, and less laborious considering that the bioNER analytics can be entirely skipped (Table 1). Yet, the measurement of D – just like other stable isotopes – requires incubations with multiple controls and laborious isotope analytics. In addition, the required IRMS instruments may not be widely accessible. Furthermore, the experiments in this proof-of-concept study had to be performed at much higher than environmentally relevant concentrations of the test compound to achieve acceptable detection limits against the natural stable isotope abundances in soil (0.015 at% D and 1.08 at% $^{13}C$). The minimum required

**Table 1 | Comparative advantages and disadvantages of H vs. C tracers for xenoNER identification**

| Isotope | $^{13}C$ Stable | $^{14}C$ Radioactive | D Stable | T Radioactive |
|---|---|---|---|---|
| Applicability | All C-compounds | | Compounds with stable C–H bonds | |
| Availability of compounds | Depending on market trends | | Wide (internal standards) | Limited |
| Access to instrumentation | Limited (IRMS) | Wide (LSC) | Limited (IRMS) | Wide (LSC) |
| Laboratory workload | High (multiple controls, isotope analytics, NER identification) | Medium (NER identification) | Medium (multiple controls, isotope analytics, no bioNER analytics) | Low (once optimized, no bioNER analytics) |
| Initial substrate concentration | High (natural $^{13}C$ abundance) | Low (radioactivity) | High (natural D abundance) | Low (radioactivity) |
| Tracer quantitation | Time-consuming | Easy | Time-consuming | Potentially easy (T: weaker β-emitter than $^{14}C$) |
| BioNER analytics | Biomolecule extraction & purification | | Not relevant: minimal D or T retention in biomolecules | |

D: deuterium, T: tritium, IRMS: isotope ratio mass spectrometry, LSC: liquid scintillation counting.

spiking concentration for $D_{NER}$ quantification depends on both the number of labeled versus unlabeled H atoms in the test compound as well as the moisture and total H content of the soil. Multiple position D-labeling may thus enable lower spiking levels but is also constrained by the number of stable C–H bonds per test compound.

Therefore, to conduct biodegradation studies and NER identification at environmentally relevant concentrations, the radioactive H-isotope – T – could be applied as a promising substitute for D. Unlike stable isotope tracers, a radiolabeled compound can typically be mixed with its unlabeled analog to the desired spiking concentration and still give quantifiable signals. T-labeling has, however, not been favored for regulatory testing of chemicals due to concerns about the potential abiotic release of T from T-labeled compounds. Substrate-H bound to O, S, or N is generally considered exchangeable, whereas many C–H bonds are stable unless enzymatically cleaved by microorganisms[38,71]. However, the stability of C–H bonds may be compromised depending on their position within a molecule or ambient conditions. We, therefore, recommend carefully selecting appropriate C–H label positions that are stable against abiotic cleavage and, if needed, performing complementary stability tests in water or abiotic soil. Our proof-of-concept study demonstrated good stability of the C–D bonds for the three selected model compounds even during acidic hydrolysis conducted under 'harsh' conditions. Thus, C–H bonds should also resist the last 'harsh' step of soil extraction, which remobilizes 'slowly desorbable' residues[19]. As negligible amounts of D from all tested D-compounds were quantified in amino acids, the same is expected for T. T-labeling could thus substitute D-labeling for a rapid xenoNER quantification in future biodegradability tests. T-tracing may also be a more accessible approach than stable isotope probing due to the faster and easier quantification of radioactivity in (xeno)NERs using standard liquid scintillation counting (LSC)[15–17,27,29,31]. Nevertheless, prior to the potential application of T-labeling for future xenoNER quantification, the stability of C–T bonds under 'harsh' extractions still needs to be verified.

Besides stability considerations, H isotopes may also cause cellular toxicity and may be more prone to kinetic isotope effects than C isotopes[51]. Kinetic isotope effects were previously reported for enzyme-mediated metabolism and may be aggravated for H due to the large relative mass difference between D or T vs. $^1H$[72,73]. However, they may work in favor of the presented H-labeling approach as microorganisms were found to discriminate against D (and possibly T) uptake[51,69]. This may be due to a higher activation energy required for the synthesis of C–D compared to C–H bonds or the potential toxicity of very high $D_2O$ concentrations to microbial cells[51,70]. Abiotic processes driving xenoNER formation should, however, not be fundamentally altered by the higher mass of D or T vs. $^1H$, so that identical xenoNER quantities are expected for compounds labeled with $^{13}C/^{14}C$ or D/T at stable C–H bonds.

Further research is nevertheless still needed to verify the applicability of H isotope tracing across soils with diverse chemical and biological properties. NER formation was shown to vary substantially between soil types[29,59], e.g., as a function of pH, organic carbon, and mineral content affecting the sorption of organic molecules. While the mechanism behind lower bioNER formation from H tracers is based on enzyme-mediated processes, making it in principle applicable independently of chemical soil composition, extreme pH ranges or a high abundance of transition metals could enhance the catalysis of abiotic H exchange in C–H bonds[74]. Therefore, it may be of interest to compare abiotic NER formation between C and H tracers across a broad range of different soils. Moreover, the abundance and physiology of different types of degraders within the soil microbial community may largely affect the assimilation of H vs. C tracers into bioNERs. For example, $CO_2$-fixating autotrophs took up more water-H than heterotrophs but also showed stronger H fractionation[51]. Differences in substrate-H utilization were also observed between different heterotrophic degraders and for favorable vs. stressful growth conditions. In heterotrophic bacteria, water-H uptake occurred mainly during anabolism, while substrate-H was mainly released to ambient water during catabolism[49]. As a complex interplay between chemical and microbiological factors affects H isotope incorporation into bioNERs, further studies are required to gain a better quantitative understanding of how much the difference between C and H isotope-labeled bioNER formation may vary.

This proof-of-concept study has shown that D- or T-labeling could become a powerful substitute or supplementary method for rapid xenoNER quantification along with $^{14}C$-labeling. By helping to efficiently identify potentially hazardous long-lasting organic contaminants, the presented H-isotope labeling approach could contribute to the advancement of green chemistry within chemical safety testing. As long as H-labels are stably attached, D- or T-labeling could be broadly applied for scientific and regulatory biodegradability testing of a wide range of organic chemicals to reveal the hidden identity of NERs.

## Methods
### Chemicals
All chemicals and reagents were of analytical grade and purchased from VWR (Darmstadt, Germany) or Carl Roth (Karlsruhe, Germany) unless otherwise stated. 2-$^{13}C$ and 3-$^{13}C_{GLP}$, and $^{13}C_{6-SMX}$ were obtained from Merck (Darmstadt, Germany), while $^{13}C_{6-2,4-D}$ was purchased from Toronto Research Chemicals (Toronto, Canada). All $^{13}C$-labeled compounds had an isotopic purity of 99 at% $^{13}C$ and chemical purity > 98%. 2-C–$D_{2-GLP}$ (99 at% D), $D_{3-2,4-D}$ (99 at% D), and $D_{4-SMX}$ (97 at% D), each with a chemical purity > 99%, were purchased from Toronto Research Chemicals (Toronto, Canada). 3-C–$D_{2-GLP}$ (98 at% D) was obtained from Cambridge Isotope Laboratories Inc. (Andover, USA). Unlabeled 2,4-D,

GLP, and SMX of analytical grade as well as D-depleted water (at% $^2$H ≤ 1 ppm) were obtained from Sigma Aldrich (Munich, Germany).

## Reference soil material

The Ap horizon of a Haplic Chernozem was sampled from the long-term agricultural 'Static Fertilization Experiment' located at Bad Lauchstädt, Germany. The soil had received various pesticides (including glyphosate and 2,4-D) for over 30 years and had been amended with 30 t manure ha$^{-1}$ every second year for over 100 years[75]. The silt loam was composed of 21% clay, 68% silt, and 11% sand and had a total organic carbon content of 2.1% and a total nitrogen content of 0.17%. The soil pH was 6.6, and the maximum water-holding capacity (WHC$_{max}$) was 37.5%[59]. The soil was sieved to 2 mm, homogenized, and stored at 4 °C at 7% WHC$_{max}$ for 2 months before the experiments.

## Model compounds

Two herbicides, GLP and 2,4-D, and one antibiotic, SMX, were selected to compare the NER and bioNER formation between D and $^{13}$C tracers. The $^{13}$C-compounds were labeled with $^{13}$C at similar positions as the D-compound (see Fig. 6). The C–D bonds of all deuterated model compounds were also proven to be stable against abiotic cleavage in water (for details see Supplementary Note 6).

## Incubation experiments

Soil incubations oriented at OECD guideline 307[16] were performed in a static system consisting of 250 mL Schott flasks filled with 60 g (wet weight) of the reference soil material. Besides treatment with the D- and $^{13}$C-labeled compounds, two different controls were included: untreated soil and soil spiked with unlabeled compound. Both controls were performed to obtain background isotopic abundances for the NER and bioNER calculations. For each test compound, analogous sterile experiments were conducted to verify whether D- and $^{13}$C-labeled compounds behave identically under abiotic conditions. All treatments were conducted in triplicate, i.e., three separate flasks were incubated per treatment. Prior to begin of the experiments, the soil was oven-dried at 40 °C over multiple days until reaching a constant weight. After thorough manual mixing, the soil for sterile controls was separated into 250 mL Schott flasks and autoclaved three times (121 °C, 40 min) on consecutive days, with the last autoclaving cycle on day 0

of the experiments. The remaining soil was stored in an airtight 2 L bottle, and the moisture content was monitored gravimetrically. To minimize a priming effect on microbial degraders, approximately four hours before starting the incubations, the soil moisture was adjusted to 20% WHC$_{max}$, and the soil was again mixed thoroughly by manual stirring. The soil for biotic treatments was then weighed into 250 mL Schott flasks, and spiking was performed separately for each bottle. To this end, aqueous solutions of the test compounds (prepared on the same day as incubation) were added dropwise to the soil, corresponding to final concentrations of 50 mg kg$^{-1}$ dry soil for GLP, 20 mg kg$^{-1}$ dry soil for SMX and 10 mg kg$^{-1}$ dry soil for 2,4-D. These much higher than environmental concentrations were selected after prior testing to yield good resolution on the IRMS instruments. Estimated detection limits for $^{13}$C$_{NERs}$ and D$_{NERs}$ of the three model compounds were derived as described in Supplementary Method 1. After spiking, the soil moisture was adjusted to 60% WHC$_{max}$ to provide optimal growth conditions for microorganisms[7]. Each treatment was then homogenized by manually stirring for two minutes. In the abiotic treatments, the soil moisture was adjusted separately for each bottle immediately after spiking the test compounds to the dry soil, as no priming effect was expected. Spiking and sampling for the abiotic treatments were conducted under sterile bench conditions. In biotic treatments with the D-compounds and unlabeled controls, D-depleted water was used for the spiking solutions and moisture adjustment in order to lower the background D in EA-IRMS measurements of water-extractable D on day 0 (Supplementary Table 2a in Supplementary Note 3).

The soil incubations were conducted in the dark at 20 °C to prevent photodegradation of the model chemicals. Soil samples were taken from the same bottles on days 4, 16/18, and 36/38 for GLP and 2,4-D and on days 18, 36, and 72 for SMX because of its slower turnover. Each time, ten roughly 0.5–2 g subsamples were taken carefully from different spots within the soil batch to prevent soil disturbance. The sampling did not cause any noticeable disturbances in soil microbial activity as soil respiration showed a continuous decrease of CO$_2$ towards the end of the incubation periods (Supplementary Fig. 6 in Supplementary Note 7). The soil was then pooled into one composite sample in a 50 mL Falcon tube (day 0: 6 g total, afterward: 18 g total), homogenized by stirring with a spatula and stored at −20 °C

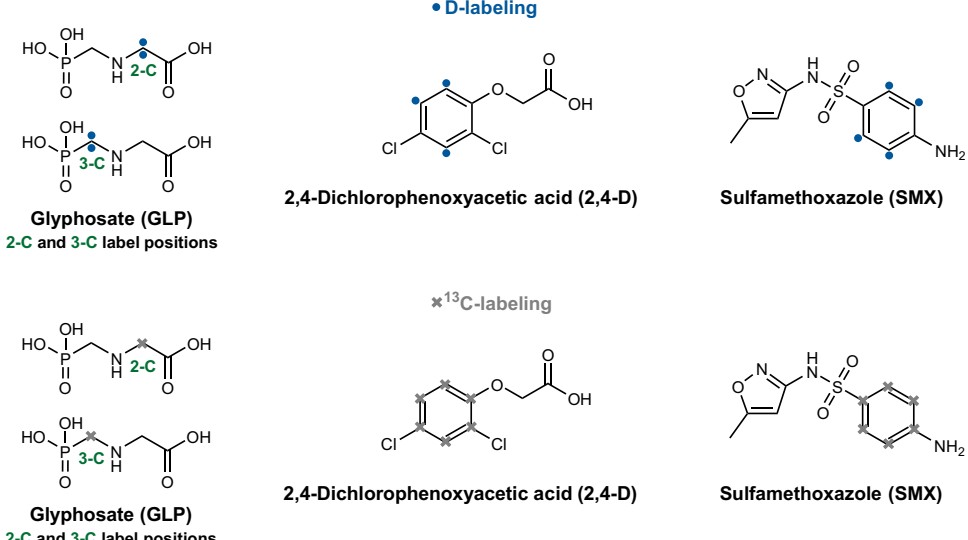

**Fig. 6 | Molecular formulas of the three test compounds: 2,4-D, GLP and SMX, indicating the positioning of D-labels and $^{13}$C-labels.** Gray: $^{13}$C-label, blue: D-label. The top panel shows the positioning of the D-label, from left to right: D$_{2\text{-GLP}}$ (2-C and 3-C labeling), D$_{3\text{-2,4-D}}$ and D$_{4\text{-SMX}}$. The bottom panel indicates the positioning of the $^{13}$C-label, from left to right: $^{13}$C$_{GLP}$ (2-C and 3-C labeling), $^{13}$C$_{6\text{-2,4-D}}$ and $^{13}$C$_{6\text{-SMX}}$.

until analysis. During spiking and sampling, treatments were handled in the same order to account for the required processing time.

## Total $^{13}$C- and D-labeled NERs

Total NERs were quantified based on the abundance of D or $^{13}$C remaining in the soil after the 'soft' extraction of model compounds and their transformation products by shaking with water-methanol (2,4-D), water-borate buffer (GLP) and water-dichloromethane (SMX) as detailed in Supplementary Note 3. Soil samples were air-dried over multiple days after the extraction and ground for homogenization. A 2–4 mg aliquot was then weighed into 3.5 × 5 mm tin cartridges (HEKAtech) for analysis by EA-high-temperature conversion-IRMS (D analysis) or EA-combustion-IRMS ($^{13}$C analysis). The total H and D content and isotopic enrichment of D (at% D/$^1$H) of the D$_{NERs}$ were measured on an EA (EuroEA3000, Euro Vector, Milan, Italy) directly connected via an open split system (ConFlo IV, Thermo Fisher Scientific, Germany) to IRMS (MAT 253 Thermo Fisher Scientific, Germany). The total amount of C and the $^{13}$C/$^{12}$C isotope ratio of the $^{13}$C$_{NERs}$ were determined using a Flash EA 2000 coupled to a Conflo IV interface and a Delta Advantage mass spectrometer (Finnigan MAT 253, Thermo Scientific, Bremen, Germany). The temperature of the oxidation reactor was 1020 °C, whereas that of the reduction reactor was 650 °C[57]. The isotopic enrichment of D or $^{13}$C in NERs was calculated as excess over the unlabeled control. The detection and quantification limits were estimated at around 3–5% of applied D and 4–5% of applied $^{13}$C (Supplementary Table 8a & 8b in Supplementary Method 1).

## Total $^{13}$C- and D-labeled amino acids (tAAs) and total $^{13}$C- and D-bioNERs

Total amino acids (tAAs) were extracted from the soil as quantitative and qualitative markers for bioNERs. The tAAs from living and decayed biomass are the most reliable quantitative biomarkers for bioNERs as their turnover is comparably slow[7,35]. The tAA analysis followed the extraction, purification, and derivatization protocol by Nowak et al.[7,35]. Briefly, for tAA extraction, a 2 g soil sample was hydrolyzed with 6 M HCl at 105 °C for 22 h and purified over cation resin (Dowex 50W-X8; 50–100 mesh) solid-phase extraction columns. Impurities were removed by consecutive washing with 2.5 M oxalic acid, 0.01 M HCl, and de-ionized water before eluting the AAs with 2.5 M NH$_4$OH. The purified extracts were derivatized by iso-propylation of carboxyl groups and trifluoro-acetylation of amino groups before measurement with gas chromatography-mass spectrometry (GC-MS, HP 6890, Agilent) operating with a BPX-5 column (30 m × 0.25 mm × 0.25 μm; SGE International, Darmstadt, Germany). Individual AAs were identified from an external AA standard containing alanine, glycine, threonine, serine, valine, leucine, isoleucine, proline, aspartate, glutamate, phenylalanine, tyrosine and lysine. Quantification was based on two internal standards, L-norleucine added after the hydrolysis and 4-aminomethylcyclohexanecarboxylic acid added before derivatization. The recovery of AAs from soil hydrolysates using this purification method was 97 ± 11%[76]. Isotopic enrichment in tAAs was measured with gas chromatography-isotope ratio-mass spectrometry (GC-IRMS), using a trace 1310 GC system connected via a GC-IsoLink and a ConFlo IV interface to a MAT 253 (all Thermo Fisher Scientific, Bremen, Germany). Sample separation was achieved with a BPX-5 column (30 m × 0.32 mm × 0.5 μm; Agilent Technology) using helium as a carrier gas. Details on the chromatographic analyses with GC-MS and GC-IRMS are provided in Supplementary Method 2. The total abundance of different AAs was measured by GC-MS, and the isotopic enrichment (at% D/$^1$H or at% $^{13}$C/$^{12}$C) in the respective molecule was determined by GC-IRMS after correcting for the isotopic shift during derivatization (Supplementary Method 3). Isotope label integration from the labeled compounds was calculated by subtracting the isotopic enrichment in unlabeled control samples. Because the conditions of hydrolysis with 6 M HCl were harsh, the D integrated into the C–D bonds of tAAs must

have been non-exchangeable, in contrast to D bound to O, N, or S, which is generally considered to be exchangeable[38]. This was proven by the comparable amounts of $^{13}$C$_{glycine}$ and D$_{glycine}$ for the 2-C-labeling position of GLP, as shown in Supplementary Fig. 10. The estimated LOD for individual AAs ranged between 0.07–2.4% of applied $^{13}$C and 0.01–3.1% of applied D for 2,4-D and GLP (Supplementary Table 8c, 8d in Supplementary Method 1).

## Data analysis

D or $^{13}$C enrichment in NERs and tAAs was calculated as the percentage of D or $^{13}$C initially applied with the labeled compounds, and values are presented as mean ± standard deviation. The detailed calculation of $^{13}$C- and D-label incorporation into total NERs and tAAs is explained in Supplementary Method 4. The mean isotopic enrichment per treatment group (mean$_{enrichment\ tNERs/AA}$) was calculated as the product of the mean total element or AA abundance in soil (mean$_{\mu mol\ tNERs/AA}$) and the difference between the mean the isotopic enrichment in the labeled treatment (mean$_{at\%\ labeled}$) and unlabeled control:

$$\text{mean}_{\text{enrichment tNERs/AA}} = \text{mean}_{\mu\text{mol tNERs/AA}}$$
$$\cdot \left( \text{mean}_{\text{at\%labeled}} - \text{mean}_{\text{at\%unlabeled}} \right) \quad (1)$$
$$= \text{mean}_{\mu\text{mol tNERs/AA}} \cdot \text{mean}_{\text{at\%enrichment}}$$

Therefore, the uncertainty of the mean D or $^{13}$C enrichment in NERs or individual AAs (SD$_{at\%\ enrichment}$) was derived considering Gaussian error propagation as follows:

$$\text{SD}_{\text{at\%enrichment}} = \sqrt{\text{SD}_{\text{at\%labeled}}{}^2 + \text{SD}_{\text{at\%unlabeled}}{}^2} \quad (2)$$

Where SD$_{at\%\ enrichment}$, SD$_{at\%\ labeled}$, and SD$_{at\%\ unlabeled}$ are the standard deviations of the mean isotopic (D or $^{13}$C) enrichment, mean isotopic abundance in the labeled treatment, and mean isotopic abundance in the unlabeled treatment, respectively. The total uncertainty in the mean tNERs or labeled AA contents (SD$_{tNERs/AA}$) was calculated as:

$$\text{SD}_{\text{tNERs/AA}} = \text{mean}_{\mu\text{mol tNERs/AA}}$$
$$\cdot \sqrt{\left( \frac{\text{SD}_{\text{at\%enrichment}}}{\text{mean}_{\text{at\%enrichment}}} \right)^2 + \left( \frac{\text{SD}_{\mu\text{mol C, H or AA}}}{\text{mean}_{\mu\text{mol C, H or AA}}} \right)^2} \quad (3)$$

Here, mean$_{at\%\ enrichment}$ is the mean isotopic enrichment, and mean$_{\mu mol\ C, H\ or\ AA}$ and SD$_{\mu mol\ C, H\ or\ AA}$ are, respectively, the mean and standard deviation of the measured total abundance of C, H or the individual AA in the model soil. The total C and H abundance were measured over all sampling days as they were nearly constant while individual AA abundances were calculated separately for each sampling point. The uncertainty in the tAA abundance was calculated analogous to Eq. (2) by taking the square root of the sum of all squared standard deviations for individual AAs that were calculated according to Eq. (3).

Unidentified NERs were calculated from the difference between the total NERs and total bioNERs (tAAs*2). The standard deviations for total bioNERs presented in the text were estimated by applying the conversion factor of 2 to the standard deviation of the tAA measurement; however, it may actually be larger due to the additional uncertainty of the conversion factor.

## Statistics

For statistical analysis, individual replicate values per treatment group were calculated from the mean isotopic abundance in unlabeled controls and the mean total H, C, or AA contents in the soil as mean$_{abundance\ C/H/AA}$*(at%$_{replicate}$ − mean$_{at\%\ unlabeled}$). Statistical tests, hence, only consider the uncertainty of the isotopic abundance in the labeled treatment, whereas error bars in Figs. 3–5 show the propagated uncertainty, including the error of the unlabeled control isotopic

abundance and the total H, C, or AA abundance. Each treatment group contained three independent measurements of the isotopic enrichment in the labeled treatment from separately processed samples, except for tAAs and glycine of 3-C–D$_2$-GLP on day 38, where only two replicates were analyzed due to sample losses.

Measured contents of total NERs and tAAs from 2,4-D and SMX were assessed for significant differences between D and $^{13}$C tracers at each time point using independent samples and two-tailed $t$ tests. Welch's degrees of freedom correction for heteroscedasticity was applied as variance ratios were > 3 in all cases. Although assessment of normality is challenging for small samples (here $n = 3$), Welch-tests were shown to be fairly robust against deviations from normality even for very small group sizes[77]. Since for GLP, four different treatments were used (D vs. $^{13}$C tracer with two different labeling positions each), differences in total NERs, tAA, and glycine contents at each time point were assessed by Kruskal-Wallis tests. In case of significant main effects, Conover-Iman tests were employed for post-hoc comparison. Because of unequal variances, normality was assessed based on quantile-quantile plots of the groupwise standardized residuals (Supplementary Figs. 2–5 and Supplementary Note 4). Due to the appearance of light-tailed distributions with data gaps, unequal variances, and very small, on day 38 additionally, unbalanced group sizes, nonparametric tests were used. $P$-values obtained from Welch tests and Conover-Iman tests were corrected for multiple comparisons using the Holm-Bonferroni method of controlling the family-wise error rate. All analyses were carried out with R Statistical Software (v 4.3.3)[78] using a significance level of $\alpha = 0.05$. Further details on the employed R packages and software output, including t and $X^2$ statistics, degrees of freedom, and exact $p$-values, are provided in Supplementary Note 4.

## Data availability

The authors declare that the data supporting the findings of this study (Figs. 3, 4, 5, and Supplementary Fig. 10) are available within the paper and its Supplementary Information. Source Data with raw data for Figs. 3–5 and Supplementary Fig. 10 are available under the Figshare link: https://doi.org/10.6084/m9.figshare.25954396. The other data that support the findings of this study are available from the corresponding author K.M.N. upon request.

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

## Acknowledgements

This research was funded by the Cefic-Long-Range Research Initiative (LRI), German Research Council (DFG, No 980/3-1) and Helmholtz Center for Environmental Research-UFZ. The authors are thankful for the use of the analytical facilities of the Laboratories for Stable Isotopes (LSI) at UFZ Leipzig. Moreover, we would like to acknowledge Ines Merbach (UFZ, Experimental Station Bad Lauchstädt) for providing the reference soil material for the incubation experiments and Roman Gunold (UFZ, Dept. Exposure Science) for helping with the LC-MS analysis of GLP, 2,4-D, and SMX.

## Author contributions

K.M.N. conceived, designed, and supervised the project. S.L. and H.A.B. performed the experiments. S.L. and K.M.N. analyzed the data. S.K. helped with the D/H and $^{13}$C/$^{12}$C analysis M.K. with the D/H exchange tests, and both S.K. and M.K. with the data interpretation. S.L. and K.M.N. wrote the manuscript. S.L., H.A.B., S.K., M.K., and K.M.N. revised the manuscript. All authors commented on the manuscript.

## Funding

## Competing interests

The authors declare no competing interests.
