## [Peer Review File · Nature Communications]

Hydrogen isotope labeling unravels origin of soil-bound organic contaminant residues in biodegradability testing

Corresponding Author: Dr Karolina Nowak

Version 0:

Reviewer comments:

Reviewer #1

(Remarks to the Author)

General:

The manuscript presents a method to distinguish experimentally bioNER from xenoNER based on the hypothesis, that Hydrogen from an organic molecule will mostly get lost in the biological Carbon cycle. The hypothesis is scientifically sound and could provide a new way to identify bioNER.

However, the work contains some mistakes/gaps that can hopefully be corrected/explained, as they can question the result of the good overall work.

In general, NER are differentiated in the entire manuscript (see point by point comments) in “potential hazardous” xenoNER and “safe” bioNER. This differentiation is not correct as the xenoNER are further differentiated into NER type I and type II [Kästner, M., Trapp, S. & Schäffer, A. Consultancy services to support ECHA in improving the 497 interpretation of Non-Extractable Residues (NER) in degradation assessment. Discussion paper-final 498 report (2018)], and type II NER are assessed “safe sink” in current regulation. It is important to correct this in the manuscript because the current manuscript could give the impression that the differentiation has been done this way to increase the relevance of the work on the bioNER. Which is completely unnecessary given the remarkable idea and quality of the work.

Technically, there are some more questions:

- At no point it was investigated, if ¹³C-NER and D-NER would provide similar results if it would be only xenoNER. This could have been easily clarified by performing additional sterile experiments. Without that information it is very difficult to assess if the differences reported are really due to bioNER. Maybe is a systematic gap – the result of SMX, where no bioNER could be found but a difference of up to 30% was observed between ¹³C-NER and D-NER give some indication for that. In the manuscript this gap is called “insignificant”. But it raises the question for the other two substances where the absolute difference between ¹³C-NER and D-NER is not higher than in SMX experiments.
- The term NER implies, that everything has been tried to extract mobile fractions of the test substance. In fact, apparently only a simple single extraction was carried out, the effectiveness of which has at least not been reported (e.g. recovery on day 0?). Without this information, serious concerns arise as to whether the extraction residue can actually be described as NER.
- The experimental approach is a bit different from an OECD 307 test, but it also says “oriented at”, which is fine. However, it remains a bit unclear, how the setup was, how sampling and homogeneity of the samples were done. All in all, this let some doubts on the reliability of the results. Which I am sure is not necessary, so please try to make that a bit clearer.

I very much like the idea presented, but there might be still a longer way before regulatory use as long as it is not clear how high the normal difference between C-NER and H-NER (whichever isotope) is in the xenoNER. Please find the point by point comments below.

Point by point comments.

[21] xenoNER are not considered a potential hazard by definition. This is currently only true for type I NER. Type II NER are assessed as safe sink in regulation like the bioNER (Type III NER).

[28] disagree! See comment to [21]. xenoNER are not generally assessed a potential hazard and bioNER are not the only NER-type assessed a safe sink. The current text gives too much weight to the presented method – type II NER are assessed

safe in current regulation.

[29] Why not Tritium-label? In the discussion T is also recommended. Would be nice to mention T instead of “innovative H-isotope labelling”.

[38] I would not agree to “poorly understood”. The sentence reads as we would not care contaminants in soils and just put it there to get rid of them. In particular for pesticides (2 of the test substances are pesticides), fate processes in soils are very good investigated, identified and reported. NER are not good understood, but degradation processes are well known.

[42] ultimate biodegradation is not the only detoxification step and degradation of organic chemicals in regulation is mostly understood as primary degradation which can also happen abiotically. Transformation products might be still hazardous, but must not be. Mineralisation is no trigger in degradation assessment (except screening test and polymers).

[54] There are rather two types of “xenoNER” that carry different hazards. This is also what is described in more detail in the following chapter. See comment to [28].

The differentiation “xenoNER = hazard” and “bioNER = safe” is just wrong. Type two xenoNER are assessed safe as well. Fig 1 The description doesn't match the figure. In the figure xenoNER are labelled “hidden hazard”, which is not true (see comments above).

The “14C-NERs” or “total NER” will always contain xenoNER and bioNER since total NER don't differ. So is not “can contain” but “contain”. There is no technique to determine a total NER fraction that does not contain bioNER.

[87] Again the same as before. xenoNER are no hazard by definition. Please correct.

[89] I don't see the difference between 14C and 13C. Even if it is not part of the routine (though xenoNER differentiation is already allowed in regulation and might become part of the routine very soon) all analysis described below can be done with 14C as well. Both isotopes can be analysed by GC-MS, by GC-IRMS and also by HR-LC-MS. Everything that can be done with the 13C-isotope label can be done the same with the 14C-isotope label. With the advantage, that the 14C additionally give the option for a very easy radioactive analysis. Combination of 14C-labels with 13C and 15N labels are also pretty common in pesticide regulation to enable transformation product identification by NMR-techniques.

[94] Could you explain, why that should be cheaper? Analysis of Tritium (radioactive H – Isotope) is very simple and doesn't need expensive sample pre-treatment and HR-MS techniques. Though the D-labelled substance itself is surely cheaper than T-labelled substances, that I would agree.

[104] “C- or N isotope labelled substrate”. Is not limited to 13C.

[114ff] Agree! That's the reason, why T-labelling is not so commonly used in those studies. Because it might be exchanged and then the label is not very specific for the original test substance any more.

[123] Good idea. I would support that hypothesis.

[130] Isn't the control a “12C-label” = non labelled (see line 356)?

[141] If the amino acids are quantitative marker for bioNER, why is the D-labelling needed? In that case the 13C amino acids should serve the same. Of course for proof of concept is fine.

[159] How reliable are D-NER if there are even negative values $2.6 \pm 3.9\%$? What would be the limit of detection for that measurement?

[176ff] I would not say that 20% to 30% difference between 13C-NER and D-NER is not significant. The absolute difference in the 2,4-D experiment was even less than that. Though I see the point. However, this shows a weak point in the all over work: the missing sterile samples and the insecurity how much would be the difference of C-NER and D-NER in the xenoNER without bioNER. According to [267ff], no bioNER were formed at SMX.

[181] You compare to 14C-NER in literature only for SMX. Very nice. But why not for GLP and 2,4-D? Both are pesticides and well investigated as 14C-label. Data should be available in literature. Would be nice to have those data for comparison.

For all three substances: why didn't you perform a sterile control as recommended by OECD 307? In this case maybe NER would have been much lower (except SMX probably) but would have been a nice proof of your hypothesis since 13C-NER and D-NER should be almost identical if the hypothesis is right.

[268] please delete the “hidden hazard” as this is not proven. Is rather the opposite when the NER are mostly abiotic NER (a sterile control would have given important additional information), it can be speculated that this is due to covalent bonding to the soil matrix. Covalently bound xenoNER are mostly type II NER and as such considered a “safe sink”.

[274] Just 7 lines before its said that no bioNER are found in the SMX experiment. Does not match.

[275] How do you explain the 20-30% difference of NER at the SMX then?

[286] I don't find it very protective to overestimate the bioNER at SMX by about 30%! For this hypothesis it needs to be figured, how much the difference of C-NER and D-NER really is. This work gives clear indication, that D-NER are generally lower than C-NER.

[291] please delete “potentially hazardous”. This is a mistake through the entire manuscript.

[304-309] right. Both, D and T labelling might be possible. That's why I don't see the difference you state in line 89. Please check line 89 again and modify.

[313] LSC instruments (e.g. Perkin Elmer) are capable to analyse T since many years. T-labelling is used in pesticide regulation, although not as widely used since a H-shift is always a worry to give unreliable results. But in general T-labelling for testing is well established already.

[333] Most likely typical for an actual agricultural soil – but for substance testing OECD 307 asks a soil that did not see the test chemicals in the last 4 years (OECD 307, No. 26).

[359] That is unfortunately a big disadvantage of the stable isotope labelling. This could be an issue for test substances that are unknown toxic to soil microorganisms.

[367] Do I understand that right? The individual samples were 60g wet weight soil (2 mm sieved) and no sacrificial sampling was done but subsampling from these little bottles? How did you homogenise the sample after application? This is very difficult in those test bottles in particular with the silty soil used. Thus, the normal procedure would be sacrificial sampling. What was the size of the subsample taken at different timepoints? Please explain.

[372] How was the recovery of the substances direct after application with those extractions? Any data available? This is very important to assess if you really analysed “NER” or rather a fraction that is not extractable with the applied simple

extraction step. The entire manuscript is about NER but this short description raises doubts as to whether NERs have been studied at all or rather another fraction.

Reviewer #2

(Remarks to the Author)

The manuscript titled 'Hydrogen Isotope Labeling Unravels the Origin of Soil-Bound Organic Contaminant Residues in Biodegradability Testing,' authored by Lennartz et al., addresses a timely concern, specifically the evaluation of various types of non-extractable residues (NERs) of micropollutants in soil. The scientific significance of this study lies in demonstrating that a deuterium-labeling approach may be both straightforward and adequate for protective NER hazard assessment, as these isotopes exhibit lower retention in the non-hazardous microbial biomass compared to the ^{13}C tracer. The implication of the study is noteworthy as it reveals that deuterated or tritium isotopes can be utilised to quantify the remaining non-extractable fraction of micropollutant residues in soil, serving as a valuable proxy for xenobiotic NERs.

While the manuscript is well-conceived and written, some drawbacks merit attention and clarification.

Conceptually, I am not convinced that Figure 2 allows for an easy grasp of the fundamental differences in using $^{13}\text{C}/^{15}\text{N}$ or D-labeled substrates. The potential advantages of using D labels are not sufficiently clear, as processes such as biosynthesis (utilization of H_2O as an H source), dilution of D_2O to water along with limited microbial uptake, and isotopic fractionation for D/H are not highlighted at the mechanistic levels and in a quantitative manner. Consequently, strong arguments in favor of D labels may be overlooked. I suggest a thorough revision of the conceptualization of using D-labeled substrates to reflect the current state of the art and to provide more consistent quantitative arguments supporting the authors' idea. Concepts such as comparative analytical drawbacks, costs, sensitivity, labeling positions, etc., may be emphasized.

Regarding the experimental design, I find the description of the incubation experiment to be insufficient, and I have reservations about the generic nature of the experiments and related data. From the experimental set-up, it is not clear how many replicate experiments were eventually used and whether sacrificial or non-sacrificial experiments were conducted. This is crucial for evaluating the reproducibility and robustness of the findings. It is mentioned (L. 366-368) that soil samples were collected non-destructively from different spots within the soil batch on different days. I am curious about how this is possible without a sacrificial approach. If a non-sacrificial approach was used, this may have severe consequences on the experiments since experimental conditions may be disturbed after each sampling, potentially affecting NER distribution. Moreover, standard biological experiments typically require at least 5 replicate, independent experiments, which seems not to be the case in this study. This requires clarification and a thorough discussion of the consequences of the experimental set-up on the results.

Another aspect is the use of high and environmentally non-relevant concentrations of the model pollutants. While high concentrations effectively allow testing the hypothesis posited by the authors, the findings may not reflect NER extent and distribution that may occur in the field. This is particularly true for sulfamethoxazole, which may occur at concentrations that are 2 to 3 orders of magnitude lower than those used in this study. This should be discussed and acknowledged. Additionally, it may be relevant to clearly state the limits of the proposed methodology at environmentally-relevant concentrations in terms of sensitivity and uncertainty of bioNER and xenoNER analysis and quantitative estimation. This can be compared with the traditional approach using $^{13}\text{C}/^{15}\text{N}$ substrates.

It would be also of interest to discuss how D-labeled substrate may behave in different type of soil and under different soil conditions.

Detailed comments:

L. 25: Use "proportion" instead of "percentage."

L. 96: Clarify the different turnover dynamics in the text and scheme.

L. 112-113: Provide a quantitative statement. How rapidly is D_2O lost? How challenging is the uptake?

L. 121: How quickly is the D in D-labeled substrate expected to be lost?

L. 140-141: Offer more background and justification for the use of amino acids as a quantitative biomarker for bioNERs.

L. 173-174: Briefly explain why and how.

L. 187-188: Clearly and scientifically justify the multiplication by 2 of labeled amino acids to estimate total bioNERs.

L. 191 and following: Clearly explain how the uncertainty associated with the proportion of NERs was derived.

L. 217-219: Explain the meaning of this statement. Clarify why it is surprising.

Fig. 5: Clarify in the text why this figure is important. Provide justification for the interest in this figure and related findings.

L. 286: Define what a 'protective NER hazard assessment' means.

Last paragraph: Clearly explain how D-labeled substrate may be used in a more generic way for different substrates, pollutants, soil types, etc.

L. 335: Provide a reference for the studied soil.

Version 1:

Reviewer comments:

Reviewer #1

(Remarks to the Author)

Review Nature communications 458241_1

Second review, line numbers from file "458241_1_merged_1717358587.pdf"

The current review has been done only on the revised manuscript without considering remarks/comments on the first revision. So might be that comments of the first and second revision repeat.

Still the project idea is outstanding but would have deserved a more careful planning of the experimental part considering principles from well established guidelines (e.g. OECD 307). Sterile experiments close a significant gap but results are only considered when it fits the hypothesis.

Point by point:

[64] using silylation or extraction with EDTA

Fig 1: type II NER are considered safe sink, not "potential" safe sink. Type I NER are "potential" hidden hazard as you also describe in the figure description. Please correct, also in the figure.

In contrast, the hazard of soil-bound contaminant residues is not clear as they can be 'safe sink' bioNERs type III or 'hidden hazard' xenoNERs type I. this is simply wrong and the added sentence does not correct it. Soil bound is covalently bound and as such it is not the original chemical anymore without any of the original properties. Bound residues are considered safe sink, not "potential safe sink"

[97] D-labelling might be cheaper, but the analysis in biodegradation studies are as laborious as ¹⁵N or ¹³C label analysis.

[160] currently there is no expensive and laborious bioNER assessment, but is just done by MTB model which is accepted in regulation for bioNER assessment.

[164] what is a standard C isotope tracers ?

[185] no! The exhaustive extraction (that btw should end with an ASE extraction according to ECHA) aims to to extract the extractable portion of a substance. If it would extract NER, which means NON EXTRACTABLE residue, this would not be NON EXTRACTABLE by definition. It does not aim to extract NER Type I. Please correct.

[187-189] That is simply wrong, please delete. After exhaustive extraction you get a material containing total NER. Total NER consist by definition of NER type I, II and III.

[200] what do NER have to do with sorption coefficients?? Sorption coefficients are valid for solid/water systems at equilibrium – but this is not the aim of an extraction. For extraction with solvents this does not apply or the solvent is not suitable. Please delete

Fig 3: nice figure! Was it triplicate measurement of the same sample or single measurement of triplicate samples? I assume it was sacrificial sampling? Maybe is described later..

[220 ff] 2,4-D: any idea, why ¹³C-NER are not higher than in the sterile experiment? That's unusual.

[235 ff] You got again not more NER than in the sterile experiment. But you have more than 60% ¹³C NER after 4 days?? And less than half of that is D-NER. So more than 40% bioNER after 4 days?? That would be a very rapid degradation kinetic. Did you really measure such a rapid mineralisation that would go along with this (according to the MTB model)? Does not sound reasonable. In normal world (¹⁴C), NER increase slowly over time, except there is very fast degradation or mainly abiotic processes. As shown in the sterile samples. But this does not match with the conclusion of having half of the NER as bioNER after such short time. Sure, that the extraction was efficient? Also for known transformation products?

[246] Adsorption of AMPA to the soil? Then the extraction was not efficient. Adsorbed (see comment for line 200) substance should be released during extraction.

[258] yes right! But this applies also to the other two substances: NER in biotic are very similar to the abiotic experiments for ALL substances. This is somehow suspicious. But for SMX I totally agree.

[262 ff] so conclusion is that almost entire NER for 2,4-D is due to bioNER? What is then NER measured in the sterile experiment? This cannot be bioNER. So this must be an abiotic NER formation in the sterile experiment - why does it not happen in the non-sterile experiment? Overestimation of bioNER by tAAS?

[325 ff] bioNER are rather underestimated? How does this match with the results of the sterile experiments?? Do all those processes not happen in non-sterile soil? Is there any other proof for higher bioNER like eg extensive mineralisation that could predict more bioNER by means of the MTB model? So why underestimated?

[342] Yes! And that is very similar finding to the sterile experiments. Question stays still, why SMX is so good in line with the sterile experiments and the other two not at all?

And there is still the gap between ¹³C and D in total NER: why is D always below the ¹³C? And we talk about a difference of more than 20% which is in magnitude – not relative of course – similar to the difference observed for 2,4-D. This were the concerns from the first manuscript draft.

[351] the three factors don't explain the sterile sample findings. You should refer more to that sterile results as they are important also for process analysis.

[365] yes, absolutely true! Completely agree.

[374] But H-NER are significantly lower than ¹³C-NER (and ¹⁴C-NER I would guess) – the observed difference of 20% can be highly relevant in persistency assessment. I doubt that this is acceptable by regulation.

Table 1: there is something wrong with the table:

cost: to my experience ¹⁴C is much more expensive than ¹³C. did you confuse that here?

safety concerns: here is everything mixed! ¹³C is not radioactive, and ¹⁴C has the half life of 5700 years. For T radioactivity is a concern. The penetration depth is of no relevance, they are both beta radiators with neglectable dose rate. The only concern is incorporation. Just let it out in the table, please.

work load: same work load for ¹⁴C and T as same instrumentation and principles are used. D should have higher workload. Again columns seems to be mixed.

NER analytics: seems also mixed, didn't you use D labels for quantification of biomolecules?

[403] This goes beyond the scope of the manuscript. If T labelling would have such advantages over ¹⁴C-labelling, it would have been used in regulation. But in contrast ¹⁴C has prevailed and very less labs are able to work with T (which needs a special radio license and cannot simply be applied in ¹⁴C-labs). This is due to the uncertainty of the H-shift between molecules. This is the major problem of T-labelling and even the authors put some effort to prove stability of the current C-H bonds, see also 424 ff. Safety concerns by penetration depths and half-life are irrelevant because of the very low dose rate. Incorporation is the major concern and this should be worse for T connected to water as result of mineralisation. CO₂ is assimilated by plants but not humans, water can be incorporated right away. Thus, handling with T is even more tricky than with ¹⁴C! The weaker beta energy means also weaker detectability. 'Finally CO₂ from combustion analysis can be easily trapped in NaOH-solution – trapping of water in a solvent is more tricky. No, T is not the better radioactive label as proven in 30 years of regulatory practice! The authors expertise is stable isotope labelling – please stay in this topic and delete this chapter about the radiolabels.

[422] agree. These kinetic isotope effects are generally neglected in regulatory testing.

[442 ff] bit to optimistic? Agree, it could be added to the ¹⁴C-label (and would produce additional radioactive waste). 12 or 5700 years half life would not fundamentally change the waste problem. All wastes have to be treated as radioactive waste following the same rules. 12 years might rather be a problem for longterm studies – e.g. lysimeter studies have a duration of up to 3 years – where a half life of 12 years will be noticeable already and makes evaluations more complicated.

Prove of concept: yes. Is a very nice idea and results indicate that this could be a chance to move forward with the bioNER problem. But not yet ready for application. This will need further research.

[470] wow, was very dry during storage!

[482ff] soil incubation experiments: the soil was dried at 40°C? And moistened only just before application of the test substances?? No preincubation at the incubation conditions? And sampling after 4 days? This is not at all representative for the soil. There must have been dramatic changes in the soil, when surviving microorganisms start to re-establish in the before dried soils. Microflora is not comparable to that of the original soil.

[503] 60% of the WHK is pretty wet in such silty soil and rather the upper moisture limit recommended in soil degradation experiments (40-60% WHK according to e.g. OECD 307)

[519] was there any storage stability testing of the substance in the freezer? How long was the storage before analysis?

[suppl S7] If there was total CO₂ determination, why no differentiation in ¹³CO₂? Would have been nice to have an idea on substance mineralisation, since current practice (MTB) is to calculate bioNER based on substance mineralisation.

Reviewer #2

(Remarks to the Author)

After careful revision of the manuscript and thorough consideration of the author comments, I believe the manuscript is now suitable for publication.

Version 2:

Reviewer comments:

Reviewer #1

(Remarks to the Author)

Review Nature communications 458241_2

Third review, line numbers from file "458241_2_merged_1722815324.pdf"

The current review has been done only on the revised manuscript. Most comments on the previous version have been corrected. The manuscript would be now ready for publication from my side. However, there is room for improvement in a follow-up study.

Last remarks:

[245ff] that is a very hypothetic explanation. In reality it is very unusual to have more NER in sterile samples than in biological active samples and is an indication that sterile and biological active samples were not really comparable. And your reference confirms that. And the same happens for substance no 2 [274 ff]– how likely is that? That is pretty unfortunate since the sterile sample serves as reference in this project.

[557ff] As already stated the incubation experiments are not at all in line with 307 requirements in terms of drying and re-wetting the soil material (not "agricultural soil"!) which might explain the unexpected results for NER in sterile and non-sterile soils. But it is well described in the manuscript and the experienced reader may judge herself/himself, if experimental procedures question the overall results of the study.

It's a pity that the reliability of the study results can be questioned due to technically questionable preparation and incubation of the samples and unknown sample stability during storage. Any analysis is only as good as the sample, this is also true for a "prove of concept" study. On the other hand, this leaves room for a follow-up study in which this part can be improved.

REVIEWER COMMENTS

Reviewer #1:

General: The manuscript presents a method to distinguish experimentally bioNER from xenoNER based on the hypothesis, that Hydrogen from an organic molecule will mostly get lost in the biological Carbon cycle. The hypothesis is scientifically sound and could provide a new way to identify bioNER. However, the work contains some mistakes/gaps that can hopefully be corrected/explained, as they can question the result of the good overall work. In general, NER are differentiated in the entire manuscript (see point by point comments) in “potential hazardous” xenoNER and “safe” bioNER. This differentiation is not correct as the xenoNER are further differentiated into NER type I and type II [Kästner, M., Trapp, S. & Schäffer, A. Consultancy services to support ECHA in improving the interpretation of Non-Extractable Residues (NER) in degradation assessment. Discussion paper-final report (2018)], and type II NER are assessed “safe sink” in current regulation. It is important to correct this in the manuscript because the current manuscript could give the impression that the differentiation has been done this way to increase the relevance of the work on the bioNER. Which is completely unnecessary given the remarkable idea and quality of the work.

Response: Thank you for this important clarification. The reason for our (perhaps overly) simplified distinction between only xenoNERs and bioNERs is that bioNERs are inherently safe, whereas xenoNERs may in principle pose a hazard. Yes, this depends very much on the type/binding mechanism of xenoNERs as well as environmental conditions favoring their release. In order to avoid any misunderstandings we revised the manuscript text to be more specific in this regard. Please also see our point-by-point responses to the comments below.

Technically, there are some more questions:

Comment 1: At no point it was investigated, if ^{13}C -NER and D-NER would provide similar results if it would be only xenoNER. This could have been easily clarified by performing additional sterile experiments. Without that information it is very difficult to assess if the differences reported are really due to bioNER. Maybe is a systematic gap – the result of SMX, where no bioNER could be found but a difference of up to 30% was observed between ^{13}C -NER and D-NER give some indication for that. In the manuscript this gap is called “insignificant”. But it raises the question for the other two substances where the absolute difference between ^{13}C -NER and D-NER is not higher than in SMX experiments.

Response: Thanks for this important remark. We really appreciate the suggestion to perform additional sterile incubations.

In the case of SMX the differences between D-NERs and ^{13}C -NERs were indeed surprisingly high. However, since ^{13}C -NERs exceeded 100%, we suspect a systematic overestimation. ^{13}C -SMX (but not SMX or D-SMX) could hardly be dissolved in water at the concentration required for spiking, and hence the aqueous spiking solution may have been inhomogeneous, although we did not see inhomogeneity with naked eye. This may explain the higher ^{13}C -NER contents. The 30%-difference between ^{13}C - and D-NERs was also ‘insignificant’ in Welch-tests after p-value correction, although unadjusted p-values indicated significant differences on day 18 and 72. When normalizing for the total isotope label recovery on day 0 (**Table S2a**), total $^{13}\text{C}_{\text{NERS}}$ and total D_{NERS} are nearly identical (Figure only for illustration and not included in the manuscript):

We included an explanation in the manuscript text, see below:

*‘Notably, higher ¹³C_{NER} values compared to D_{NERs} might be due to a higher amount of ¹³C_{6-SMX} accidentally added to the soil on day 0 (total ¹³C-label recovery on day 0: 130%, **Supplementary Table S2a**). SMX is a hardly biodegradable antibiotic expected to form mostly xenONERs; thus, the ¹³C_{NERs} in the biotic treatment should be nearly identical to the ¹³C_{NERs} and D_{NERs} measured in sterile soil (Fig. 3c)’. Line 255-259, pg. 10.*

Nevertheless, to exclude systematic differences, we also conducted new sterile experiments with all D- and ¹³C-compounds. In those experiments, both 2,4-D and SMX (but not GLP which is easily dissolved in water) were spiked in a methanol solution to prevent potential issues due to solubility (this was not done in the biotic experiments to avoid toxic effects on microorganisms). The results show nearly identical abiotic NER formation between the D and ¹³C labels for all compounds and at all sampling days. The results are now included in the manuscript text, see in pg 7-8, and below:

‘Total ¹³C- and D-labeled non-extractable residues (NERs) in sterile soils

*Total ¹³C-NERs and D-NERs were quantified by elemental analyzer-isotope ratio mass spectrometry (EA-IRMS) as the amount of isotope label remaining in the soil after exhaustive solvent-extraction of the test compounds (for details see **Supplementary Text S3**). The exhaustive solvent-extraction aimed at remobilizing soil-sorbed type I xenONERs that are of highest relevance for risk assessment. The total ¹³C_{NERs} and D_{NERs} from 2,4-D and GLP shown in **Fig. 3** thus comprise the sum of bioNERs, physically entrapped xenONERs type I, and xenONERs type II. Due to the lower SMX recovery in screening tests compared to 2,4-D and GLP (**Text S3**), the NERs from SMX might still contain strongly sorbed xenONERs type I.*

*Abiotically formed ¹³C_{NERs} and D_{NERs} of each model compound were nearly identical at all sampling dates (**Fig. 3a-c**; for detailed statistics see **Supplementary Text S4**) proving that D was stable against abiotic breakage of C–D bonds. On the final date, the lowest NERs were observed for 2,4-D (¹³C_{NERs}: 19±5.1%; D_{NERs}: 29±11%), followed by 2-GLP (¹³C_{NERs}: 46±7.4%; D_{NERs}: 46±8.1%) and 3-GLP (¹³C_{NERs}: 63±9.3%; D_{NERs}: 53±11%) while the highest NERs were formed by SMX (¹³C_{NERs}: 88±13%; D_{NERs}: 83±15%). The ¹³C_{NERs} from ¹³C_{6-2,4-D} were comparable to the 15±1.8% measured by Girardi et al. ⁵⁵ in a sterile soil after 32 days, indicating an overall low sorption of 2,4-D to soil, in line with previously reported sorption coefficients (mean logK_d across 271 different samples = 0.16) ⁶⁴. Contrastingly, higher amounts of both ¹³C_{NERs} and D_{NERs} from GLP and SMX suggest a higher sorption affinity, in accordance with higher sorption coefficients reported in literature for GLP (mean log K_d = 1.9) ^{65,66}*

and SMX ($\log K_f = 0.78$ in silt loam)^{67,68}. We noticed that the label position of GLP (2-C_{GLP} and 3-C_{GLP}) slightly affected the ¹³C/D_{NERs}, which for 3-C_{GLP} were somewhat higher than for 2-C_{GLP} (Supplementary Text S4.1.1). However, this divergence was consistent at all sampling dates, and might be due to slightly different amounts of added ¹³C or D-GLP to soil, especially of 3-¹³C-GLP, whose available amount was only ~ 1 mg.

Fig. 3: Total NER formation of the three model compounds in sterile soils. a: 2,4-D, b: GLP (2-C and 3-C labeling position) and c: SMX. Columns represent the average of triplicate measurements ($n = 3$) and error bars the propagated standard deviation considering the uncertainty of the total element (C/H) abundance as well as the isotopic abundance (at% ¹³C/¹²C or at% D/¹H) in the labeled treatment and unlabeled background. Corresponding formulas are provided in the methods section. Dots show the individual replicate values used for statistical analysis. These only represent the uncertainty of the isotopic abundance in the labeled treatment as the mean element abundances and isotopic background values were used for calculation of replicate values. Holm-adjusted p -values for Welch tests (2,4-D, SMX) or main effects of Kruskal-Wallis tests (GLP) are shown above the bars. Letters indicate statistically significant differences in Conover-Iman tests (p -adjusted < 0.05), and exact p -values for GLP are provided in Supplementary Text S4.1.1.

Comment 2: The term NER implies, that everything has been tried to extract mobile fractions of the test substance. In fact, apparently only a simple single extraction was carried out, the effectiveness of which has at least not been reported (e.g. recovery on day 0?). Without this information, serious concerns arise as to whether the extraction residue can actually be described as NER.

Response: Thank you very much for this hint. For each compound, a two-solvent extraction (water + organic solvent for 2,4-D and SMX; and water-borax for GLP) was applied. We now included details

about the extraction procedures and recovery rates directly after soil spiking with the unlabeled test compound as well as on day 0 (2 hours after soil spiking plus defrosting and sample processing) in the SI (**Supplementary Text S3**), along with NERs on day 0. In brief, total recoveries directly after soil spiking with unlabeled test compounds were as follows: 2,4-D: $>98 \pm 5\%$, GLP: $93\% \pm 3\%$ and SMX: $80\% \pm 10\%$. As stated in line 184-186, pg 7, we aimed at the remobilizing soil-sorbed type I xenoNERs that are of highest relevance for risk assessment. Therefore, ^{13}C or D-NERs comprised the sum of bioNERs, physically entrapped xenoNERs type I, and xenoNERs type II, with the exception of SMX, which may still contain low amounts of soil-sorbed type I xenoNERs due to the lower recovery compared to 2,4-D and GLP.

Comment 3: The experimental approach is a bit different from an OECD 307 test, but it also says “oriented at”, which is fine. However, it remains a bit unclear, how the setup was, how sampling and homogeneity of the samples were done. All in all, this let some doubts on the reliability of the results. Which I am sure is not necessary, so please try to make that a bit clearer.

Response: Thanks for pointing out missing details in the description of the experimental setup. We have added further information on the setup, spiking and sampling in the materials and methods subsection ‘Incubation experiments’. See in pg. 18-19, and below:

‘Incubation experiments

*Soil incubations oriented at OECD guideline 307¹⁶ were performed in a static system consisting of 250 mL Schott flasks filled with 60 g wet weight of the agricultural soil. Besides treatment with the D and ^{13}C -labeled compounds two different controls were included: untreated soil and soil spiked with unlabeled compound. Both controls were performed to obtain background isotopic abundances for the NER and bioNER calculations. For each test compound, analogous sterile experiments were conducted to verify whether D- and ^{13}C -labeled compounds behave identical under abiotic conditions. All treatments were conducted in triplicate, i.e. three separate flasks were incubated per treatment. Prior to begin of the experiments, the soil was oven-dried at 40 °C over multiple days until reaching a constant weight. After thorough manual mixing, the soil for sterile controls was separated into 250 mL Schott flasks and autoclaved three times (121°C, 40 min) on consecutive days, with the last autoclaving cycle on day 0 of the experiments. The remaining soil was stored in an airtight 2 L bottle and the moisture content was monitored gravimetrically. To minimize a ‘priming effect’ on microbial degraders, approximately four hours before starting the incubations, the soil moisture was adjusted to 20% WHC_{max} and the soil was again mixed thoroughly by manual stirring. The soil for biotic treatments was then weighed into 250 mL Schott flasks and spiking was performed separately for each bottle. To this end, aqueous solutions of the test compounds were added dropwise to the soil, corresponding to final concentrations of 50 mg kg⁻¹ dry soil for GLP, 20 mg kg⁻¹ dry soil for SMX and 10 mg kg⁻¹ dry soil for 2,4-D. These much higher than environmental concentrations were selected after prior testing to yield good resolution on the IRMS instruments. Estimated detection limits for $^{13}\text{C}_{\text{NERS}}$ and D_{NERS} of the three model compounds were derived as described in **Supplementary Text S6**. After spiking, the soil moisture was adjusted to 60% WHC_{max} to provide optimal growth conditions for microorganisms⁷. Each treatment was then homogenized by manually stirring for two minutes. In the abiotic treatments, the soil moisture was adjusted separately for each bottle immediately after spiking of the test compounds as no priming effect was expected. Spiking and sampling for the abiotic treatments was conducted under sterile bench conditions. In biotic treatments with the D-compounds and unlabeled controls, D-depleted water was used for the spiking solutions and moisture adjustment in order to lower the background D in EA-IRMS measurements of water-extractable D on day 0 (**Supplementary Table S2a**).*

The soil incubations were conducted in the dark at 20°C to prevent photodegradation of the model chemicals. Soil samples were taken non-destructively on days 4, 16/18 and 36/38 for GLP and 2,4-D and on days 18, 36 and 72 for SMX because of its slower turnover. Each time, ten roughly 0.5-2 g subsamples were taken carefully from different spots within the soil batch in order to prevent soil disturbance. Non-destructive sampling did not cause any noticeable disturbances in soil microbial activity as soil respiration showed a continuous decrease of CO₂ towards the end of the incubation periods (Supplementary Text S7). The soil was then pooled into one composite sample in a 50 mL Falcon tube (day 0: 6 g total, afterwards: 18 g total), homogenized by stirring with a spatula, and stored at -20°C until analysis. During spiking and sampling, treatments were handled in the same order to account for the required processing time.'

Comment 4: I very much like the idea presented, but there might be still a longer way before regulatory use as long as it is not clear how high the normal difference between C-NER and H-NER (whichever isotope) is in the xenoNER. Please find the point by point comments below.

Response: Thank you. We agree that omitting results from sterile experiments was a major limitation of the original manuscript, and hope that the added information can help to clarify the magnitude of the difference between C-NER and H-NERs. See also our response to **Comment 1**.

Point by point comments.

Comment 5: [21] xenoNER are not considered a potential hazard by definition. This is currently only true for type I NER. Type II NER are assessed as safe sink in regulation like the bioNER (Type III NER).

Response: It was originally phrased like this since also type II NER can be remobilized, though over much larger timescales than type I NERs (comparable to humic matter degradation rates, [Kästner, M., Trapp, S. & Schäffer, A. Consultancy services to support ECHA in improving the interpretation of Non-Extractable Residues (NER) in degradation assessment. Discussion paper-final report. 2018]). However, the risks associated with type II NERs are indeed much smaller than those of type I NERs (and considered negligible in recent publications/regulatory views). Therefore, we agree that the original statement may have been misleading. Hence, we re-wrote this in the caption of **Fig. 1** as well as in the manuscript text accordingly, see lines 65-67, pg. 3, and below:

'Depending on the stability of covalent bonds and environmental conditions, type II xenoNERs could be either 'hidden hazard' or 'safe sink', but they are generally considered hardly remobilizable¹⁹⁻²².

Comment 6: [28] disagree! See comment to [21]. xenoNER are not generally assessed a potential hazard and bioNER are not the only NER-type assessed a safe sink. The current text gives too much weight to the presented method – type II NER are assessed safe in current regulation.

Response: True – generic statement about potential risks of xenoNERs was deleted. Now it reads:

'Present-day methodologies using radiocarbon or stable (¹³C, ¹⁵N) isotope labeling cannot easily differentiate soil-bound parent chemicals or transformation products (xenoNERs) from harmless soil-bound biomolecules of microbial degraders (bioNERs).' Line 20-23, pg. 1.

Comment 7: [29] Why not Tritium-label? In the discussion T is also recommended Would be nice to mention T instead of "innovative H-isotope labelling".

Response: Both isotopes are now mentioned, see in Line 28-29, pg. 1

Comment 8: [38] I would not agree to "poorly understood". The sentence reads as we would not care contaminants in soils and just put it there to get rid of them. In particular for pesticides (2 of

the test substances are pesticides), fate processes in soils are very good investigated, identified and reported. NER are not good understood, but degradation processes are well known.

Response: Agree, the phrasing was a bit unspecific. Now we reworded it to emphasize uncertainties of bound residues rather than fate assessment in soil as a whole.

'Compared to water and air, assessing the fate of organic contaminants in soil is complicated by their binding to or entrapment in the solid matrix, making soils a potential 'temporary reservoir' or 'permanent sink' for contaminants^{2,8'} Line 36-38, pg. 2.

Comment 9: [42] ultimate biodegradation is not the only detoxification step and degradation of organic chemicals in regulation is mostly understood as primary degradation which can also happen abiotically. Transformation products might be still hazardous, but must not be. Mineralisation is no trigger in degradation assessment (except screening test and polymers).

Response: Thank you very much for this hint. We agree that mineralization of the C-compound to CO₂ and H₂O is not the only pathway for compound elimination. Also permanent binding of compounds can be regarded as an efficient elimination pathway. Therefore, we rewrote this sentence accordingly:

'Contrastingly, permanent binding of organic contaminants to soil or their mineralization to CO₂ and H₂O together with assimilation into microbial biomass can greatly reduce this risk^{2,6,10,11'} Line 40-42, pg. 2.

Comment 10: [54] There are rather two types of "xenoNER" that carry different hazards. This is also what is described in more detail in the following chapter. See comment to [28]. The differentiation "xenoNER = hazard" and "bioNER = safe" is just wrong. Type two xenoNER are assessed safe as well.

Response: Thank you very much. We clarified the different hazard carried by NERs type I, II and III in the manuscript text accordingly, see below and in lines 54-67, pg. 2-3:

'The first two NER types are xenobiotic NERs (xenoNERs) comprising the un-degraded parent chemical or its transformation products that are adsorbed or physically entrapped in soil (type I), or covalently bound to reactive surface groups of the soil (type II)¹⁹⁻²². In the environment, type I xenoNERs can detach from soil over varying time scales and thus pose a delayed 'hidden hazard'¹⁹⁻²². Changing environmental conditions, e.g., freezing, thawing, and altered precipitation, temperature, pH or vegetation cover due to global warming could all exacerbate the release of potentially harmful type I xenoNERs²³⁻²⁶. To identify potentially mobile residues in laboratory tests, exhaustive extraction schemes are advocated that remobilize soil-adsorbed type I xenoNERs by using a mixture of aqueous and organic solvents along with physical agitation, heat or pressure^{19,21,22}. The physically entrapped type I xenoNERs are 'freed' after breakdown of soil aggregates using silylation, while xenoNERs type II can only be released after bond breakage using acidic or alkaline hydrolysis^{19,27,28}. Depending on the stability of covalent bonds and environmental conditions, type II xenoNERs could be either 'hidden hazard' or 'safe sink', but they are generally considered hardly remobilizable¹⁹⁻²².'

Comment 11: Fig 1 The description doesn't match the figure. In the figure xenoNER are labelled "hidden hazard", which is not true (see comments above). The "¹⁴C-NERs" or "total NER" will always contain xenoNER and bioNER since total NER don't differ. So is not "can contain" but "contain". There is no technique to determine a total NER fraction that does not contain bioNER.

Response: Thank you very much for this suggestion. We changed for 'xenoNERs' as follows: type I: 'hidden hazard', type II: 'potential safe sink' and bioNERs type II: 'safe sink' in the **Fig. 1**. We also deleted 'can' in the figure caption, see in line 81, pg 3.

Comment 12: [87] Again the same as before. xenoNER are no hazard by definition. Please correct.

Response: thank you. We corrected it according everywhere in the manuscript text as well as in the caption of **Fig. 1**. See also our response to **Comment 10 & 11**.

Comment 13: [89] I don't see the difference between ^{14}C and ^{13}C . Even if it is not part of the routine (though xenoNER differentiation is already allowed in regulation and might become part of the routine very soon) all analysis described below can be done with ^{14}C as well. Both isotopes can be analysed by GC-MS, by GC-IRMS and also by HR-LC-MS. Everything that can be done with the ^{13}C -isotope label can be done the same with the ^{14}C -isotope label. With the advantage, that the ^{14}C additionally give the option for a very easy radioactive analysis. Combination of ^{14}C -labels with ^{13}C and ^{15}N -labels are also pretty common in pesticide regulation to enable transformation product identification by NMR-techniques.

Response: Thank you for this feedback. We agree with that; therefore, we re-wrote this sentence in order to stress more the importance of H-labeling over the C & N, see below and in lines 89-96, pg. 4.

'Although differentiation between xenoNERS (type I and type II) and bioNERS (type III) is vital to assess the hazard posed by organic contaminants in soil, it is not part of routine biodegradability tests with ^{14}C -tracers, which readily quantify only total $^{14}\text{C}_{\text{NERS}}$ (Fig. 1) ^{15-17,20,21,32,33}. Well-established protocols for identification and rough quantification of bioNERS based on the analysis of stable carbon (^{13}C) or nitrogen (^{15}N) isotopes in biomolecules in biodegradability tests with ^{13}C or ^{15}N -tracers are very laborious. Both ^{13}C or ^{15}N -labeling is also limited by the scarce supply and high costs of ^{13}C or ^{15}N -labeled compounds, and ^{15}N is additionally restricted to N-containing molecules ^{7,8,22,34-36}.

Comment 14: [94] Could you explain, why that should be cheaper? Analysis of Tritium (radioactive H – Isotope) is very simple and doesn't need expensive sample pre-treatment and HR-MS techniques. Though the D-labelled substance itself is surely cheaper than T-labelled substances, that I would agree.

Response: Was meant as alternative to ^{13}C or ^{15}N , which both requires laborious bioNER analytics due to their high retention in bioNERS and limited availability on the market. T may still be better than D, but it was more difficult to prove the concept by quantifying T-amino acids as the protocols for analyzing the T in biomolecules are not developed in our laboratory. For D, we could easily amend existing protocols (for ^{13}C and ^{15}N) to readily use D in this 'proof-of concept' study. We amended part of this sentence as follows below, but we also discussed the feasibility of T-labeling in more detail in **Discussion** (for more details, please refer to our response to **Comment 29**), which will be a more appropriate 'place' for a broad audience to understand the overall concept of the paper.

'We here show that the stable H isotope – deuterium (D) – can be used as an alternative to existing tracers that is both cheaper and more accessible than ^{13}C or ^{15}N due to the common use of deuterated standards in analytical chemistry ³⁷. Furthermore, hypothesizing a minimal retention of H in biomolecules and thus in bioNERS, H-labeling could enable a time-efficient distinction between xenoNERS and bioNERS.' Line 96-100, pg. 4.

Comment 15: [104] "C- or N isotope labelled substrate". Is not limited to ^{13}C .

Response: Thank you, we amended it accordingly and explicitly mention the applicability for ^{14}C and T as well in the caption of **Fig. 2** (line 125-126, pg 5):

'Fig. 2: Formation of bioNERS during the biodegradation of a heavy C- or H-labeled substrate (here: ^{13}C - and D-glyphosate). Please note that the same processes apply to ^{14}C and T (tritium) as well, but are for simplicity only shown for $^{13}\text{C}/\text{D}$.'

However, please note that the **Fig. 2** has been changed a bit; therefore, the N has been deleted in the revised version of manuscript.

Comment 16: [114ff] Agree! That's the reason, why T-labelling is not so commonly used in those studies. Because it might be exchanged and then the label is not very specific for the original test substance any more.

Response: thank you for this comment. We now discuss in more detail the applicability, limitations and advantages of T-labeling in the discussion, including potential issues due to label stability. Please note that the original statement was therefore deleted from the introduction. For more details, please refer to our response to **Comment 29**.

Comment 17: [123] Good idea. I would support that hypothesis.

Response: Thank you. We also stressed out more the importance of application of H-labeling for a rapid quantitation of xenoNERs in the discussion, see below:

'This proof-of-concept study has shown that D or T labeling could become a powerful substitute or supplementary method for rapid xenoNER quantification along with ¹⁴C-labeling. By minimizing consumables including radioactive waste, rendering the process of NER analytics more efficient, and helping to effectively identify potentially hazardous long-lasting organic contaminants, the presented H-isotope labeling approach could contribute to the advancement of green chemistry within chemical safety testing. As long as H labels are stably attached, D- or T-labeling could be broadly applied for scientific and regulatory biodegradability testing of a wide range of organic chemicals to reveal the hidden identity of NERs.' Line 442-449, pg.16.

Comment 18: [130] Isn't the control a "¹²C-label" = non labelled (see line 356)?

Response: Correct, the ¹³C was a second (comparative) treatment and both D and ¹³C had their respective unlabeled ¹H/¹²C controls. ¹³C could however also be thought of as a 'control' for comparably high bioNER formation; but this may sound confusing so we changed it. Please see below:

'The ¹³C-labeling served for the comparison of biogenic and total NERs formation between H and the standard C isotope tracers.' Line 163-164, pg 6.

Comment 19: [141] If the amino acids are quantitative marker for bioNER, why is the D-labelling needed? In that case the ¹³C amino acids should serve the same. Of course for proof of concept is fine.

Response: True, ¹³C-amino acids can also help to estimate bioNER (and consequently xenoNER) contents. However, amino acid analysis requires laborious extraction, purification and derivatization, as well as analysis by both GC-MS and GC-IRMS. The motivation behind the D-labeling approach is to minimize the time and resources needed for NER characterization. Moreover, the ratio of tAAs to total bioNERs is only estimated as roughly 1:2, but it may differ depending on the compound analyzed and its turnover, and hence sometimes doesn't give an accurate quantitative result. As an example from our results (results section '**Proportion of ¹³C- and D-biogenic NERs (bioNERs)**, pg 10-13) within the total NERs') - the tAA biomarker gave a good approximation of total bioNERs for 2,4-D. However, this was not the case for 2-¹³C-GLP, for which the total bioNER estimate was strongly underestimated. ¹³C-label from the 2-¹³C-GLP is routed via the sarcosine pathway; therefore, a high retention of the ¹³C in the bioNERs is expected due to a direct assimilation of 2-¹³C-glycine into microbial biomass and formation of unlabeled AMPA.

Comment 20: [159] How reliable are D-NER if there are even negative values $2.6\pm 3.9\%$?? What would be the limit of detection for that measurement?

Response: Thank you for raising this important issue. Throughout the manuscript we report propagated uncertainties which consider the standard deviations from three measurements: 1) total element (C/H) or amino acids contents in the dry soil, 2) isotopic abundances (at% $^{13}\text{C}/^{12}\text{C}$ or at% D/H) in the labeled treatment, and 3) isotopic abundance in unlabeled control. Hence relatively high (relative) standard deviations can occur, especially when the measured isotopic abundances are very close between the labeled and unlabeled samples and approach the limit of detection. Please see also the paragraphs on 'Data analysis' in the **Methods** section for a mathematical description of the uncertainty propagation. Estimating the LOD for total NERs and amino acids is thus not straightforward as three different types of measurements are required to get one value for NERs/AA. However, we attempted to estimate the LODs for each of our model compounds based on generic specifications of the precision of isotope ratio measurements on IRMS instruments and based the measured uncertainty of the background isotopic abundances in our soil as described in **Supplementary Text S6**. The estimated method LODs for our NER measurements ranged from 3-5% of the applied isotope and for individual AAs from 0.01-3.1% of applied isotope. However, these estimates may be more applicable for limits of quantification and especially for AA they are rather conservative as they were based on standard deviations of background isotopic abundances averaged over all incubation time points, hence considering any possible biological variation. IRMS may be able to achieve much lower LODs for total NERs and bioNERs for more homogenous samples or when different soil sample weights (for EA-IRMS) and injection volumes (for GC-IRMS) are chosen. Please also note that for the specific case of NERs from 2,4-D, the calculation of mean total C/H per soil and its uncertainty was slightly changed from the previous manuscript version. Initially, a few control samples of unextracted soil (i.e. not NERs) had been accidentally included when calculating mean and standard deviations of the total element abundances. After revisiting the data, we now excluded them and only calculated the total element abundance from soil samples after extraction in the labeled and unlabeled treatments.

Comment 21: [176ff] I would not say that 20% to 30% difference between ^{13}C -NER and D-NER is not significant. The absolute difference in the 2,4-D experiment was even less than that. Though I see the point. However, this shows a weak point in the all over work: the missing sterile samples and the insecurity how much would be the difference of ^{13}C -NER and D-NER in the xenoNER without bioNER. According to [267ff], no bioNER were formed at SMX.

Response: Thank you very much. We agree with this suggestion. Therefore, we performed additional sterile incubations for all three tested compounds, which results showed comparable contents of ^{13}C - and D-NERs for each compounds. More details were included in **Comment 1**.

Comment 22: [181] You compare to ^{14}C -NER in literature only for SMX. Very nice. But why not for GLP and 2,4-D? Both are pesticides and well investigated as ^{14}C -label. Data should be available in literature. Would be nice to have those data for comparison.

Response: We agree that we missed that. Now you can find also comparison of our data with these obtained by others. Please see below:

'The $^{13}\text{C}_{\text{NERs}}$ from $^{13}\text{C}_{6-2,4-D}$ were lower than previously reported for $^{14}\text{C}_{6-2,4-D}$ ($26\pm 0.2\%$)⁶⁹ and $^{13}\text{C}_{6-2,4-D}$ ($39\pm 2.6\%$)⁵⁵ in soils with similar properties.', line 224-226, pg. 9.

‘Overall, $^{13}\text{C}_{\text{NERs}}$ from both 2- and 3- $^{13}\text{C}_{\text{GLP}}$ were close to the previously reported 40-50% of initially added ^{13}C ^{58 61} as well as ~40% of initially added ^{14}C ⁷⁰’. line 248-250, pg. 10.

Comment 23: For all three substances: why didn't you perform a sterile control as recommended by OECD 307? In this case maybe NER would have been much lower (except SMX probably) but would have been a nice proof of your hypothesis since ^{13}C -NER and D-NER should be almost identical if the hypothesis is right.

Response: Thank you. We performed sterile incubations as suggested. More details have already been included in **Comment 1**.

Comment 24: [268] please delete the “hidden hazard” as this is not proven. Is rather the opposite when the NER are mostly abiotic NER (a sterile control would have given important additional information), it can be speculated that this is due to covalent bonding to the soil matrix. Covalently bound xenoNER are mostly type II NER and as such considered a “safe sink”.

Response: Thank you. We amended it accordingly throughout the manuscript. See also our responses to **Comments 10-12**.

Comment 25: [274] Just 7 lines before its said that no bioNER are found in the SMX experiment. Does not match.

Response: We agree, the statement did not apply for SMX. We changed it accordingly:

‘The D-labeling approach showed lower estimates of total NERs for all three model compounds 2,4-D, GLP and SMX as compared to ^{13}C -labeling. For the two biodegradable herbicides 2,4-D and GLP, also D_{bioNERs} were lower than $^{13}\text{C}_{\text{bioNERs}}$ while no bioNERs was measured for the antibiotic SMX.’ line 345-348, pg. 13.

Comment 26: [275] How do you explain the 20-30% difference of NER at the SMX then?

Response: thank you. We explained this difference in **Comment 1** as well as in the manuscript text.

Comment 27: [286] I don't find it very protective to overestimate the bioNER at SMX by about 30%! For this hypothesis it needs to be figured, how much the difference of C-NER and D-NER really is. This work gives clear indication, that D-NER are generally lower than C-NER.

Response: Thank you. Due to a substantial revision of the discussion section, the ‘protective risk assessment’ has been deleted. Furthermore, we suggested a possible reason for 30% overestimation in **Comment 1**.

Comment 28: [291] please delete “potentially hazardous”. This is a mistake through the entire manuscript.

Response: thank you. We deleted it throughout the manuscript.

Comment 29: [304-309] right. Both, D and T labelling might be possible. That's why I don't see the difference you state in line 89. Please check line 89 again and modify.

Response: Thank you. We amended that accordingly and specified the changes in **Comment 13**. We also discussed a possibility of using D- and T-labeling and limitations in the discussion part, please see below:

‘Despite the advantages of D-labeling over C or N tracers in terms of associated costs, labor and identification of NERs, D-isotope measurements require specialized IRMS instruments which may not be widely accessible. Moreover, the experiments in this proof-of-concept study had to be performed at much higher than environmentally relevant test compound concentrations to achieve acceptable detection limits against the natural stable isotope abundances in soil (0.015 at% D and 1.08 at% ^{13}C ;

Supplementary Table S6). The minimum required spiking concentration for D_{NER} quantification depends on both the number of labeled versus unlabeled H atoms in the test compound as well as the moisture and total H content of the soil. Multiple-position D-labeling may thus enable lower spiking levels but is constrained by the number of stable C–H bonds per test compound.

Therefore, to conduct biodegradation studies and NER identification at environmentally relevant concentrations, the radioactive H-isotope – tritium (T) – could be applied as a promising substitute for D. Radioisotope tracing using standard liquid scintillation counters (LSC) is widely established and currently the preferred method for rapid and reliable quantification of isotope label distribution within a test system^{3,7,15–18,21,30}. Unlike for stable isotope tracers, radiolabeled compounds can typically be mixed with their unlabeled counterparts to the desired spiking concentration and still give quantifiable signals. T tracing is generally cheaper and more sensitive than using equal amounts of ^{14}C as its specific radioactivity is much higher (1.066 TBq/mmol or 28.7 Ci/mmol) than that of ^{14}C (2.309 GBq/mmol or 0.0624 Ci/mmol)⁷⁵. Moreover, T is a weak β -emitter unable to penetrate human skin and a relatively fast-decaying radioisotope (half-life of 12.3 years compared 5700 years for ^{14}C), making it a safer and easier to handle alternative than ^{14}C ^{76,77}.

H isotopes may however be more prone to label instability, kinetic isotope effects or cellular toxicity⁵¹ than C isotopes, potentially raising doubts about the validity of result obtained by D or T tracing. Substrate-H bound to O, S or N is generally considered exchangeable, whereas many C–H bonds are stable unless enzymatically cleaved by microorganisms^{38,78}. However, the stability C–H bonds of may be compromised depending on their position within a molecule or ambient conditions (see also **Text S5**). We therefore recommend to carefully select appropriate C–H label positions that are stable against abiotic cleavage and, if needed, perform complementary stability tests in water or abiotic soil.

Kinetic isotope effects were previously reported for enzyme-mediated metabolism and may be aggravated for H due to the large relative mass difference between D or T vs. ^1H ^{75,77}. However, they may work in favor of the presented H-labeling approach as microorganisms were found to discriminate against D (and possibly T) uptake^{51,79}. This may be due to a higher activation energy required for the synthesis of C–D compared to C–H bonds or potential toxicity of very high D_2O concentrations to microbial cells^{51,74}. Abiotic processes driving xenoNER formation should however not be fundamentally altered by the higher mass of D or T vs. ^1H , so that identical xenoNER quantities are expected for compounds labeled with $^{13}\text{C}/^{14}\text{C}$ or D/T at stable C–H bonds.’ Line 387-423, pg. 15-16.

Comment 30: [313] LSC instruments (e.g. Perkin Elmer) are capable to analyse T since many years. T-labelling is used in pesticide regulation, although not as widely used since a H-shift is always a worry to give unreliable results. But in general T-labelling for testing is well established already.

Response: Thank you. We agree with that. We changed it in the discussion, see the response above to **Comment 29**.

Comment 31: [333] Most likely typical for an actual agricultural soil – but for substance testing OECD 307 asks a soil that did not see the test chemicals in the last 4 years (OECD 307, No. 26).

Response: Yes, that is correct, however, we did not strictly follow the OECD 307 approach in every aspect but just ‘oriented at OECD’ (see line 162, pg 6) to use it as a guidance to base our own setup on. We agree that the OECD guideline suggests to use a reference soil which did not see any chemicals in the last 4 years. However, in case we would have a pristine soil, we may not see any biodegradation for e.g. GLP or 2,4-D, which we actually wanted to have. Our goal was to test compounds with different biodegradation potential – high, medium and low/none, which was also

needed for testing our hypothesis with regard to H-retention in microbial biomass. In case of pristine soil, it would be harder to predict biodegradability and achieve this goal.

Comment 32: [359] That is unfortunately a big disadvantage of the stable isotope labelling. This could be an issue for test substances that are unknown toxic to soil microorganisms.

Response: Thanks for pointing this out. Indeed, excessive concentrations may be problematic, but are needed for reliable results against stable isotope natural abundance. Therefore, radioactive T could be a good solution in favor of the lower concentration and thus reduction of the potential concentration-induced toxicity (see the revised text of the discussion in our response to the **Comment 29**).

Comment 33: [367] Do I understand that right? The individual samples were 60g wet weight soil (2 mm sieved) and no sacrificial sampling was done but subsampling from these little bottles? How did you homogenise the sample after application? This is very difficult in those test bottles in particular with the silty soil used. Thus, the normal procedure would be sacrificial sampling. What was the size of the subsample taken at different timepoints? Please explain.

Response: Please see pg. 18-19 in our response to **Comment 3** for details. We agree that ideally, a larger soil batch with a smaller fraction removed as subsamples per day would have been used, or alternatively, separate bottles with sacrificial sampling. However, due to the high costs of the isotope-labeled compounds and their limited availability during the time of the experiment (only <100 mg custom-synthesized in some cases), the amount of soil had to be minimized. While this setup was not ideal to mimic a realistic, undisturbed soil system, the aim of the study was not to assess the fate of the compounds strictly according to OECD 307 or as close to field conditions as possible, but rather to compare the results obtained by D and ¹³C tracers under identical experimental conditions. However, we tried to avoid soil disturbance during sampling by taking soil samples carefully from the batch and avoided soil mixing. Moreover, our soil respiration data did not show any noticeable disturbances in soil microbial activity as soil respiration showed a continuous decrease of CO₂ towards the end of the incubation periods (see **Supplementary Text S7**). However, potentially small variations between treatments due to the sampling/homogenization procedure cannot be 100% excluded despite our attempts to avoid them. This was a tradeoff we had to accept due to the limited availability of the isotope-labeled compounds.

Comment 34: [372] How was the recovery of the substances direct after application with those extractions? Any data available? This is very important to assess if you really analysed “NER” or rather a fraction that is not extractable with the applied simple extraction step. The entire manuscript is about NER but this short description raises doubts as to whether NERs have been studied at all or rather another fraction.

Response: Thank you. We tested the recovery of three tested compounds directly after the spiking. The results were as follows: 2,4-D: >98 ± 5%, GLP: 93% ± 3% and SMX: 80% ± 10%, see **Supplementary Text S3** and our response to **Comment 2**.

Reviewer #2:

Comment 1: The manuscript titled 'Hydrogen Isotope Labeling Unravels the Origin of Soil-Bound Organic Contaminant Residues in Biodegradability Testing,' authored by Lennartz et al., addresses a timely concern, specifically the evaluation of various types of non-extractable residues (NERs) of micropollutants in soil. The scientific significance of this study lies in demonstrating that a deuterium-labeling approach may be both straightforward and adequate for protective NER hazard assessment, as these isotopes exhibit lower retention in the non-hazardous microbial biomass compared to the ^{13}C tracer. The implication of the study is noteworthy as it reveals that deuterated or tritium isotopes can be utilised to quantify the remaining non-extractable fraction of micropollutant residues in soil, serving as a valuable proxy for xenobiotic NERs. While the manuscript is well-conceived and written, some drawbacks merit attention and clarification. Conceptually, I am not convinced that Figure 2 allows for an easy grasp of the fundamental differences in using $^{13}\text{C}/^{15}\text{N}$ or D-labeled substrates. The potential advantages of using D labels are not sufficiently clear, as processes such as biosynthesis (utilization of H_2O as an H source), dilution of D_2O to water along with limited microbial uptake, and isotopic fractionation for D/H are not highlighted at the mechanistic levels and in a quantitative manner. Consequently, strong arguments in favor of D labels may be overlooked. I suggest a thorough revision of the conceptualization of using D-labeled substrates to reflect the current state of the art and to provide more consistent quantitative arguments supporting the authors' idea. Concepts such as comparative analytical drawbacks, costs, sensitivity, labeling positions, etc., may be emphasized.

Response: Thank you very much for this valuable feedback. We changed **Fig. 2** to better show the mechanism driving C and H flow during catabolism and anabolism. We also provided – where possible – quantitative estimates, e.g. for the dilution of D_2O with unlabeled water based on our results. We further included some information on the difference in ^{13}C vs. D turnover dynamics based on published papers. However, these estimates may vary depending on amount of D/ ^{13}C in a substrate, biodegradability of the D-substrate to D_2O , soil moisture, microbiome, etc. Therefore, these quantitative estimates were only included in the text, but not in the **Fig. 2** as we can at this stage not make generic quantitative statements. We also cannot give exact prices for each isotope tracer as this varies each year, depending on the market needs/interest in a given isotope-labeled compound as well as the prices of the starting products. However, we included **Table 1** comparing the applicability of carbon (^{13}C and ^{14}C tracers) and hydrogen (D and T) for characterization of NERs, which is now included in the **Discussion** section.

See the revised text and **Fig. 2**, as well as new **Table 1** below:

*'Because of different turnover dynamics, H is expected to be much less retained in microbial biomass than C. In a prior one-year incubation study, for example, substrate-derived H in soil and microbial lipids was respectively 6-fold and over 10-fold lower compared to substrate-C³⁸. The main reason is that total and bioavailable C is limited in soil whereas H – due its presence in soil water – is approximately 6-11 times more abundant considering the stoichiometric requirements to build different biomolecules (see **Supplementary Text S1**). Microorganisms therefore constantly recycle C-substrates and necromass (biomass residues) of primary degraders for both energy contained in C–H bonds (catabolism) and C-building blocks required for biomass synthesis (anabolism) (**Fig. 2**)^{39–43}. Substrate- or necromass-derived C also can be directly assimilated as a C-monomer (e.g. amino acid) into macromolecules (e.g. proteins). Moreover, also unutilized C in decaying microbial biomass is eventually stabilized in the soil matrix as bioNERS^{8,18,44,45}. C can be slowly released from decaying soil biomass as CO₂ (**Fig. 2**) before re-uptake by newly growing microorganisms during CO₂ fixation^{8,18,44,45} (accounting for ~4% of released CO₂ from 2,4-D⁴⁶). Therefore, during microbial degradation of a ¹³C-labeled substrate, a high retention of the ¹³C tracer in bioNERS is generally observed.*

*In contrast, the direct incorporation of D from a D-labeled organic substrate into microbial biomass is expected to be low. After catabolic C–D bond cleavage, substrate-derived D is first released within a few minutes⁴⁷ from coenzymes to ambient water (**Fig. 2**) as explained in more detail in **Supplementary Fig. S7**. While this process is fast, the dynamics of enzymatic C–D bond cleavage will vary depending on the biodegradability of the D-substrate. For example, in monomeric biomolecules like glucose or amino acids, 70% of the C–D bonds were broken already within the first 7 days of soil incubation³⁸. Line 101-122, pg. 4.*

*'Besides substrate-H, ambient water provides a highly abundant H source for de novo formation of C–H bonds in C-monomers during anabolism^{48–50}. In prior studies with heavy water (D₂O), up to 79% of the D-water was assimilated into microbial biomass⁴⁸ and incorporation was already visible after 20 minutes⁵¹. However, due to a strong dilution of the substrate-derived D with ambient water-H, potentially amplified by isotopic fractionation^{50,52,53}, D reuptake into newly synthesized biomass and thus into bioNERS will be low. The dilution of D₂O with ambient water varies depending on the substrate concentration, biodegradation pathways and soil water contents but was estimated to result in D concentrations only about 0.001% above its natural abundance (0.015 at%) for substrate concentrations between 10-50 mg kg⁻¹ dry soil (**Supplementary Text S2**). A high retention of substrate-D in bioNERS could only occur during assimilation of D-monomers retaining C–D bonds of the primary D-substrate into macromolecules (e.g. glycine from glyphosate; **Fig. 2**)'. Line 146-156, pg. 6.*

Fig. 2: Formation of bioNERs during the biodegradation of a heavy C- or H-labeled substrate (here: ¹³C- and D-glyphosate). Please note that the same processes apply to ¹⁴C and T (tritium) as well, but are for simplicity only shown for ¹³C/D. The C/H-cycle starts with breakdown of C–H/D bonds of the C–D-substrate by degrading microorganisms in multiple oxidation steps (catabolism) to free both energy and C(¹²C/¹³C)-building blocks needed for biomass synthesis (anabolism). The C-building blocks mined during the central C-metabolism (e.g. acetyl-group generated from the pyruvate produced by the GLC pathway⁴⁸) are integrated into C-monomers during anabolism (1st level). Contrastingly, majority of the substrate-D after its C–D bonds breakage is first lost via the coenzyme NAD(D) into ambient water, where it dilutes strongly with unlabeled-H. Then, the D/H is transferred by the coenzyme NADP(D) to C-building blocks during de novo C–H bonds formation of C-monomers in anabolism⁴⁹. When biomass decays, the necromass is either assimilated by other living microorganisms (primary consumers) or stabilized in the soil matrix forming bioNERs. A direct assimilation of C-monomers (here glycine) into microbial macromolecules is also possible after a partial breakdown of C-substrates (like glyphosate) or C-necromass. ¹³CO₂ is slowly released from the soil matrix during microbial degradation at each trophic level whereas D₂O is lost more rapidly and hardly taken up again due to an estimated 100,000-times dilution with unlabeled H₂O. Hence the retention of an easily biodegradable H-substrate in soil is about sixfold lower (5%) after one year than for C (30%)³⁸. The N-cycle is similar to the C-cycle except that N-substrates are mineralized to gaseous N₂O or N₂; and the substrate-derived N, e.g. NH₂-group is transferred to C-monomers (e.g. amino acids) in anabolism; therefore, it is not shown or further discussed here. GLC: glycolysis, PPP: pentose phosphate pathway, TCA: tricarboxylic acid cycle, NAD(P): nicotinamide adenine dinucleotide (phosphate), NAD(P): oxidized; NAD(P)D: reduced.

Table 1. Comparative advantages and disadvantages of H vs. C tracers for NER characterization.

Isotope	¹³ C	¹⁴ C	D	T
Applicability	All C-compounds		Compounds with stable C–H bonds	
Availability of compounds	Depending on market trends		Widely (internal standards)	Limited
Costs	Medium to high	Low to medium	Low to medium	Medium to high
Sensitivity	Limited ($\delta \leq 0.3\%$)	Good (2.309 GBq/mmol)	Limited ($\delta \leq 5\%$)	Very good (1.066 TBq/mmol)
Availability of Instrumentation	Limited (IRMS)	Widely (LSC)	Limited (IRMS)	Widely (LSC)
Safety concerns	Radioactive waste: 5700 years Penetration depth (air): 24 cm	Radioactive waste: 12.3 years Penetration depth (air): 6 nm	Not relevant	
Workload (total NERs)	Low-medium	Low	Low-medium	Low
Workload (NER identification)	Very high	Medium to high	Low	Very low
NER analytics	type I (extraction + silylation), type II (bond cleavage), type III: (biomolecule extraction)		type I (extraction + silylation), type II: quantification as total NERs	

Bq: becquerel, G: giga, T: tera, D: deuterium, T: tritium, IRMS: isotope ratio mass spectrometry, LSC: liquid scintillation counter

Comment 2: Regarding the experimental design, I find the description of the incubation experiment to be insufficient, and I have reservations about the generic nature of the experiments and related data. From the experimental set-up, it is not clear how many replicate experiments were eventually used and whether sacrificial or non-sacrificial experiments were conducted. This is crucial for evaluating the reproducibility and robustness of the findings. It is mentioned (L. 366-368) that soil samples were collected non-destructively from different spots within the soil batch on different days. I am curious about how this is possible without a sacrificial approach. If a non-sacrificial approach was used, this may have severe consequences on the experiments since experimental conditions may be disturbed after each sampling, potentially affecting NER distribution. Moreover, standard biological experiments typically require at least 5 replicates, independent experiments, which seems not to be the case in this study. This requires clarification and a thorough discussion of the consequences of the experimental set-up on the results.

Response: Thank you for this feedback. We agree that the description of the experimental design in the Method section was too short. Therefore, we added more details about the set-up, replicates, sampling procedure. Please see pg. 18-19 and the text below for details. We agree that ideally, a larger soil batch with a smaller fraction removed as subsamples per day would have been used, or alternatively, separate bottles with sacrificial sampling. However, due to the high costs of the isotope-labeled compounds and their limited availability during the time of the experiment (only <100 mg custom-synthesized in some cases), the amount of soil had to be minimized. While this setup was not ideal to mimic a realistic, undisturbed soil system, the aim of the study was to compare the results obtained by D and ¹³C tracers under identical experimental conditions. However, we tried to avoid soil disturbance during sampling by taking soil sampling carefully from the batch and avoided soil mixing after the initial spiking. Moreover, our soil respiration data did not show any noticeable disturbances in soil microbial activity as soil respiration showed a continuous decrease of CO₂ towards the end of the incubation periods (see **Supplementary Text S7**). However, potentially small variations between treatments due to the sampling/homogenization procedure cannot be

100% excluded despite our attempts to avoid them. This was a tradeoff we had to accept due to the limited availability of the isotope-labeled compounds.

'Incubation experiments

Soil incubations oriented at OECD guideline 307¹⁶ were performed in a static system consisting of 250 mL Schott flasks filled with 60 g wet weight of the agricultural soil. Besides treatment with the D and ¹³C-labeled compounds two different controls were included: untreated soil and soil spiked with unlabeled compound. Both controls were performed to obtain background isotopic abundances for the NER and bioNER calculations. For each test compound, analogous sterile experiments were conducted to verify whether D- and ¹³C-labeled compounds behave identical under abiotic conditions. All treatments were conducted in triplicate, i.e. three separate flasks were incubated per treatment. Prior to begin of the experiments, the soil was oven-dried at 40 °C over multiple days until reaching a constant weight. After thorough manual mixing, the soil for sterile controls was separated into 250 mL Schott flasks and autoclaved three times (121°C, 40 min) on consecutive days, with the last autoclaving cycle on day 0 of the experiments. The remaining soil was stored in an airtight 2 L bottle and the moisture content was monitored gravimetrically. To minimize a 'priming effect' on microbial degraders, approximately four hours before starting the incubations, the soil moisture was adjusted to 20% WHC_{max} and the soil was again mixed thoroughly by manual stirring. The soil for biotic treatments was then weighed into 250 mL Schott flasks and spiking was performed separately for each bottle. To this end, aqueous solutions of the test compounds were added dropwise to the soil, corresponding to final concentrations of 50 mg kg⁻¹ dry soil for GLP, 20 mg kg⁻¹ dry soil for SMX and 10 mg kg⁻¹ dry soil for 2,4-D. These much higher than environmental concentrations were selected after prior testing to yield good resolution on the IRMS instruments. Estimated detection limits for ¹³C_{NERs} and D_{NERs} of the three model compounds were derived as described in **Supplementary Text S6**. After spiking, the soil moisture was adjusted to 60% WHC_{max} to provide optimal growth conditions for microorganisms⁷. Each treatment was then homogenized by manually stirring for two minutes. In the abiotic treatments, the soil moisture was adjusted separately for each bottle immediately after spiking of the test compounds as no priming effect was expected. Spiking and sampling for the abiotic treatments was conducted under sterile bench conditions. In biotic treatments with the D-compounds and unlabeled controls, D-depleted water was used for the spiking solutions and moisture adjustment in order to lower the background D in EA-IRMS measurements of water-extractable D on day 0 (**Supplementary Table S2a**).

The soil incubations were conducted in the dark at 20°C to prevent photodegradation of the model chemicals. Soil samples were taken non-destructively on days 4, 16/18 and 36/38 for GLP and 2,4-D and on days 18, 36 and 72 for SMX because of its slower turnover. Each time, ten roughly 0.5-2 g subsamples were taken carefully from different spots within the soil batch in order to prevent soil disturbance. Non-destructive sampling did not cause any noticeable disturbances in soil microbial activity as soil respiration showed a continuous decrease of CO₂ towards the end of the incubation periods (**Supplementary Text S7**). The soil was then pooled into one composite sample in a 50 mL Falcon tube (day 0: 6 g total, afterwards: 18 g total), homogenized by stirring with a spatula, and stored at -20°C until analysis. During spiking and sampling, treatments were handled in the same order to account for the required processing time.'

Comment 3: Another aspect is the use of high and environmentally non-relevant concentrations of the model pollutants. While high concentrations effectively allow testing the hypothesis posited by the authors, the findings may not reflect NER extent and distribution that may occur in the field. This is particularly true for sulfamethoxazole, which may occur at concentrations that are 2 to 3 orders of magnitude lower than those used in this study. This should be discussed and acknowledged. Additionally, it may be relevant to clearly state the limits of the proposed methodology at

environmentally-relevant concentrations in terms of sensitivity and uncertainty of bioNER and xenoNER analysis and quantitative estimation. This can be compared with the traditional approach using $^{13}\text{C}/^{15}\text{N}$ substrates.

Response: We agree that the concentrations of three tested substrates were well above concentrations found in the environment. However, we did not aim at biodegradability testing at environmentally relevant concentrations, but instead test whether the H-labeling approach could be used to ease NER characterization due to a minimal retention of H in bioNERs. High concentrations were necessary for reliable isotope detection essential for testing our hypothesis. Please see **Supplementary Text S6 (Tables S5a-S5d)** for estimated detection limits for total NERs and bioNERs in our experiments. Estimating the LOD for total NERs and amino acids is thus not straightforward as three different types of measurements are required to get one value for NERs/AA. However, we attempted to estimate the LODs for each of our model compounds based on generic specifications of the precision of isotope ratio measurements on IRMS instruments and based the measured uncertainty of the background isotopic abundances in our soil. The estimated method LODs for our NER measurements ranged from 3-5% of the applied isotope and for individual AAs from 0.01-3.1% of applied isotope. However, these estimates may be more applicable for limits of quantification and especially for AA they are rather conservative as they were based on standard deviations of background isotopic abundances averaged over all incubation time points, hence considering any possible biological variation. IRMS may be able to achieve much lower LODs for total NERs and bioNERs for more homogenous samples or when different soil sample weights (for EA-IRMS) and injection volumes (for GC-IRMS) are chosen.

We agree that biodegradability tests at environmental concentration are more relevant; therefore, future research should be directed towards that. We therefore gave some suggestions for future H isotope testing and limitations of different types of H isotopes in the **Discussion** section. Please refer to line 387-423, pg. 15-16 and below:

*'Despite the advantages of D-labeling over C or N tracers in terms of associated costs, labor and identification of NERs, D-isotope measurements require specialized IRMS instruments which may not be widely accessible. Moreover, the experiments in this proof-of-concept study had to be performed at much higher than environmentally relevant test compound concentrations to achieve acceptable detection limits against the natural stable isotope abundances in soil (0.015 at% D and 1.08 at% ^{13}C ; **Supplementary Table S6**). The minimum required spiking concentration for D_{NER} quantification depends on both the number of labeled versus unlabeled H atoms in the test compound as well as the moisture and total H content of the soil. Multiple-position D-labeling may thus enable lower spiking levels but is constrained by the number of stable C-H bonds per test compound.*

Therefore, to conduct biodegradation studies and NER identification at environmentally relevant concentrations, the radioactive H-isotope – tritium (T) – could be applied as a promising substitute for D. Radioisotope tracing using standard liquid scintillation counters (LSC) is widely established and currently the preferred method for rapid and reliable quantification of isotope label distribution within a test system ^{3,7,15-18,21,30}. Unlike for stable isotope tracers, radiolabeled compounds can typically be mixed with their unlabeled counterparts to the desired spiking concentration and still give quantifiable signals. T tracing is generally cheaper and more sensitive than using equal amounts of ^{14}C as its specific radioactivity is much higher (1.066 TBq/mmol or 28.7 Ci/mmol) than that of ^{14}C (2.309 GBq/mmol or 0.0624 Ci/mmol) ⁷⁵. Moreover, T is a weak β -emitter unable to penetrate human skin and a relatively fast-decaying radioisotope (half-life of 12.3 years compared 5700 years for ^{14}C), making it a safer and easier to handle alternative than ^{14}C ^{76,77}.

*H isotopes may however be more prone to label instability, kinetic isotope effects or cellular toxicity⁵¹ than C isotopes, potentially raising doubts about the validity of result obtained by D or T tracing. Substrate-H bound to O, S or N is generally considered exchangeable, whereas many C–H bonds are stable unless enzymatically cleaved by microorganisms^{38,78}. However, the stability C–H bonds of may be compromised depending on their position within a molecule or ambient conditions (see also **Text S5**). We therefore recommend to carefully select appropriate C–H label positions that are stable against abiotic cleavage and, if needed, perform complementary stability tests in water or abiotic soil.*

Kinetic isotope effects were previously reported for enzyme-mediated metabolism and may be aggravated for H due to the large relative mass difference between D or T vs. ¹H^{75,77}. However, they may work in favor of the presented H-labeling approach as microorganisms were found to discriminate against D (and possibly T) uptake^{51,79}. This may be due to a higher activation energy required for the synthesis of C–D compared to C–H bonds or potential toxicity of very high D₂O concentrations to microbial cells^{51,74}. Abiotic processes driving xenoNER formation should however not be fundamentally altered by the higher mass of D or T vs. ¹H, so that identical xenoNER quantities are expected for compounds labeled with ¹³C/¹⁴C or D/T at stable C–H bonds.'

Comment 4: It would be also of interest to discuss how D-labeled substrate may behave in different type of soil and under different soil conditions.

Response: Thank you for this feedback. We added some suggestions on that in discussion for future, see in line 424-441, pg. 16, and below:

'Further research is nevertheless still needed to verify the applicability of H isotope tracing across soils with diverse chemical and biological properties. NER formation was shown to vary substantially between soil types^{17,58}, e.g. as a function of pH, organic carbon and mineral content affecting the sorption of organic molecules. While the mechanism behind lower bioNER formation from H tracers is based on enzyme-mediated processes, making it in principle applicable independently of chemical soil composition, extreme pH ranges or a high abundance of transition metals could enhance the catalysis of abiotic H exchange in C–H bonds⁸⁰. Therefore, it may be of interest to compare abiotic NER formation between C- and H-tracers across a broad range of different soils. Moreover, the abundance and physiology of different types of degraders within the soil microbial community may largely affect the assimilation of H vs. C tracers into bioNERs. For example, CO₂ fixating autotrophs took up more water-H than heterotrophs but also showed stronger H fractionation⁵¹. Differences in substrate-H utilization were also observed between different heterotrophic degraders and for favorable vs. stressful growth conditions. In heterotrophic bacteria, water-H uptake occurred mainly during anabolism while substrate-H was mainly released to ambient water during catabolism⁴⁸. As a complex interplay between chemical and microbiological factors affects H-isotope incorporation into bioNERs, further studies are required to gain a better quantitative understanding of how much the difference between ¹³C_{bioNER} and D_{bioNER} formation may vary.'

Detailed comments:

Comment 5: L. 25: Use "proportion" instead of "percentage."

Response: Changed accordingly, see in line 26, pg. 1.

Comment 6: L. 96: Clarify the different turnover dynamics in the text and scheme.

Response: Thank you for this feedback. We explained the different turnover dynamics of ¹³C vs. D, both in **Fig. 2** and in the manuscript text. The D is turned over much faster than ¹³C due to instant D dilution with H-water after the breakage of C-D bonds. For example, a prior one-year incubation study showed substrate-derived H in soil and microbial lipids was respectively 6-fold and over 10-

fold lower compared to substrate-C. See also our response to **Comment 1** which includes more explanation as well as the new **Fig. 2**.

Comment 7: L. 112-113: Provide a quantitative statement. How rapidly is D₂O lost? How challenging is the uptake?

Response: Thank you for this comment. We have tried to provide some quantitative estimates based on literature and simplified calculations in line 117-154, pg. 5-6. However, please note that it is really hard to give an accurate number for how quickly D₂O is lost. For example, the study by Paul et al. (2016) showed 70% of the C–D bonds of easily biodegradable substrates like glucose or amino acids were broken already within the first 7 days of soil incubation; thus the D was released to ambient water as D₂O rapidly. Thereafter, after 28 days, only 5% of the initially added D-substrate remained in soil and which was constant throughout the year. However, these results are from a study conducted in a closed vessel system and at a constant temperature 20°C, where D₂O reached an equilibrium with the D-vapor. In an open system, the D₂O will be lost more rapidly, especially upon temperature variations. In general, the loss of D₂O from a D-substrate to ambient water is expected to be fast once microbial cleavage of C-D bonds begins as the involved enzymatic reactions process rapidly within minutes. This however implies that the velocity of D₂O loss also depends on how quickly microorganisms can access and degrade a given substrate, which is affected by a variety of different compound, soil and microbiome characteristics.

The uptake of H from water is easy and rapid, as ambient water is a highly abundant H source especially used by growing microorganisms in anabolism. Berry et al. (2014) showed that D uptake from D-enriched water was visible in microbial biomass already after 20 minutes (Berry et al, 2014). However, the D-uptake derived from a biodegraded D-substrate present in soil (at concentration 10-50 mg kg⁻¹) should be extremely low – maximum 0.001% of the total H-water – because the D₂O gets almost 100% diluted based on our calculations detailed in **Supplementary Text S2**.

Comment 8: L. 121: How quickly is the D in D-labeled substrate expected to be lost?

Response: After the C–D bonds of a D-substrate are cleaved during microbial degradation, the D is lost rapidly (within minutes) from coenzymes (NAD/H) to ambient water in catabolism (namely during oxidation-reduction processes shown in **Supplementary Fig. S7**). See also response to **Comment 7**.

Comment 9: L. 140-141: Offer more background and justification for the use of amino acids as a quantitative biomarker for bioNERS.

Response: We used amino acids to approximate the total bioNERS as they are the most abundant building blocks in microbial biomass and their stability in soil is high. This information is included in lines 540-543, pg. 20 of the manuscript text; see also below:

*‘Total amino acids (tAAs) were extracted from the soil as quantitative and qualitative markers for bioNERS. The tAAs from living and decayed biomass are the most reliable quantitative biomarkers for bioNERS as their turnover is comparably slow^{7,35}. tAAs can comprise up to ~50-55% of the microbial biomass, allowing for quantitative estimation of the total bioNERS (2*tAAs = bioNERS).’*

However, the ratio of tAAs to total bioNERS is only estimated as roughly 1:2, it may differ depending on the compound analyzed and its turnover and may sometimes not be a good quantitative estimate for total bioNERS. As an example from our results (results section ‘**Proportion of ¹³C- and D-biogenic NERS (bioNERS)**, pg 10-13) within the total NERS’) - the tAA biomarker gave a good approximation of total bioNERS for 2,4-D. However, this was not the case for 2-¹³C-GLP, for which the total bioNER

estimate was strongly underestimated. The ^{13}C -label from 2- ^{13}C -GLP is routed via the sarcosine pathway; therefore, a high retention of the ^{13}C in the bioNERS is expected due to a direct assimilation of 2- ^{13}C -glycine into microbial biomass and formation of unlabeled AMPA. Our motivation for using this new H-labeling approach was therefore to solve the limitations and uncertainty of bioNER analytics approximated by amino acid contents by skipping it entirely.

Comment 10: L. 173-174: Briefly explain why and how.

Response: We now explained it briefly (see below). To form AMPA, GLP is cleaved between the 2-C and 3-C position, and only the 3-C is preserved in AMPA. Therefore, any D or ^{13}C label at the 2-C position of GLP will not be retained in AMPA. Detailed degradation pathways are also included in **Supplementary Fig. S8 & S9**.

*'The differences in NER formation between D and ^{13}C tracers were thus less pronounced for 3-C_{GLP}. This could be due to the adsorption of labeled D₂-AMPA to the soil matrix, which is only formed from 3-C_{GLP} as only the third C of GLP is preserved in AMPA (i.e., 3-C-D₂-GLP, **Supplementary Fig. S9**)'. Line 245-248, pg. 10.*

Comment 11: L. 187-188: Clearly and scientifically justify the multiplication by 2 of labeled amino acids to estimate total bioNERS.

Response: The estimate is based on the mass fraction of amino acids within microbial biomass, which was estimated to be approximately 55%. As stated in our response to **Comment 9**, this is not an absolute number but rather an estimate of total bioNERS. It may not be accurate if isotope tracers are integrated to a different extent into amino acids and other biomolecules, e.g. carbohydrates or fatty acids. For more details, please refer to our response in **Comment 9**.

Comment 12: L. 191 and following: Clearly explain how the uncertainty associated with the proportion of NERS was derived.

Response: We have now included further details on how the uncertainty for different measurements was calculated in the 'Data analysis' section in the **Methods** part. The uncertainty in the tAA contents was derived by combining the propagated uncertainties in the contents of each individual amino acid (derived according to **eq. 3**, line 586) into one standard deviation for the sum of all amino acids according to **eq. 2** (line 581). The uncertainty of total bioNERS was estimated by applying the same conversion factor of 2 that was used to scale from mean tAAs to mean total bioNERS, thus maintaining the same relative standard deviation. See also line 571-593, pg. 21-22 or below:

*'D or ^{13}C enrichment in NERS and AAs was calculated as the percentage of D or ^{13}C initially applied with the labeled compounds and values are presented as mean \pm standard deviation. The detailed calculation of ^{13}C and D label incorporation into total NERS and tAAs is explained in **Supplementary Text S9**. The mean isotopic enrichment per treatment group ($mean_{enrichment\ tNERS/AA}$) was calculated as the product of the mean total element or AA abundance in soil ($mean_{\mu\text{mol}\ tNERS/AA}$) and the difference between mean the isotopic enrichment in the labeled treatment ($mean_{at\% \text{ labeled}}$) and unlabeled control:*

$$\begin{aligned} mean_{enrichment\ tNERS/AA} &= mean_{\mu\text{mol}\ tNERS/AA} \times (mean_{at\% \text{ labeled}} - mean_{at\% \text{ unlabeled}}) \\ &= mean_{\mu\text{mol}\ tNERS/AA} \times (mean_{at\% \text{ enrichment}}) \quad (\text{eq. 1}) \end{aligned}$$

Therefore, the uncertainty of the mean D or ^{13}C enrichment in NERS or individual AAs ($SD_{at\% \text{ enrichment}}$) was derived considering Gaussian error propagation as follows:

$$SD_{at\% \text{ enrichment}} = \sqrt{SD_{at\% \text{ labeled}}^2 + SD_{at\% \text{ unlabeled}}^2} \quad (\text{eq. 2}),$$

where $SD_{at\% \text{ enrichment}}$, $SD_{at\% \text{ labeled}}$ and $SD_{at\% \text{ unlabeled}}$ are the standard deviations of the mean isotopic (D or ^{13}C) enrichment, mean isotopic abundance in the labeled treatment, and mean isotopic abundance in the unlabeled treatment, respectively. The total uncertainty in the mean tNERS or labeled AA contents ($SD_{tNERS/AA}$) was calculated as:

$$SD_{tNERS/AA} = \text{mean}_{\mu\text{mol } tNERS/AA} \sqrt{\left(\frac{SD_{at\% \text{ enrichment}}}{\text{mean}_{at\% \text{ enrichment}}}\right)^2 + \left(\frac{SD_{\mu\text{mol } C,H \text{ or } AA}}{\text{mean}_{\mu\text{mol } C,H \text{ or } AA}}\right)^2} \quad (\text{eq. 3}).$$

Here, $\text{mean}_{at\% \text{ enrichment}}$ is the mean isotopic enrichment, and $\text{mean}_{\mu\text{mol } C,H \text{ or } AA}$ and $SD_{\mu\text{mol } C,H \text{ or } AA}$ are respectively the mean and standard deviation of the measured total abundance of C, H or the individual amino acid in the model soil. The total C and H abundance were measured over all sampling days as they were nearly constant while individual AA abundances were calculated separately for each sampling point. The uncertainty in the total amino acid abundance was calculated analogous to **eq. 2** by taking the square root of the sum of all squared standard deviations for individual AAs that were calculated according to **eq. 3**.

Comment 13: L. 217-219: Explain the meaning of this statement. Clarify why it is surprising.

Response: Based on our hypothesis, we would expect biodegradable compounds to form substantially higher amounts of $^{13}C_{\text{bioNERS}}$ than D_{bioNERS} , as was observed for 2,4-D (6 to 7-fold difference) or 3-C-GLP (5 to 25-fold difference). Therefore, the relatively small difference (2.6-fold on the final day) between $^{13}C_{\text{bioNERS}}$ than D_{bioNERS} from 2-C-GLP was surprising, especially since total NER contents differed by about 5-fold. We consequently explain that the reason for this relatively 'small' difference is the monomeric utilization of the glyphosate degradation product glycine as an amino acid building block for proteins. See also response to **Comment 14** below. We believe that now that this should be clearer after we explained the flux of ^{13}C - or D -derived substrate to microbial biomass in **Fig. 2**.

Comment 14: Fig. 5: Clarify in the text why this figure is important. Provide justification for the interest in this figure and related findings.

Response: Thank you. We moved the **Fig. 5** to Supplementary information and now it is **Supplementary Fig. S10**, because we only briefly mentioned in the text. As explained in the revised **Fig. 2** (see our response to **Comment 1**), ^{13}C or D label can be integrated into microbial biomass as a simple atomic building block (especially C), or as a monomer retaining the ^{13}C or D label directly into macromolecules after partial breakdown of the primary ^{13}C - or D -substrate. For example, as illustrated in **Fig. 2**, glyphosate can be partially biodegraded to the amino acid glycine, which retains the original ^{13}C or D from glyphosate. Therefore, our aim was to compare the fraction of ^{13}C - or D -glycine within the pool of total amino acids. The major and nearly identical contribution of both ^{13}C - and D_{glycine} from 2-C-labeled GLP to the total amino acid pool suggests that the assimilation of substrate-derived ^{13}C or D into bioNERS was mainly due to the direct assimilation of glycine into microbial macromolecules, especially in the sarcosine degradation pathway. This also explains why we observed higher D_{bioNERS} from 2- D_2 -GLP compared to D_{bioNERS} from 3- D_2 -GLP.

Comment 15: L. 286: Define what a 'protective NER hazard assessment' means.

Response: Thank you. Due to a substantial revision, 'protective NER hazard assessment' is no longer included in the discussion text in the revised manuscript.

Comment 16: Last paragraph: Clearly explain how D -labeled substrate may be used in a more generic way for different substrates, pollutants, soil types, etc.

Response: Thank you. We included this information in the discussion. The text was already included in our response to **Comment 4**.

Comment 17: L. 335: Provide a reference for the studied soil.

Response: Thank you for pointing that out. We missed that. We now included the reference by Naveed et al. (2014) in line 466, pg. 17.

REVIEWER COMMENTS

Reviewer #1 (Remarks to the Author):

Review Nature communications 458241_1

Second review, line numbers from file “458241_1_merged_1717358587.pdf”

The current review has been done only on the revised manuscript without considering remarks/comments on the first revision. So might be that comments of the first and second revision repeat. Still the project idea is outstanding but would have deserved a more careful planning of the experimental part considering principles from well established guidelines (e.g. OECD 307). Sterile experiments close a significant gap but results are only considered when it fits the hypothesis.

Response: Thank you very much for your feedback. We really appreciate it. We explained all raised issues and explained that we performed a proof-of-concept study which primarily aimed at comparing ^{13}C and D incorporations into biogenic NERs as well as testing C–D bond stability in sterile waters and sterile soils. Of course, we agree that this approach is not yet ready for an immediate use. Prior to its application for distinguishing between xenobiotic NERs and biogenic NERs further studies are needed which we suggested in the discussion of the manuscript.

Point by point:

Comment 1[64] using silylation or extraction with EDTA. **Response:** added as suggested, see in line 63-64, pg 2-3.

Fig 1: type II NER are considered safe sink, not “potential” safe sink. Type I NER are “potential” hidden hazard as you also describe in the figure description. Please correct, also in the figure. In contrast, the hazard of soil-bound contaminant residues is not clear as they can be ‘safe sink’ bioNERs type III or ‘hidden hazard’ xenoNERs type I. this is simply wrong and the added sentence does not correct it. Soil bound is covalently bound and as such it is not the original chemical anymore without any of the original properties. Bound residues are considered safe sink, not “potential safe sink”.

Response: Thank you, corrected as requested, please see an updated version of the Fig. 1.

Fig. 1: Biodegradability testing of the model compound glyphosate in soil using heavy isotope labeling (here ^{14}C) according to OECD guideline 307¹⁶. The ^{14}C -mass balance comprises easily quantifiable mineralization ($^{14}\text{CO}_2$), the ^{14}C -labeled solvent-extractable chemical or its transformation products, as well as the difficult to identify soil-bound ^{14}C as non-extractable residues (NERs). The $^{14}\text{C}_{\text{NERs}}$ are mostly quantified as total NERs that contain both the untransformed chemical and its transformation products entrapped in or bound to soil as xenoNERs as well as biomolecules derived from microbial biodegradation as bioNERs. Mineralization of the compound leads to its ultimate detoxification, while extractable and thus potentially mobile chemical residues could relocate to other environmental compartments and cause toxic effects. In contrast, the hazard of NERs is not clear as they can be ‘safe sink’ bioNERs type III and xenoNERs type II or potential ‘hidden hazard’ xenoNERs type I.

Comment 2 [97] D-labelling might be cheaper, but the analysis in biodegradation studies are as laborious as ^{15}N or ^{13}C label analysis & [160] currently there is no expensive and laborious bioNER assessment, but is just done by MTB model which is accepted in regulation for bioNER assessment.

Response: Thank you for this hint. Yes, we agree that biodegradation studies using stable isotopes are comparably laborious, regardless of the type of isotope used. However, when using D, compared to ^{13}C and ^{15}N , the laborious analytics of biogenic NERs can be skipped due to the minimal of D in biogenic NERs. As we discussed in the results section, both the total amino acids*2 and MTB modelling approaches do not provide robust estimates of total bioNERs in all cases (e.g. when a compound or its transformation products are used as monomeric building blocks for biomass). This limitation of the MTB approach has also been acknowledged by Trapp et al. (2020) in their discussion section ‘Flaws of the MTB and bioNER assessment’ [p.12 in Trapp, S., Brock, A. L., Kästner, M., Schäffer, A. & Hennecke, D. Critical evaluation of the microbial turnover to biomass approach for the estimation of biogenic non-extractable residues (NER). *Environmental sciences Europe* **34**, 15; 10.1186/s12302-022-00592-5 (2022)]. H-labelling approach could alleviate this issue to a large extent by directly measuring mostly xenoNERs. We clarified it in lines 452-459, pg 17-18, and below:

*‘The H-labeling approach could therefore solve the problem of the uncertainty associated with calculating total bioNERs based on tAAs*2 or the MTB approach. Although this proof-of-concept study demonstrated D-labeling as a powerful tool for time-efficient xenoNER quantification, it has several limitations. Compared to ^{13}C or ^{15}N tracer, D-labeling is undoubtedly cheaper in terms of associated costs of labeled compound, and less laborious considering that the bioNER analytics can be entirely skipped (Table 1). Yet, like other stable isotopes, the D-labeling requires equitably laborious incubations with multiple controls and quantitative as well as qualitative analytics.’*

Comment 3 [164] what is a standard C isotope tracers ? **Response:** the word ‘standard’ has been deleted, see line162, pg 6.

Comment 4 [185] no! The exhaustive extraction (that btw should end with an ASE extraction according to ECHA) aims to extract the extractable portion of a substance. If it would extract NER, which means NON EXTRACTABLE residue, this would not be NON EXTRACTABLE by definition. It does not aim to extract NER Type I. Please correct. & [187-189] That is simply wrong, please delete. After exhaustive extraction you get a material containing total NER. Total NER consist by definition of NER type I, II and III.

Response: Thank you for pointing this out. We deleted the original statement and now explain in more detail why we deviated from the standard extraction scheme. We did not use ‘harsh’ extractions using heat or pressure in order to minimize the risk of potential release of D from D-labelled compounds. Thus, it is possible that the ‘NER’ could still contain ‘slowly desorbable’ residues according to the ECHA guidance. However, this is only a proof-of-concept study to demonstrate the *potential* of using hydrogen tracers. It did not aim at following strictly all incubation conditions and extraction schemes of the regulatory guideline, but instead to prove that D-isotope is minimally retained in biogenic NERs, which could solve the limitations of the two current approaches – total amino acids*2 and MTB modelling. We totally agree that further studies – strictly according to guidelines – may be needed for this approach to find regulatory acceptance. We therefore pointed out that the ‘NERs’ measured in this study are operationally defined by the extraction scheme employed. Please see below the text that has been included in lines 184-203, pg 8:

*‘The extraction efficiencies of the tested compounds from the soil directly after spiking were as follows: $98\pm5\%$ (2,4-D), $93\pm3\%$ (GLP) and $80\pm10\%$ (SMX; for details see **Supplementary Text S3**). However, the ^{13}C and D_{NERs} measured on day 0 sampling of both sterile (**Fig. 3**) and biologically active soils (**Text S3**) suggest a lower extraction efficiency, especially of GLP and SMX. Due to the laborious preparation of all parallel experimental treatments for incubation, it was impossible to perform soil extractions immediately after spiking of the test compounds (see **Text S3**). Therefore, we cannot exclude abiotic NER formation already in soils sampled on day 0. Moreover, in order to minimize the risk of potential release of the D from C–D bonds, we skipped the final ‘harsh’ extraction step mandated by ECHA, which uses heat or pressure aiming at the extraction of ‘slowly desorbable’ residues¹⁷. The total ^{13}C and D_{NERs} from 2,4-D, GLP and SMX (**Fig. 3** and **4**) thus comprised the sum of bioNERs, xenoNERs type I & II, and possibly also ‘slowly desorbable’ residues¹⁹ which aggrandize total NER estimates. Although the ‘slowly desorbable’ residues are not considered ‘NERs’¹⁷, we kept the NER term for simplicity here. Please note that the current study was not performed for regulatory purposes but proof-of-concept, and hence it did not aim to accurately follow*

*all extraction schemes or soils incubation conditions outlined in OECD guideline 307¹⁶. The aim of this study was to compare: (I) amounts of both ¹³C- and D-compounds in sterile waters and ¹³C_{NERs} and D_{NERs} (as defined by the employed methodology) in sterile soils (**hypothesis 1: stable C–D bonds under abiotic conditions**), and (II) the ¹³C and D incorporations into bioNERs in biologically active soils (**hypothesis 2: minimal retention of D in bioNERs**) using the same experimental conditions and extraction protocols for both H- and C-tracers’.*

Comment 5 [200] what do NER have to do with sorption coefficients?? Sorption coefficients are valid for solid/water systems at equilibrium – but this is not the aim of an extraction. For extraction with solvents this does not apply or the solvent is not suitable. Please delete.

Response: deleted as requested.

Comment 6 Fig 3: nice figure! Was it triplicate measurement of the same sample or single measurement of triplicate samples? I assume it was sacrificial sampling? Maybe is described later.

Response: Thank you. The uncertainty is derived from single measurement of triplicate samples, each taken from a separately incubated bottle. It was not a sacrificial sampling due to the limited available amounts and high costs of isotope-labelled compounds, which partly had to be custom-synthesized for this study. Sacrificial sampling would have required at least four times higher amounts of each test compound, which was unfortunately not feasible. However, identical experimental conditions, sampling procedures and extraction protocols were applied for D and ¹³C-labelled analogues, and hence to us this setup seemed sufficient to provide valid proof-of-concept for the hypothesis that that D-isotope is minimally retained in biogenic NERs. We have also shown that the sampling method did not noticeably affect soil microbial activity. See lines 588-593 (pg 23), **Supplementary Text S8**, and below:

*‘Soil samples were taken from the same bottles on days 4, 16/18 and 36/38 for GLP and 2,4-D and on days 18, 36 and 72 for SMX because of its slower turnover. Each time, ten roughly 0.5-2 g subsamples were taken carefully from different spots within the soil batch in order to prevent soil disturbance. The sampling did not cause any noticeable disturbances in soil microbial activity as soil respiration showed a continuous decrease of CO₂ towards the end of the incubation periods (**Fig. S6 in Supplementary Text S8**)’.*

Comment 7 [220 ff] 2,4-D: any idea, why ¹³C-NER are not higher than in the sterile experiment? That’s unusual.

Response: Thank you for this comment. Although it is stated in the ECHA guidance that ‘*if abiotically formed NERs are much lower than biotically formed NERs, this gives a clear indication on bioNER formation*’, the reverse is not necessarily the case, i.e. abiotic NERs do not always have to be lower than biotic NERs. This is because biodegradation and abiotic NER formation occur simultaneously and competitively in biologically active soils. In the case of ¹³C_{6-2,4-D}, the NERs in biologically active soils were either comparable or lower than those in sterile soils. 2,4-D can be readily mineralized (~80%) as shown in this study; thus, biological process were much faster than abiotic processes explaining the lower biotic compared to abiotic NERs. We explained this finding in more detail in line 245-259, pg, 10-11; see also below:

*‘Both the total ¹³C_{NERs} and D_{NERs} in biologically active soils were either comparable (¹³C: day 16/18) or lower (¹³C: day 36; D: day 16/18 and 36) than in sterile soils (**Fig. 3a**). Although this finding might seem to contradict the statement that ‘*if abiotically formed NERs are much lower than biotically formed NERs, this gives a clear indication on bioNER formation*’¹⁷, bioNER formation does not necessitate that abiotic NERs are lower than NERs in biological active soil. This is because in biologically active soil, abiotic interactions leading to xenoNER formation compete with simultaneously proceeding biodegradation processes, which include both bioNER formation and complete mineralization of the compound. Thus, when a compound is quickly mineralized in biologically active soil, it may be removed before it can form xenobiotic NERs. In sterile soil, no mineralization occurs, and thus total abiotic NERs may be higher than biotic NERs. Interestingly, in a previous study by Girardi et al.⁵⁷, the ¹³C_{NERs} from ¹³C_{6-2,4-D} in sterile soils (15±1.8%) were lower than in biologically active soil (39±2.6%). However, much less ¹³C_{6-2,4-D} was mineralized in the study by Girardi et al.⁵⁷ (46±2.9% after 32 days) than in this study (78±8.8% after 36 days, see **Table S4** in*

Supplementary Text S5), suggesting that the rapid biodegradation of 2,4-D in our study prevented formation of xenoNERs in the biologically active soil’.

Comment 8 [235 ff] You got again not more NER than in the sterile experiment. But you have more than 60% ^{13}C NER after 4 days?? And less than half of that is D-NER. So more than 40% bioNER after 4 days?? That would be a very rapid degradation kinetic. Did you really measure such a rapid mineralisation that would go along with this (according to the MTB model)? Does not sound reasonable. In normal world (^{14}C), NER increase slowly over time, except there is very fast degradation or mainly abiotic processes. As shown in the sterile samples. But this does not match with the conclusion of having half of the NER as bioNER after such short time. Sure, that the extraction was efficient? Also for known transformation products?

Response: Thank you. Yes, we agree that these findings are quite surprising. Yes, the extraction of the glyphosate transformation product AMPA (91±2%) from the soil directly after spiking was efficient. Here, the (bio)NER formation from glyphosate was; however, dictated rather by the labelling position of $^{13}\text{C}/\text{D}$ (2 versus 3). See in lines 274-282, pg 11-12 and below:

‘Similar to 2,4-D, the total $^{13}\text{C}_{\text{NERs}}$ from both 2- $^{13}\text{C}_{\text{GLP}}$ and 3- $^{13}\text{C}_{\text{GLP}}$ in biologically active soils were comparable with the $^{13}\text{C}_{\text{NERs}}$ in sterile soils, whilst the D_{NERs} from their D-labeled counterparts were lower (Fig. 3b). Also in a prior study, the amounts of ^{13}C and $^{15}\text{N}_{\text{NERs}}$ (~25%) in biotic water-sediment incubated with $^{13}\text{C}_3^{15}\text{N}_{\text{GLP}}$ were comparable to those in abiotic controls³⁶. In this study, mineralization of GLP was similar between 3- $^{13}\text{C}_{\text{GLP}}$ (50±17%; Table S4) and 2- $^{13}\text{C}_{\text{GLP}}$ (40±12%) suggesting at first glance a comparable biodegradation of GLP labeled at the C2 and C3 positions. However, mineralization of a compound is not always the only factor dictating the total amounts of $^{13}\text{C}_{(\text{bio})\text{NERs}}$ as shown for $^{13}\text{C}_{\text{GLP}}$. The labeling position of GLP played a crucial role here as detailed in the following sections.’

For the 2-C labelling position of GLP, high biogenic NER formation is expected as the ^{13}C or D label of the parent glyphosate will be retained when the biodegradation follows the sarcosine pathway. Also, the biodegradation of 2-C-labelled glyphosate via the sarcosine pathway will occur without the release of CO_2 ; instead, the isotope label will be retained in glycine. As monomeric substrate utilization without CO_2 release is not accounted for in the MTB model, in the case of 2-C-glyphosate, the MTB model gives a too low estimates for ^{13}C -biogenic NER. We explained this in more detail in the text in a newly separated sub-section ‘Labeling position of GLP and its relevance for the ^{13}C and D-labeling pattern of bioNERs’. See below the copied entire section (pg 14-16):

‘Labeling position of GLP and its relevance for the ^{13}C - and D-labeling pattern of bioNERs

2-C-labeling position (2- C_{GLP}). A closer look at the composition of ^{13}C and D_{IAAs} revealed that the contents of $^{13}\text{C}_{\text{glycine}}$ (1.7±0.6% – 3.6±1.0% of applied ^{13}C ; **Supplementary Fig. S10a**) were comparable to those of D_{glycine} (1.9±0.6% – 2.7±1.5% of applied D) on all sampling days ($p>0.05$, **Text S4.1.4**), and both were fairly stable between day 4 and day 38. Nearly identical amounts of D_{glycine} and its ^{13}C -analogue suggest that its C–D bonds resisted harsh acidic hydrolysis. The hydrolyzed D_{IAAs} from the soil are thus reliable for tracking the D integration into C–D bonds of amino acids during microbial metabolism of a D-compound.

With $^{13}\text{C}_{\text{glycine}}$ comprising 16-30% of $^{13}\text{C}_{\text{IAAs}}$ and D_{glycine} 76-81% of D_{IAAs} , glycine was clearly the most predominant AA in the tAA pool (**Fig. S10a**). These findings are unique to 2- C_{GLP} due to the retention of the isotope label at 2-C position during degradation. When 2- C_{GLP} is degraded to sarcosine, which can then be oxidized to glycine in the sarcosine pathway (**Fig. S8**), both sarcosine and glycine will preserve the D- or ^{13}C -labels of the parent 2- C_{GLP} ^{36,64,67}. Thus, a disproportionally high integration of the isotope label from 2- C_{GLP} into glycine can be expected. $^{13}\text{C}_{\text{glycine}}$ and D_{glycine} from 2- $^{13}\text{C}_{\text{GLP}}$ and 2- $\text{C}-\text{D}_2\text{-GLP}$ were likely assimilated into microbial biomass as a monomeric ‘building block’ for proteins which is more energy-efficient than the biosynthesis of macromolecules derived from smaller C-precursors like acetyl-groups (**Fig. 2**)^{36,59}. $^{13}\text{C}_{\text{glycine}}$ and D_{glycine} could then have been partially mineralized to $^{13}\text{CO}_2$ and D_2O . The ^{13}C from $^{13}\text{CO}_2$ and the ^{13}C incorporated into other biomolecules of microbial degraders (1st level) was possibly recycled for the synthesis of new biomolecules by the consumers (2nd, 3rd, 4th level, etc.; **Fig. 2**). Therefore, other $^{13}\text{C}_{\text{amino acids}}$ than $^{13}\text{C}_{\text{glycine}}$ were also highly enriched in ^{13}C due to these ^{13}C -recycling processes. Notably, the share of D_{glycine} (76-81%) in the D_{IAAs} was much higher than that of its ^{13}C analogue (16-30%) suggesting that D was only minimally retained in other $D_{\text{amino acids}}$. Unlike ^{13}C , the D-label is rapidly released as D_2O after the cleavage of C–D bonds of either 2- $\text{C}-\text{D}_2\text{-GLP}$ or D-biomolecules of necromass (**Fig. 2**). The

D_2O is then diluted with unlabeled H_2O , leading to estimated D_2O concentrations in soil water of only 0.00012 – 0.00076% for the sarcosine pathway (**Fig. S1** in **Text S2**). Therefore, only low amounts of D -label could have been re-incorporated into the other D_{amino} acids. Overall, based on the shares of the two isotopes in the tAA pool and their recycling processes, we can deduce that GLP is preferentially degraded into the amino acid glycine in the sarcosine pathway, dictating possibly high $bioNER$ formation for $2-C_{GLP}$.

Still, a large proportion of the total $^{13}C_{NERS}$ from $2-^{13}C_{GLP}$ (64-71%; **Fig. 5b**) remained unidentified. We speculate that the $^{13}C_{bioNERS}$ for $2-^{13}C_{GLP}$ could have been underestimated due to the uncertainty of the $bioNER$ approximation based on the $tAAs*2$ and MTB approaches. Notably, $2-^{13}C_{glycine}$ formed from $2-^{13}C_{GLP}$ in the sarcosine pathway (**Fig. S8**) is directly incorporated into microbial biomass without the release of $^{13}CO_2$, as demonstrated by Wang et al.³⁶ The predicted $^{13}C_{bioNER}$ amounts for $2-^{13}C_{GLP}$ (6.7– 16% for the sarcosine pathway) based on the measured $^{13}CO_2$ were thus likely underestimated, showing limitations of the MTB approach when monomeric substrate utilization occurs (**Fig. 2**). Another good example proving the limitations of both the $tAAs*2$ and MTB approaches can be taken from a recent degradation study of $^{13}C_2$ -glycine by Aslam et al.⁶⁴, where 37% of $^{13}C_2$ -glycine was measured in $^{13}CO_2$, 8.7% in $^{13}C_{tAAs}$, and 34% in total $^{13}C_{NERS}$. Based on the $^{13}C_{tAAs}*2$, about 17.4% of total $^{13}C_{bioNERS}$ were formed which comprised 51% of the total $^{13}C_{NERS}$; thus, the other 49% of $^{13}C_{NERS}$ was unidentified. The total $^{13}C_{bioNER}$ amounts estimated for $^{13}C_2$ -glycine using the MTB model were underestimated as well (min: 4% and max: 8.8%, **Table S6**). Glycine is an easily biodegradable biomolecule⁶⁸, therefore, it cannot form $^{13}C_{xenoNERS}$ and all of the $^{13}C_{NERS}$ from $^{13}C_2$ -glycine should be exclusively $^{13}C_{bioNERS}$.

$2-^{13}C_{GLP}$ or $2-C-D_2-GLP$ is degraded to unlabeled AMPA in the AMPA pathway (**Fig. S9**). Therefore, a large portion of unidentified $^{13}C_{NERS}$ from $2-^{13}C_{GLP}$ could stem either from unidentified $^{13}C_{bioNERS}$ not considered in the MTB or $tAAs*2$ calculation, or from the un-degraded parent molecule which could be the primary source for $^{13}C_{xenoNERS}$.

3-C-labeling position (3- C_{GLP}). Contrastingly, neither $^{13}C_{glycine}$ nor $D_{glycine}$ was the dominant amino acid within the tAA pool of $3-C_{GLP}$ (**Fig. S10b**). When the $3-C_{GLP}$ is degraded via the sarcosine pathway, the major resulting degradation product glycine will be unlabeled (**Fig. S9**), explaining the much lower amounts of $^{13}C_{bioNERS}$ and $D_{bioNERS}$ compared to $2-C_{GLP}$. Although the contents of $^{13}C_{glycine}$ were not significantly different from those of $D_{glycine}$ for $3-C_{GLP}$ on all sampling days ($p > 0.05$, see **Text S4.1.4**); their amounts were at least 15-fold lower than for $2-C_{GLP}$. The percentages of both $^{13}C_{glycine}$ (1.1 – 15% of $^{13}C_{tAAs}$) and $D_{glycine}$ (3.2 – 25% of D_{tAAs}) in the tAA pool were also comparatively lower than for $2-C_{GLP}$.

In contrast to $2-C_{GLP}$, $3-C_{GLP}$ is degraded to labeled AMPA ($^{13}C_{AMPA}$ or D_2-AMPA) in the AMPA pathway (**Fig. S9**) as the third C of GLP is preserved in AMPA. As ^{13}C and $D_{bioNERS}$ were lower than for $2-C_{GLP}$, higher portions of ^{13}C and $D_{xenoNERS}$ could have been formed from $3-C_{GLP}$. Since similar behavior of the parent compounds is expected, this could be due to abiotic interactions between labeled $^{13}C_{AMPA}$ or D_2-AMPA with reactive groups of soil'.

Comment 9 [246] Adsorption of AMPA to the soil? Then the extraction was not efficient. Adsorbed (see comment for line 200) substance should be released during extraction.

Response: The entire sentence 'adsorption of AMPA to the soil' has deleted.

Comment 10 [258] yes right! But this applies also to the other two substances: NER in biotic are very similar to the abiotic experiments for ALL substances. This is somehow suspicious. But for SMX I totally agree.

Response: For 2,4-D it was due to a competition of a rapid 2,4-D biodegradation with abiotic interactions leading to xenobiotic NER formation (which are usually slower). In the case of glyphosate the labelling position was the key factor determining the amounts and type of NER formation. We have already explained it in detail in our responses to **Comment 7** and **8**.

Comment 11 [262 ff] so conclusion is that almost entire NER for 2,4-D is due to $bioNER$? What is then NER measured in the sterile experiment? This cannot be $bioNER$. So this must be an abiotic NER formation in the sterile experiment - why does it not happen in the non-sterile experiment? Overestimation of $bioNER$ by $tAAs$?

Response: Yes, we expect that for the rapidly mineralized 2,4-D (~80%) almost all $NERs$ are biogenic. The results of the MTB model also suggest high biogenic NER formation (min: 13.5%, max: 32.6%), which is

similar to the estimate based on ^{13}C -amino acids*2 ($20\pm 3.2\%$). $\text{BioNER}_{\text{min}}$ (13.5%) were also nearly identical to the total ^{13}C -NERs ($14\pm 4.7\%$), which were a bit lower than the estimated total bioNERs based on tAAs*2. Hence, the conversion factor of 2 may indeed have been a bit too high for 2,4-D experiment on the final day (it may be only ~ 1.5), but we still expect all NERs from 2,4-D to be biogenic on the final day. In the sterile experiment there must be exclusively xenobiotic NERs; we explained in detail in our response to **Comment 7** that biological processes can be faster than abiotic processes of NER formation. As 2,4-D was rapidly mineralized; the resulting NERs were low in biotic soils than the NERs in sterile soils.

Comment 12 [325 ff] bioNER are rather underestimated? How does this match with the results of the sterile experiments?? Do all those processes not happen in non-sterile soil? Is there any other proof for higher bioNER like eg extensive mineralisation that could predict more bioNER by means of the MTB model? So why underestimated?

Response: Yes, we expect that bioNERs are underestimated for $2\text{-}^{13}\text{C}$ -labeled glyphosate due to the uncertainty of both the total amino acids*2 and MTB model approaches. When $2\text{-}^{13}\text{C}$ -labeled glyphosate is biodegraded via the sarcosine pathway, the ^{13}C -label from the parent molecule will be contained in the major intermediate $2\text{-}^{13}\text{C}$ -glycine and without the release of $^{13}\text{CO}_2$. The calculation of biogenic NERs using the MTB model relies on measured CO_2 , thus biogenic NERs in the $2\text{-}^{13}\text{C}$ -labeled glyphosate are possibly underestimated. We showed an example of the uncertainty of these two approaches for $^{13}\text{C}_2$ -glycine, see in line 390-404, pg 15-16, and below:

*'Still, a large proportion of the total $^{13}\text{C}_{\text{NERs}}$ from $2\text{-}^{13}\text{C}_{\text{GLP}}$ (64-71%; **Fig. 5b**) remained unidentified. We speculate that the $^{13}\text{C}_{\text{bioNERs}}$ for $2\text{-}^{13}\text{C}_{\text{GLP}}$ could have been underestimated due to the uncertainty of the bioNER approximation based on the tAAs*2 and MTB approaches. Notably, $2\text{-}^{13}\text{C}_{\text{glycine}}$ formed from $2\text{-}^{13}\text{C}_{\text{GLP}}$ in the sarcosine pathway (**Fig. S8**) is directly incorporated into microbial biomass without the release of $^{13}\text{CO}_2$, as demonstrated by Wang et al.³⁶ The predicted $^{13}\text{C}_{\text{bioNER}}$ amounts for $2\text{-}^{13}\text{C}_{\text{GLP}}$ (6.7– 16% for the sarcosine pathway) based on the measured $^{13}\text{CO}_2$ were thus likely underestimated, showing limitations of the MTB approach when monomeric substrate utilization occurs (**Fig. 2**). Another good example proving the limitations of both the tAAs*2 and MTB approaches can be taken from a recent degradation study of $^{13}\text{C}_2\text{-glycine}$ by Aslam et al.⁶⁴, where 37% of $^{13}\text{C}_2\text{-glycine}$ was measured in $^{13}\text{CO}_2$, 8.7% in $^{13}\text{C}_{\text{LAAs}}$, and 34% in total $^{13}\text{C}_{\text{NERs}}$. Based on the $^{13}\text{C}_{\text{LAAs}}$ *2, about 17.4% of total $^{13}\text{C}_{\text{bioNERs}}$ were formed which comprised 51% of the total $^{13}\text{C}_{\text{NERs}}$; thus, the other 49% of $^{13}\text{C}_{\text{NERs}}$ was unidentified. The total $^{13}\text{C}_{\text{bioNER}}$ amounts estimated for $^{13}\text{C}_2\text{-glycine}$ using the MTB model were underestimated as well (min: 4% and max: 8.8%, **Table S6**). Glycine is an easily biodegradable biomolecule⁶⁸, therefore, it cannot form $^{13}\text{C}_{\text{xenoNERs}}$ and all of the $^{13}\text{C}_{\text{NERs}}$ from $^{13}\text{C}_2\text{-glycine}$ should be exclusively $^{13}\text{C}_{\text{bioNERs}}$.'*

Comment 13 [342] Yes! And that is very similar finding to the sterile experiments. Question stays still, why SMX is so good in line with the sterile experiments and the other two not at all? And there is still the gap between ^{13}C and D in total NER: why is D always below the ^{13}C ? And we talk about a difference of more than 20% which is in magnitude – not relative of course – similar to the difference observed for 2,4-D. This were the concerns from the first manuscript draft. & [374] But H-NER are significantly lower than ^{13}C -NER (and ^{14}C -NER I would guess) – the observed difference of 20% can be highly relevant in persistency assessment. I doubt that this is acceptable by regulation.

Response: Thank you. We explained the differences between the three compounds in our responses to **comment 7 & 8**. The observed 20% difference for $^{13}\text{C}_6\text{-SMX}$ was not significant statistically, and we have explained that we suspect an unnoticed experimental error during the preparation of the spiking solution of $^{13}\text{C}_{\text{SXM}}$, which was much harder to dissolve than unlabelled SMX or D_{SMX} . See also lines 283-291 (pg 12), **Text S3**, and below:

*'SMX formed very high amounts of total NERs (**Fig. 4c**). The D_{NERs} ($68\pm 5.6\%$ of the applied D) appeared slightly lower than $^{13}\text{C}_{\text{NERs}}$ ($90\pm 6.8\%$ of the applied ^{13}C) on day 18 ($p_{\text{adj}}=0.059$), but no significant differences in NER contents were found between D and ^{13}C tracers on any sampling day ($p_{\text{adj}}>0.05$, **Fig. 4c**). The $^{13}\text{C}_{\text{NERs}}$ peaked at 107 ± 12 to $115\pm 9.7\%$ and the D_{NERs} at 88 ± 5.2 to $82\pm 10\%$ on day 36-72. Notably, higher $^{13}\text{C}_{\text{NER}}$ values compared to D_{NERs} might be due to a higher amount of $^{13}\text{C}_6\text{-SMX}$ accidentally added to the soil on day 0 (total ^{13}C -label recovery on day 0: 130%, **Table S2a in Text S3**). SMX is a hardly biodegradable antibiotic (only $2.3\pm 0.5\%$ $^{13}\text{CO}_2$ after 72 days, **Table S4**) expected to form mostly xenoNERs;*

thus, the $^{13}\text{C}_{\text{NERs}}$ in the biotic treatment should be nearly identical to the $^{13}\text{C}_{\text{NERs}}$ and D_{NERs} measured in sterile soil (Fig. 3c).’

Biotic D_{NERs} from SMX (Fig. 4c) were very close to both abiotic D_{NERs} and abiotic $^{13}\text{C}_{\text{NERs}}$ (Fig. 3c), while only the biotic $^{13}\text{C}_{\text{NERs}}$ are somewhat higher – also clearly higher than the abiotic $^{13}\text{C}_{\text{NERs}}$ and above 100% of the nominally added concentration. In the abiotic experiments, $^{13}\text{C}_{\text{SMX}}$ could be spiked with a pure methanol solution as toxicity to microorganisms was not relevant; hence we expect that the discrepancy between D_{NERs} and $^{13}\text{C}_{\text{NERs}}$ in biotic soil were specific to the compound SMX and how it was spiked in the biotic experiments. We agree that the current results may leave some doubt about regulatory applicability due to the 20% gap, and therefore, further studies with an additional set of compounds may be needed. Possibly, also the higher uncertainty in stable isotope compared to ^{14}C tracing may have contributed. Therefore, in further test beyond this proof-of-concept study, ideally radioactive H-labelling should be used which may offer an easier and more reliable quantitative analysis than stable isotope tracing.

Comment 14 [351] the three factors don’t explain the sterile sample findings. You should refer more to that sterile results as they are important also for process analysis.

Response: Thank you for this comment. We also explained the differences in NER formation in sterile and biologically active soils for 2,4-D in our response to **comment 7** and glyphosate in our response to **comment 8**. In order to avoid confusion, we corrected this sentence as follows:

‘As hypothesized, the C–D bonds of all tested compounds were stable against abiotic cleavage for the duration of soil incubations as demonstrated for D_{NERs} in sterile soils (Fig. 3) and for D-compounds in sterile water solutions (Supplementary Text S6). Also, as assumed, the D tracer was far less retained in microbial biomass than the ^{13}C tracer, which can be attributed to three factors:’. Line 427-430, pg 17.

Comment 15 [365] yes, absolutely true! Completely agree. **Response:** oh yeah!

Comment 16 Table 1: there is something wrong with the table:

cost: to my experience ^{14}C is much more expensive than ^{13}C . did you confuse that here?
safety concerns: here is everything mixed! ^{13}C is not radioactive, and ^{14}C has the half life of 5700 years. For T radioactivity is a concern. The penetration depth is of no relevance, they are both beta radiators with neglectable dose rate. The only concern is incorporation. Just let it out in the table, please.

work load: same work load for ^{14}C and T as same instrumentation and principles are used. D should have higher workload. Again columns seems to be mixed. NER analytics: seems also mixed, didn’t you use D labels for quantification of biomolecules? **Response:** Table 1 has been updated, see lines 461-462, pg 18.

Comment 17 [403] This goes beyond the scope of the manuscript. If T labelling would have such advantages over ^{14}C -labelling, it would have been used in regulation. But in contrast ^{14}C has prevailed and very less labs are able to work with T (which needs a special radio license and cannot simply be applied in ^{14}C -labs). This is due to the uncertainty of the H-shift between molecules. This is the major problem of T-labelling and even the authors put some effort to prove stability of the current C-H bonds, see also 424 ff. Safety concerns by penetration depths and half-life are irrelevant because of the very low dose rate. Incorporation is the major concern and this should be worse for T connected to water as result of mineralisation. CO_2 is assimilated by plants but not humans, water can be incorporated right away. Thus, handling with T is even more tricky than with ^{14}C ! The weaker beta energy means also weaker detectability. ‘Finally CO_2 from combustion analysis can be easily trapped in NaOH-solution – trapping of water in a solvent is more tricky. No, T is not the better radioactive label as proven in 30 years of regulatory practice! The authors expertise is stable isotope labelling – please stay in this topic and delete this chapter about the radiolabels. **Response:** the text related to costs, workload and safety concerns has been deleted. Table 1 has also been updated.

Comment 18 [422] agree. These kinetic isotope effects are generally neglected in regulatory testing. **Response:** yes, we agree!

Comment 19 [442 ff] bit too optimistic? Agree, it could be added to the ^{14}C -label (and would produce additional radioactive waste). 12 or 5700 years half-life would not fundamentally change the waste problem. All wastes have to be treated as radioactive waste following the same rules. 12 years might rather be a problem for long-term studies – e.g. lysimeter studies have a duration of up to 3 years – where a half-life of 12 years will be noticeable already and makes evaluations more complicated. **Response:** the text related to half-life and waste has been deleted.

Comment 20 Prove of concept: yes. Is a very nice idea and results indicate that this could be a chance to move forward with the bioNER problem. But not yet ready for application. This will need further research. **Response:** Thank you. We agree that the H-labeling approach could help to solve biogenic NER issue, especially that the currently available two approaches (total amino acids*2 and MTB model) approximating biogenic NERs are not consistent in all cases, e.g. as shown for $^{13}\text{C}_2$ -glycine (our response to **Comment 8**). But we also agree that further research is still needed. We explained it in more detail in the text, see below:

*'The H-labeling approach could therefore solve the problem of the uncertainty associated with calculating total bioNERs based on tAAs*2 or the MTB approach. Although this proof-of-concept study demonstrated D-labeling as a powerful tool for time-efficient xenoNER quantification, it has several limitations. Compared to ^{13}C or ^{15}N tracer, D-labeling is undoubtedly cheaper in terms of associated costs of labeled compound, and less laborious considering that the bioNER analytics can be entirely skipped (Table 1). Yet, like other stable isotopes, the D-labeling requires equitably laborious incubations with multiple controls and quantitative as well as qualitative analytics.'* Line 452-459, pg 17-18.

'Therefore, to conduct biodegradation studies and NER identification at environmentally relevant concentrations, the radioactive H-isotope – tritium (T) – could be applied as a promising substitute for D. Unlike for stable isotope tracers, a radiolabeled compound can typically be mixed with its unlabeled analogue to the desired spiking concentration and still give quantifiable signal. Yet, T-labeling has not been favored for regulatory testing of chemicals due to concerns about the potential abiotic release of T from T-labeled compounds. Substrate-H bound to O, S or N is generally considered exchangeable, whereas many C–H bonds are stable unless enzymatically cleaved by microorganisms^{38,71}. However, the stability C–H bonds may be compromised depending on their position within a molecule or ambient conditions (Text S6). We therefore recommend to carefully select appropriate C–H label positions that are stable against abiotic cleavage and, if needed, perform complementary stability tests in water or abiotic soil. Our proof-of-concept study demonstrated a good stability of the C–D bonds for three selected model compounds even during acidic hydrolysis. The acidic hydrolysis is conducted under 'harsh' conditions; thus, C–H bonds could also resist the last 'harsh' step of soil extraction remobilizing 'slowly desorbable' residues¹⁷. As negligible amounts of the D-isotope from all tested D-compounds were quantified in amino acids, the same is expected for T-isotopes. T-labeling could thus substitute D-labeling as an alternative to carbon tracers for future xenoNER quantification. Moreover, T-tracing could potentially be a more accessible approach than stable isotope probing due to the easier and more rapid quantitation of radioactivity in (xeno)NERs using standard liquid scintillation counters (LSC)^{15,16,19–21,31}. Nevertheless, prior to the potential application of T-labeling for future xenoNER quantification, the stability of C–T bonds under 'harsh' extractions still needs to be verified.' Line 470-490, p 18-19.

Comment 21 [470] wow, was very dry during storage! & [482ff] soil incubation experiments: the soil was dried at 40°C? And moistened only just before application of the test substances?? No preincubation at the incubation conditions? And sampling after 4 days? This is not at all representative for the soil. There must have been dramatic changes in the soil, when surviving microorganisms start to re-establish in the before dried soils. Microflora is not comparable to that of the original soil. & [503] 60% of the WHK is pretty wet in such silty soil and rather the upper moisture limit recommended in soil degradation experiments (40-60% WHK according to e.g. OECD 307).

Response: Please note that the aim of this proof-of-concept study was not to follow strictly all OECD 307 incubation guidelines as mentioned in our response to **Comment 4**. Yes, we agree that soil was very dry. However, the evaporation was needed to ensure a better D_2O detection for establishing a D-isotope mass balance (which was beyond the focus of this proof-of-concept study). Although the microbial activity might be changed due to drying of the soil, the mineralization & biogenic NER results were quite comparable to

the study conducted by Aslam et al. (2023) using the same reference soil but without drying it. The early day 4 sampling was planned only for glyphosate to track an early incorporation of the isotope label into glycine in the sarcosine pathway.

Comment 22 [519] was there any storage stability testing of the substance in the freezer? How long was the storage before analysis? **Response:** The substance was surely stable/unchanged as it has been prepared on the same day when the incubation vessels were prepared, i.e. it was not stored prior to spiking of the reference soil. We stated that also in lines 574, p 23.

Comment 23 [suppl S7] If there was total CO₂ determination, why no differentiation in ¹³CO₂? Would have been nice to have an idea on substance mineralisation, since current practice (MTB) is to calculate bioNER based on substance mineralisation. **Response:** we included endpoint mineralization data in **Supplementary Table S4** as well as our biogenic NER calculations using the MTB model in the **Supplementary Text S5**; both of them were also included and discussed in the manuscript text.

Reviewer #2 (Remarks to the Author):

After careful revision of the manuscript and thorough consideration of the author comments, I believe the manuscript is now suitable for publication. **Response:** Thank you very much for your support! We really appreciate it.